## TOOLS

# Engineered kinases as a tool for phosphorylation of selected targets in vivo

Katarzyna Lepeta[1], Chantal Roubinet[2], Milena Bauer[1], M. Alessandra Vigano[1], Gustavo Aguilar[1], Oguz Kanca[3], Amanda Ochoa-Espinosa[4], Dimitri Bieli[5], Clemens Cabernard[6], Emmanuel Caussinus[7], and Markus Affolter[1]

**Reversible protein phosphorylation by kinases controls a plethora of processes essential for the proper development and homeostasis of multicellular organisms. One main obstacle in studying the role of a defined kinase–substrate interaction is that kinases form complex signaling networks and most often phosphorylate multiple substrates involved in various cellular processes. In recent years, several new approaches have been developed to control the activity of a given kinase. However, most of them fail to regulate a single protein target, likely hiding the effect of a unique kinase–substrate interaction by pleiotropic effects. To overcome this limitation, we have created protein binder-based engineered kinases that permit a direct, robust, and tissue-specific phosphorylation of fluorescent fusion proteins in vivo. We show the detailed characterization of two engineered kinases based on Rho-associated protein kinase (ROCK) and Src. Expression of synthetic kinases in the developing fly embryo resulted in phosphorylation of their respective GFP-fusion targets, providing for the first time a means to direct the phosphorylation to a chosen and tagged target in vivo. We presume that after careful optimization, the novel approach we describe here can be adapted to other kinases and targets in various eukaryotic genetic systems to regulate specific downstream effectors.**

## Introduction

Reversible protein phosphorylation by kinases constitutes one of the most common regulatory mechanisms used to control cell signaling across eukaryotes. There are about 500 protein kinases encoded in the mammalian genome and about 250 in *Drosophila* (Manning et al., 2002; Morrison et al. 2000). Eukaryotic protein kinases catalyze the transfer of a phosphate group from ATP to the substrate protein to control protein activity and localization, providing rapid information transfer in the cell in response to an appropriate signal (Cohen 2002). Consequently, protein kinases orchestrate multiple intracellular processes including cell fate determination, cell migration, cell division, and response to environmental stimuli. In multicellular animals, protein kinases are essential for proper development and homeostasis. Dysregulation of kinase signaling is a hallmark of numerous diseases including cancer (Endicott et al. 2012; Cohen 2001). Thus, synthetic approaches for regulating kinase activity, be it for dissecting cell signaling pathways, for building synthetic biology modules, or for therapeutic purposes, are of great need (McCormick et al., 2020).

According to their substrate specificity, eukaryotic protein kinases are generally subdivided into three groups: serine/threonine kinases, tyrosine kinases, and dual-specificity kinases (Miller and Turk 2018; Manning et al., 2002; Du and Lovly 2018). In vivo, kinases are part of complex signaling networks. In most cases, a single kinase phosphorylates numerous targets, which in response regulate different cellular processes. This multiplicity of targets deters the study of specific kinase–substrate interactions in vivo. Hence, regulating the phosphorylation of a single protein target and dissecting the resulting phenotype is both a very desirable and a very challenging task. In recent years, several new approaches for manipulating kinase activity or localization have been developed. These include small molecule kinase inhibitors, engineered isolated kinase domains, or light-controlled regulation of kinase activity (extensively reviewed in McCormick et al. [2020]; Leopold et al. [2018]; Zhang et al. [2009]; Gil et al. [2020]). However, although some of these approaches allow for precise temporal and spatial control of the activity of a given kinase (opto-based tools), most of them fail to regulate a single protein target, likely causing pleiotropic effects, thereby hiding the effect of a unique kinase–substrate interaction. In addition, optical manipulation is only possible up to a limited tissue depth, precluding the use of opto-based tools in non-light permeable tissues (Leopold et al., 2018; Fleming et al., 2019; Karginov et al., 2014).

[1]Biozentrum, University of Basel, Basel, Switzerland;   [2]MRC Laboratory for Molecular Cell Biology, University College London, London, UK;   [3]Department of Molecular and Human Genetics, Baylor College of Medicine, Houston, TX;   [4]Department of Biomedicine, University of Basel, Basel, Switzerland;   [5]Mabylon AG, Schlieren, Switzerland; [6]Department of Biology, University of Washington, Seattle, WA;   [7]1 Avenue Georges Ferrenbach, Kaysersberg-Vignoble, France.

Correspondence to Katarzyna Lepeta: katarzyna.lepeta@unibas.ch;   Markus Affolter: markus.affolter@unibas.ch;   Emmanuel Caussinus: emmanuel.caussinus@free.fr.

**Rockefeller University Press**
J. Cell Biol. 2022 Vol. 221 No. 10    e202106179

**https://doi.org/10.1083/jcb.202106179**    1 of 28

In the last decade, multiple small, genetically encodable protein binders against fluorescent proteins have been characterized. These binders are derived from either single chain antibodies, so-called nanobodies (reviewed in Helma et al. [2015]; Bieli et al. [2016]), designed ankyrin repeat proteins (DARPins; Saerens et al., 2005; Brauchle et al., 2014; Plückthun. 2015), or from other scaffolds (Škrlec et al., 2015; Jost and Plückthun. 2014), and can be functionalized by fusing them to various effector domains, providing a growing toolkit to study protein function more directly in living organisms (extensively reviewed in Aguilar et al. [2019]; Harmansa et al. [2017]; Gil et al. [2020]; Lepeta [2022]).

Here, we characterize protein binder-based engineered kinases for direct, robust, and tissue-specific phosphorylation of the target fluorescent protein fusions in vivo. To assess the general applicability of our approach, we have created two synthetic kinases belonging to either serine/threonine (S/T) kinases or tyrosine (Y) kinases, the two main substrate-based kinase groups.

We first used Rok kinase, the *Drosophila* ortholog of Rho-associated protein kinase (ROCK; Mizuno et al., 1999; Winter et al., 2001), which is essential for myosin II activation. Rok is a serine/threonine (S/T) kinase regulated by the small GTPase Rho1 (Riento and Ridley. 2003). To generate an engineered Rok kinase that allows for targeted fluorescent fusion protein phosphorylation, we fused the catalytic domain of Rok kinase with different protein binders: a nanobody directed against GFP and its close derivatives (vhhGFP4; Saerens et al., 2005), a destabilized GFP-binding nanobody (dGBP1; Tang et al., 2016), or a DARPin against mCherry (2m22; Brauchle et al., 2014). We show that synthetic Rok kinases are functional enzymes and can robustly activate myosin II through phosphorylation of Sqh::GFP or Sqh::mCherry in different morphogenetic processes in a developing fly embryo, namely in dorsal closure and during the development of the tracheal system. Tissue-specific expression of the engineered Rok kinase in Sqh::GFP animals leads to the formation of clusters of activated actomyosin around the Sqh foci. This provoked ectopic tension forces that perturbed cell behavior and resulted in defects of dorsal closure and tracheal branching.

To validate our novel approach, we also generated engineered Src kinases based on the *Drosophila Src oncogene at 42A* (*Src42A*), one of the two homologs of mammalian Proto-oncogene tyrosine-protein kinase Src (c-Src; Simon et al., 1985; Takahashi et al., 1996). To generate synthetic versions of Src, we fused the destabilized GFP-binding nanobody (dGBP1) with the C-terminal part of Src spanning the ATP-binding site, the Src catalytic domain, and the C-terminal regulatory tyrosine residue with mutated Y511F. We showed that engineered Src robustly phosphorylates GFP-tagged Bsk, providing evidence for a second engineered nanobody-based kinase to efficiently phosphorylate a tagged substrate in a chosen tissue in vivo.

## Results

### Design and expression of engineered N-Rok kinase for targeted protein phosphorylation

As a proof of concept for a synthetic kinase that allows for targeted fluorescent fusion protein phosphorylation (Fig. 1 a), we first used Rok kinase, the *Drosophila* ortholog of Rho-associated protein kinase (ROCK; Mizuno et al., 1999; Winter et al. 2001). In its inactive state, the C-terminal Pleckstrin homology (PH) domain of Rok inhibits the activity of the N-terminal kinase domain (N-Rok). Upon binding of the activated form of Rho1, Rok changes its conformation and N-Rok becomes active (Amano et al., 1999). Rok acts in synergy with myosin light-chain kinase (MLCK) to activate Spaghetti squash (Sqh/Myosin II regulatory light chain) with a sequential phosphorylation on Ser21 and Thr20, which in turn activates myosin II (Mizuno et al., 1999; Winter et al., 2001; Verdier et al., 2006; Fig. 1 b). Myosin II is the major contractile protein complex of non-muscle cells and as such a prominent target of Rok/ROCK. The other known substrates of Rok are Myosin-binding subunit, Diaphanous/Diaphanous-related Formin, α-Adducin, LIM-kinase 1, MAP2, Tau, and Combover (Verdier et al., 2006; Mizuno et al., 2002; Fagan et al. 2014; Fukata et al., 1999; Tominaga et al., 2000).

To generate an engineered Rok, we cloned a fusion construct of the N-terminal catalytic domain of *Drosophila* Rho-kinase (hereafter called N-Rok; Fig. 1 c) with a small fluorescent protein binder recognizing either GFP or mCherry (Fig. 1, d–f). We deleted the C-terminal part of Rok since its removal generates a constitutively active form of the latter (Amano et al., 1999). We generated a series of constructs to express various combinations of N-Rok and either vhhGFP4 (GFP nanobody; Saerens et al., 2005) or 2m22 (mCherry DARPin; Brauchle et al., 2014) under the control of the UAS-Gal4 system (Brand and Perrimon. 1993; Fig. 1, d and f and Data S1—FASTA file). In N-RokDead::vhhGFP4 and N-RokDead::2m22, the catalytic domain was rendered inactive by a K116G single amino acid substitution (Winter et al., 2001). Similarly, we engineered a N-Rok::dGBP1 fusion that harbors a destabilized GFP-binding nanobody (dGBP1; Tang et al., 2016), which is stable upon binding of GFP or its close derivates but degraded in their absence (Fig. 1 e).

To test the functionality of our engineered kinases, we transfected a stable *Drosophila* S2 cell line expressing Sqh::GFP (S2 Sqh::GFP; Rogers et al., 2004) with pActin_Gal4 and a series of plasmids expressing Rok variants, namely pUASTattB_N-Rok::vhhGFP4-HA, pUASattB_N-RokDead::vhhGFP4-HA, pUASTattB_N-Rok-HA, and pUASattB_N-RokDead-HA. In the interphase, unphosphorylated Sqh localizes mainly to the cytoplasm and its phosphorylation at mitotic entry triggers cortical recruitment (Royou et al., 2002). To address whether the expression of engineered Rok was sufficient to recruit Sqh::GFP to the cell cortex, where Rok normally exerts its function, we documented the subcellular localization of Sqh::GFP by confocal microscopy in interphasic cells 40 h after transfection.

In mock-transfected cells, the Sqh::GFP signal was predominantly cytoplasmic with no obvious cortical localization (Fig. 2 a). In addition to this signal, we observed a prominent cortical enrichment of Sqh::GFP upon the expression of N-Rok::vhhGFP4-HA (Fig. 2, a and b). By contrast, cells expressing the non-functional construct (N-RokDead::vhhGFP4-HA) displayed similar Sqh::GFP levels to those seen in mock-transfected cells (Fig. 2, a and b). Importantly, the expression of Rok lacking the fused GFP nanobody (activated N-Rok-HA or kinase-dead

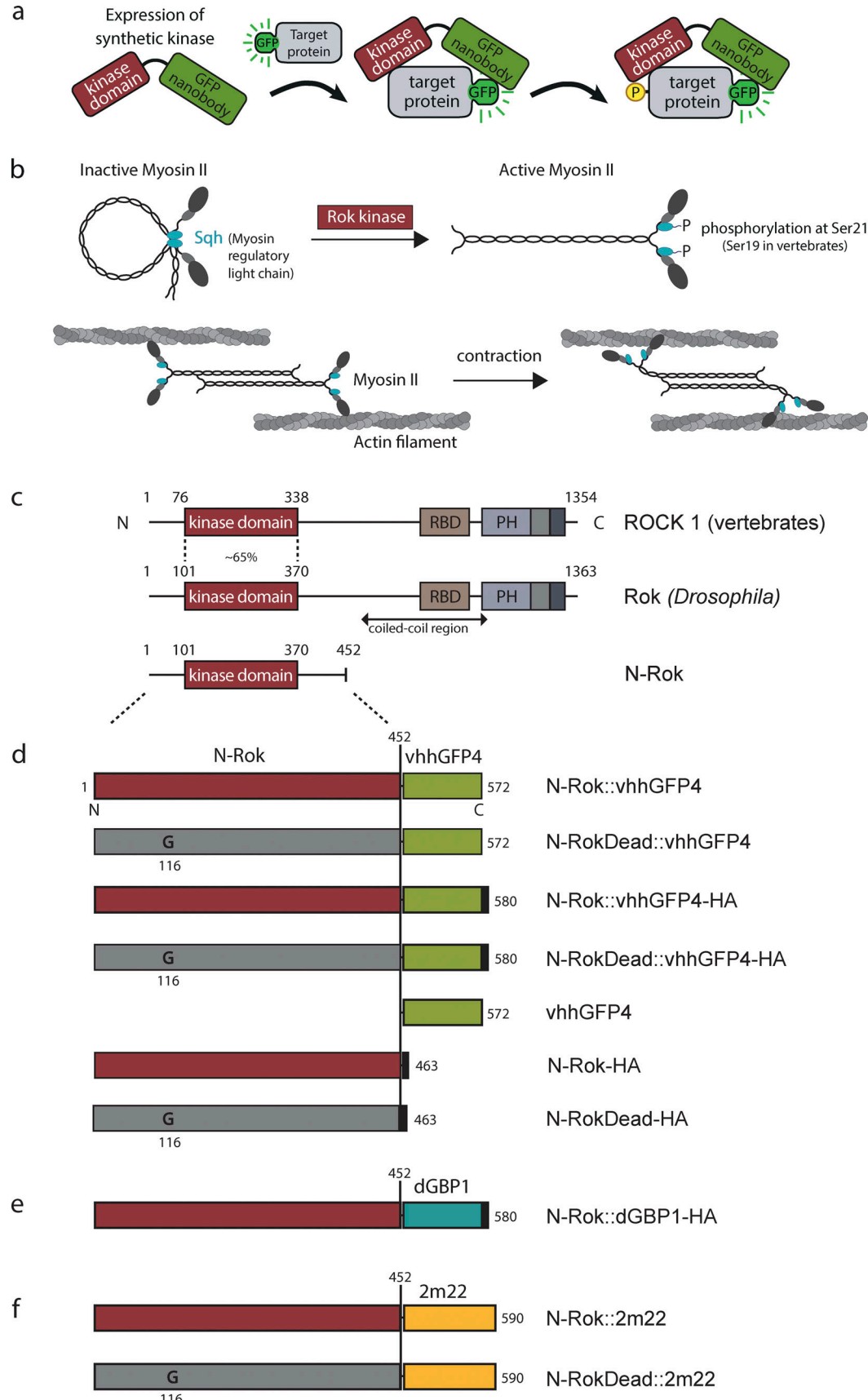

Figure 1. **Schematic illustration of engineered kinases. (a)** Overview of engineered kinase work concept. A synthetic kinase uses a small fluorescent protein binder (here: vhhGFP4, a GFP nanobody) to bring a constitutively active kinase domain (kinase) in close proximity to a fluorescent fusion protein (target). The

persistence of the kinase domain around the fluorescent fusion protein allows for efficient phosphorylation (P) of the target. **(b)** Schematic illustration of non-muscle myosin II activation and actomyosin contractility. Non-phosphorylated Sqh/Myosin-II regulatory light chain (MRLC) assembles into an inactive compact molecule through a head to tail interaction. Reversible phosphorylation of Sqh at Ser21 (Ser19 in mammalian MRLC) results in myosin II molecule unfolding, allowing association with other myosin II molecules in an anti-parallel fashion and binding to the actin filaments through the head domains. An ATP-dependent conformational change in myosin II drives the actin filament sliding in an anti-parallel manner and results in contraction. **(c)** The structure of mammalian and *Drosophila* ROCK proteins. The *Drosophila* Rok kinase region shares ∼65% identity with the corresponding isolated domain of mammalian ROCK1 (Verdier et al., 2006). **(d–f)** The N-terminal kinase region (N-Rok) of *Drosophila* Rok (amino acid 1–452) was used for synthetic kinase constructs shown in (d–f). **(d)** Linear representation of N-Rok::vhhGFP4, N-RokDead::vhhGFP4, N-Rok::vhhGFP4-HA, N-RokDead::vhhGFP4-HA, vhhGFP4, N-Rok-HA and N-RokDead-HA. **(e)** Linear representation of N-Rok::dGBP1, in which a destabilized GFP-binding nanobody (dGBP1; Tang et al., 2016) substitutes vhhGFP4 from d. **(f)** Linear representation of N-Rok::2m22 and N-RokDead::2m22 in which a DARPin recognizing mCherry substitutes vhhGFP4 from d. In some constructs, the human influenza hemagglutinin tag (HA; black squares) was added to the C-terminus of the synthetic kinases to allow detection by immunofluorescence. N-RokDead contains a K116G single amino acid substitution (G) that abolishes catalytic activity. The proteins are aligned vertically with the N-Rok domain. Numbers refer to amino acid positions from N-terminus (N) to C-terminus (C).

N-RokDead-HA) did not recruit Sqh::GFP at the cortex (Fig. 2, a and b).

Of note, expression of N-Rok::vhhGFP4-HA and N-Rok-HA induced cell blebbing and resulted in a high percentage of dead cells. Indeed, only the cells expressing very little active Rok survived, indicating that N-Rok might be toxic in S2 Sqh::GFP cells. This toxicity, presumably resulting from the high expression levels of active Rok with the actin promoter, did not hamper the analysis of single cells (Fig. 2, a and b), but impeded any valuable biochemical analyses. In a fraction of surviving cells expressing N-Rok::vhhGFP4-HA, we detected an enrichment of the anti-Sqh2P signal at the cell cortex using antibody staining, consistent with the higher Sqh::GFP signal at these sites (Fig. 2 c). Control cells displayed granular Sqh2P signal without enrichment at the cell cortex. However, a more direct biochemical and mechanical readout of myosin II activation is of importance to validate more accurately the relevance of our synthetic kinases approach.

To test the engineered kinases in a developing animal and to overcome the limitations of cell culture system, we generated transgenic *Drosophila* lines using the plasmid constructs mentioned above, such that the expression of synthetic Rok kinase fusion proteins can be controlled with the UAS-Gal4 system (Brand and Perrimon. 1993). To achieve the same level of expression of the different constructs, we used phiC31-mediated integration (Bischof et al., 2007) at a unique site located in an intergenic region on the third chromosome (ZH-86Fb). The first genomic insertions we generated were *N-Rok::vhhGFP4*$^{ZH-86Fb}$ and *N-RokDead::vhhGFP4*$^{ZH-86Fb}$ (Fig. 1 d).

### N-Rok::vhhGFP4$^{ZH-86Fb}$ efficiently phosphorylates Sqh::GFP in vivo

To assess the function of the synthetic Rok kinases in flies, we first investigated whether N-Rok::vhhGFP4$^{ZH-86Fb}$ can regulate Sqh::GFP activity and whether it can phosphorylate it on serine 21 (Ser19 in vertebrates), the primary activating residue in *Drosophila* (Fig. 1 b; Jordan and Karess 1997). We performed both fluorescence and immunofluorescence analyses on *Drosophila* embryos carrying either *sqh_Sqh::GFP* or *sqh_Sqh::mCherry*, two functional transgenes (Jordan and Karess 1997; Martin et al. 2009; Royou et al., 2004). We used *engrailed Gal4 (enGal4)* to drive the expression of either *UAS_N-Rok::vhhGFP4*$^{ZH-86Fb}$ or *UAS_N-RokDead::vhhGFP4*$^{ZH-86Fb}$ in the posterior compartment of each segment of the embryonic epidermis (Tabata et al. 1992).

As previously reported (Young et al., 1993; Kiehart et al., 2000), actomyosin formed a cable-like structure surrounding the dorsal opening during the stages 13–14 (dorsal closure) in *Sqh::GFP* embryos (Fig. 2 d, white arrow). Expression of N-Rok::vhhGFP4$^{ZH-86Fb}$ in *Sqh::GFP* embryos dramatically altered the distribution of Sqh::GFP, leading to the formation of large foci within the *engrailed* expression domain from the late germ band retraction stage onward (stage 12). During the dorsal closure stages, Sqh::GFP clusters were most prominent in epidermal cells surrounding the amnioserosa (Fig. 2 d, yellow arrows on the magnified image on the right panel), disrupting the uniform actomyosin cable structure observed in control *Sqh::GFP* embryos (Fig. 2 d, top panel, white arrow). In contrast, expression of inactive kinase N-RokDead::vhhGFP4$^{ZH-86Fb}$ did not lead to Sqh::GFP clustering (Fig. 2 d) nor did it disrupt the uniform actomyosin cable. Of note, we observed Sqh::GFP signal enhancement in the posterior compartments of these embryos (*enGal4* domain); however, this signal was uniformly spread along cell boundaries, as normally observed for Sqh::GFP, without the formation of foci. We assume that this signal enhancement is due to the binding of vhhGFP4 to GFP, as it was previously reported that vhhGFP4 binding results in a significant increase in the fluorescence signal (Harmansa et al., 2017; Kirchhofer et al., 2010). Supporting this interpretation, we observed a similar signal increase upon the expression of vhhGFP4 binder alone in *Sqh::GFP* embryos (Fig. S1).

In embryos expressing N-Rok::vhhGFP4$^{ZH-86Fb}$, antibody staining revealed that phosphorylated Sqh (p-Sqh) was present in intense spots overlapping the ectopic Sqh::GFP foci (*n* = 27/30 embryos included in the analysis), demonstrating efficient Sqh::GFP phosphorylation in vivo (Fig. 2 d and yellow arrows on the magnified panels on the right). Such p-Sqh-rich structures were never observed in the adjacent control epidermal stripes. Embryos expressing N-RokDead::vhhGFP4$^{ZH-86Fb}$ (*n* = 22) displayed diffuse and weak p-Sqh signal, as observed in control embryos lacking the synthetic kinase (*n* = 31; Fig. 2 d). Importantly, the expression of N-Rok::vhhGFP4$^{ZH-86Fb}$ in the absence of Sqh::GFP (*n* = 13) showed p-Sqh levels similar to controls.

To further confirm that the observed potent effects of N-Rok::vhhGFP4$^{ZH-86Fb}$ on Sqh::GFP accumulation and phosphorylation were indeed due to the protein binder-GFP interaction, we investigated the Sqh::GFP signal and p-Sqh levels in embryos expressing N-RokHAZH-86Fb without the fused nanobody (*n* = 6). In this condition, both the Sqh::GFP and the p-Sqh pattern

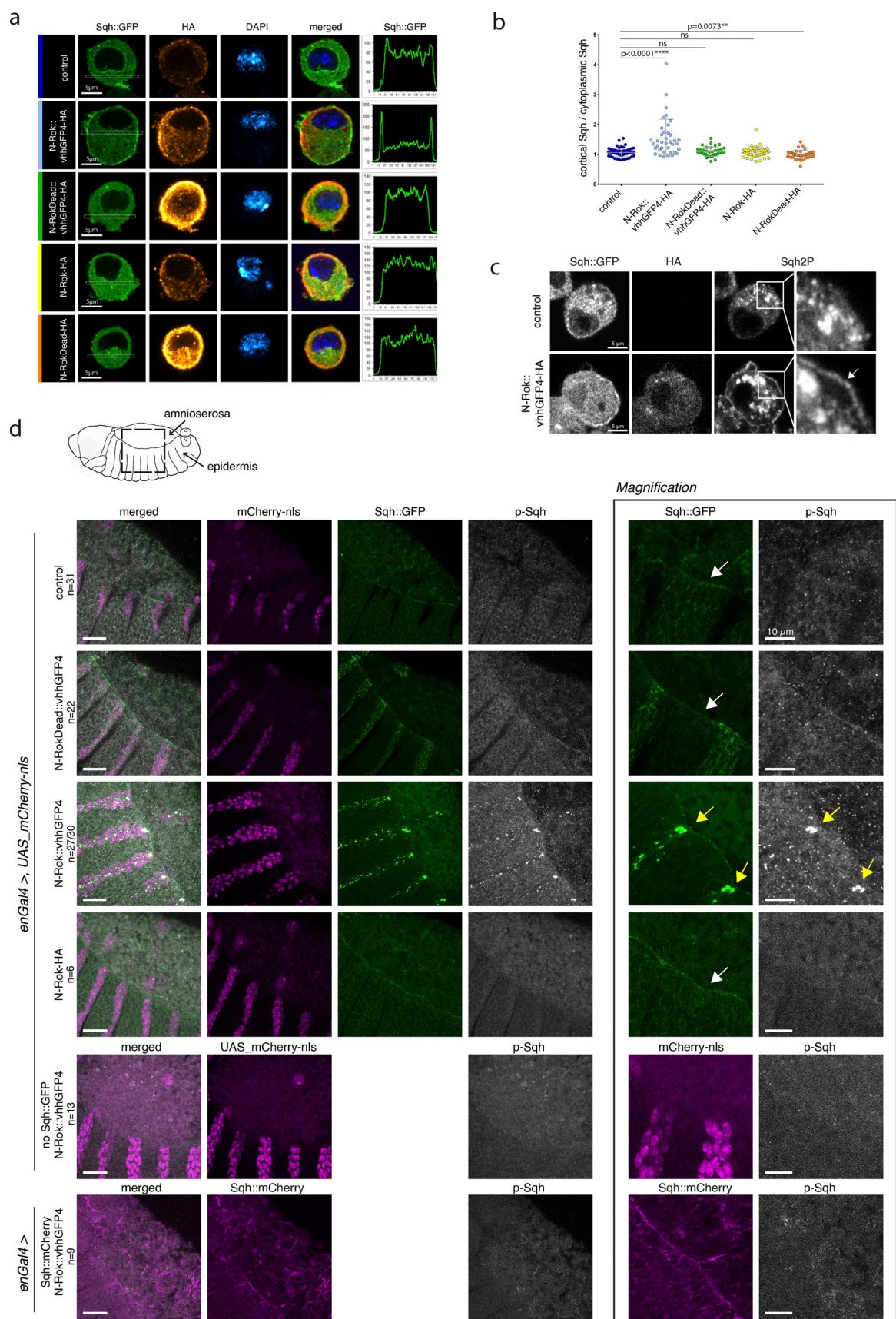

Figure 2. **N-Rok::vhhGFP4 is a functional enzyme and efficiently phosphorylates Sqh::GFP in vivo in a tissue-specific manner. (a)** Expression of N-Rok::vhhGFP4 efficiently recruits Sqh::GFP at the cell cortex in interphasic cells. Stable S2 cell line expressing Sqh::GFP, transfected with N-Rok::vhhGFP4-HA,

N-RokDead::vhhGFP4-HA, N-Rok-HA, or N-RokDead-HA. Sqh::GFP is shown in green, anti-HA staining in red (transfected cells), DAPI in dark blue. Graphs on the right show the ratio of cortical Sqh::GFP/cytoplasmic Sqh::GFP of a representative cell shown for each genotype. Scale bars are 5 µm. **(b)** Quantification of the data shown in a. $n$ = 218 cells, data from three independent experiments. Bars indicate mean ± SD. Asterisks denote statistical significance, derived from unpaired and two-sided Mann–Whitney tests since normality tests showed non-normal distributions: ∗∗∗∗, P ≤ 0.0001; ∗∗, P ≤ 0.01; and n.s., not significant. **(c)** Stable S2 cell line expressing Sqh::GFP transfected with N-Rok::vhhGFP4-HA and stained with anti-Sqh2P antibody. The right panel shows the magnification of the indicated cell region, and the white arrow points to the enrichment of phospho-Sqh signal at the cell cortex. Scale bars are 5 µm. **(d)** Panels show lateral views of fixed *Drosophila* embryos at stage 13–14 (dorsal closure) expressing Sqh::GFP and the synthetic kinase in the *engrailed* domain (visualized by co-expression of mCherry-nls). Embryos were stained with anti-phospho-Sqh/MRLC antibody. The right panel shows magnification of the respective Sqh::GFP and p-Sqh images for each of the embryo genotypes shown on the left. White arrows point to the actomyosin cable around the dorsal hole, yellow arrows point to Sqh::GFP and p-Sqh foci. Please note that the "control" image is the same as in Fig. 4 a to have a single reference image for all N-Rok variants used. For every expressed synthetic kinase, the number of considered embryos is indicated ($n$). For N-Rok::vhhGFP4 the $n$ number is given as a proportion of embryos in which clear p-Sqh clumps corresponding to Sqh::GFP foci were clearly visible to the total number of embryos included in the analysis. Scale bar, 20 µm; in the zoomed panel, 10 µm.

looked similar to the one seen in control embryos (Fig. 2 d, white arrow pointing to the actomyosin cable), indicating that the formation of Sqh::GFP foci and phosphorylation of Sqh strictly depends on the binding of the kinase to the fluorescent fusion protein. Further evidence for this interpretation was obtained via the expression of N-Rok::vhhGFP4$^{ZH-86Fb}$ in *Sqh::mCherry* embryos, which did neither lead to the formation of Sqh::mCherry foci nor to an alteration of the p-Sqh signal ($n$ = 9; Fig. 2 d).

Taken together, our results show that N-Rok::vhhGFP4$^{ZH-86Fb}$ is able to phosphorylate Sqh::GFP on Ser21, the most important residue for myosin II activation (Jordan and Karess. 1997). Moreover, we demonstrate that direct interaction between the protein binder and the GFP-tagged target is required for the activity of the kinase domain on the target, thereby allowing for efficient recruitment and phosphorylation.

### Synthetic Rok kinase modulates mechanical properties of cells through myosin II activity in vivo

Phosphorylation of Sqh/Myosin regulatory light chain leads to myosin II activation and the assembly of bipolar myosin filaments that bind F-actin to generate contractility (Fig. 1 b; reviewed in Vicente-Manzanares et al. [2009]). After confirming that N-Rok::vhhGFP4$^{ZH-86Fb}$ binds to and phosphorylates Sqh::GFP (above), we decided to test whether synthetic Rok was able to specifically modulate mechanical properties of cells through myosin II activity. We focused on dorsal closure, a key morphogenetic process midway through *Drosophila* embryogenesis, where the lateral epidermis from the two sides of the embryo moves up and converges at the dorsal midline to close the epidermal hole, initially covered by amnioserosa cells (Young et al., 1993; Pasakarnis et al., 2016; Fig. 3 a scheme, Fig. 3 a "control" and Video 1). Dorsal closure is a well-characterized model to study epithelial cell sheet movements and fusion, fundamental also in other systems, such as neural tube closure or wound healing in vertebrates. Its strict dependency on actomyosin-driven force generation (Kiehart et al. 2017) makes it an ideal system to test our synthetic Rok kinase.

In a similar paradigm as for assessing p-Sqh levels, we used the *enGal4* driver to express UAS_N-Rok::vhhGFP4$^{ZH-86Fb}$ in a striped pattern in the embryonic epidermis and analyzed its effects on epithelial cell behavior. We performed live imaging in either *Sqh::GFP* or *Sqh::mCherry* embryos, following their development from dorsal closure (stage 13–15) onward.

In contrast to control *Sqh::GFP* embryos, in which dorsal closure occurred normally ($n$ = 12/14; Fig. 3 a and Video 1), the expression of N-Rok::vhhGFP4$^{ZH-86Fb}$ in the *Sqh::GFP* background led to abnormal dorsal closure ($n$ = 13; in most severe cases resulting in a dorsal open phenotype). Closure defects correlated with the formation of Sqh::GFP foci and additional cable structures under tension in cells expressing N-Rok::vhhGFP4$^{ZH-86Fb}$ (Fig. 3 a and Video 1). These structures resulted in a disruption of the actomyosin cable, in local invaginations (presumably due to ectopic pulling forces within the entire tissue), and deformation of the entire epidermis, which prevented normal closure. Contrary to N-Rok::vhhGFP4$^{ZH-86Fb}$, *Sqh::GFP* embryos, in which N-RokDead::vhhGFP4$^{ZH-86Fb}$ was expressed ($n$ = 8), displayed a well-discernible actomyosin cable around the amnioserosa, and dorsal closure proceeded normally (Fig. 3 a and Video 1). As described above for the fixed samples, we observed a uniform Sqh::GFP signal enhancement on cell boundaries in the expression domain of N-RokDead::vhhGFP4$^{ZH-86Fb}$ in live embryos, an effect most likely due to the binding of vhhGFP4 to GFP (Harmansa et al., 2017; Kirchhofer et al., 2010). In the embryos expressing N-Rok-HA$^{ZH-86Fb}$ without the nanobody ($n$ = 8/9), the actomyosin cable was formed as in wild-type embryos, accompanied by normal dorsal closure (Fig. S2 and Video 2); no uniform Sqh::GFP signal enhancement was seen in these conditions. Most of the embryos expressing N-Rok::vhhGFP4$^{ZH-86Fb}$ without Sqh::GFP closed normally ($n$ = 11/14; Fig. 3 a and Video 1). Expression of N-Rok::vhhGFP4$^{ZH-86Fb}$ in *Sqh::mCherry* embryos ($n$ = 4) also led to normal closure (Fig. S2 and Video 3).

The phenotypes described in this and the following section were fully penetrant and occurred both in the presence and the absence of a wild-type *sqh* gene, rendering the sexing of the embryos unnecessary. Interestingly, we noticed during the course of our experiments that the phenotype was more severe when embryos were mounted by gluing them to the coverslip (dorsal open, see Fig. 3 b), while a slightly milder phenotype was seen when the embryos were imaged without gluing, in a glass-bottom dish (aberrant closure, see Fig. 3 a; see Materials and methods section for procedure details). We presume that these differences result from differences in external tension induced by each of the mounting techniques. Only for Fig. 3, b and c and corresponding Video 4 the embryos were imaged with gluing to the coverslip technique; all the other presented embryos were imaged on a glass-bottom dish.

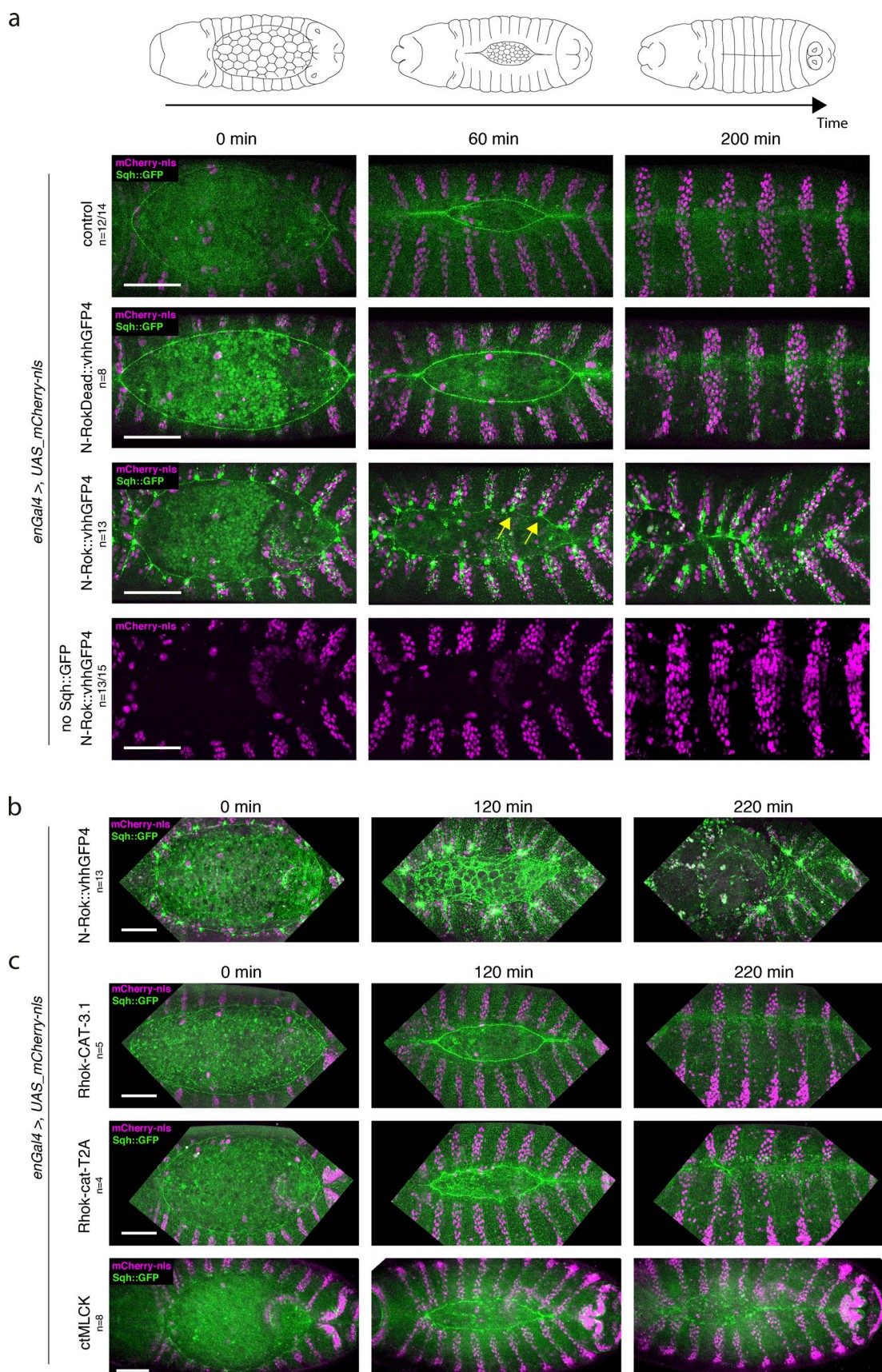

Figure 3. **Synthetic Rok kinase modulates mechanical properties of cells through the phosphorylation of Sqh::GFP and myosin II activity.** **(a)** Schematic illustration of dorsal closure process in the developing fly embryo. Dorsal closure was used as a model to assess myosin II activation by means of

Sqh::GFP phosphorylation with synthetic Rok. In *sqh_Sqh::GFP* embryos, expression of N-Rok::vhhGFP4 leads to abnormal dorsal closure. All panels show stills from live-imaging with dorsal views of the developing *sqh_Sqh::GFP* embryos at stages 13/14–16 (dorsal closure) expressing variants of the synthetic kinase in the *engrailed* domain (visualized by co-expression of mCherry-nls). Note the yellow arrows pointing to the Sqh::GFP foci and actomyosin cable invaginations in N-Rok::vhhGFP4 panel. **(b)** Stills from live imaging with dorsal views of the developing *sqh_Sqh::GFP* embryo expressing N-Rok::vhhGFP4 in the *engrailed* domain mounted with gluing to the coverslip technique to show a more severe dorsal open phenotype than embryos imaged on a glass-bottom dish shown in panel a. **(c)** Dorsal closure was used to compare N-Rok::vhhGFP4 with the previously published effectors that are known to activate myosin II. All panels show stills from live-imaging with dorsal views of the developing *sqh_Sqh::GFP* embryos at stages 13/14–16 (dorsal closure) expressing myosin II activating tool in the *engrailed* domain (visualized by co-expression of mCherry-nls). Only for Fig. 3, b and c and corresponding movies, the embryos were imaged with gluing to the coverslip technique; all the other presented embryos were imaged on a glass-bottom dish. Scale bars: 50 µm. Images are representative of *n* embryos indicated for every expressed synthetic kinase variant.

We conclude from these results that N-Rok::vhhGFP4$^{ZH-86Fb}$ efficiently targets and phosphorylates Sqh::GFP but not Sqh::mCherry or endogenous Sqh, as well as affects mechanical properties of cells through myosin II activity.

## Comparison of engineered Rok with previously available tools for direct activation of myosin II

To compare N-Rok::vhhGFP4$^{ZH-86Fb}$ to the already published effectors that are known to activate myosin II, we used the *en-Gal4* driver and embryonic dorsal closure to benchmark N-Rok::vhhGFP4$^{ZH-86Fb}$ compared to a few other actomyosin activating methods. Two constitutively active forms of Rok (Rok-CAT and Rok-cat) have been generated, and several independent random genomic insertions of *Rok-CAT* (Winter et al., 2001) and *Rok-cat* (Verdier, Guang-Chao-Chen, and Settleman 2006) generated by P element mediated transformation are available. In this work, we only report results obtained with *Rok-CAT3.1* and *Rok-catT2A*, the insertions that yielded the highest levels of expression of these transgenes (Winter et al., 2001; Verdier et al., 2006). In addition, we compared N-Rok::vhhGFP4$^{ZH-86Fb}$ with overexpression of SqhE20E21, a phosphomimetic that emulates diphosphorylated Sqh (Winter et al., 2001), as well as with ctMLCK, a constitutively active form of chicken Myosin light-chain kinase, which is known to phosphorylate Sqh (Kim et al., 2002). Table 1 summarizes the different effectors we tested, the genetic backgrounds in which we performed those tests, and their effect on dorsal closure.

*Sqh::GFP* embryos expressing Rok-CAT (*n* = 5) did not show dorsal open phenotype and were viable (Fig. 3 c and Video 4). In contrast, expression of Rok-cat (*n* = 4) in *Sqh::GFP* embryos was larval lethal and resulted in either a mild stripe misalignment or a more severely aberrant dorsal closure with ectopic clusters of tension (Fig. 3 c and Video 4). Expression of SqhE20E21 under the control of *enGal4* (*n* = 5, Fig. S2 and Video 2) or under the endogenous promoter of *sqh* (Winter et al., 2001) were both viable conditions that produced adult flies (Table 1). Sqh::GFP embryos expressing ctMLCK showed varying effects on dorsal closure: the process was either aberrant (*n* = 5/8) or successfully completed, but at a lower speed than observed normally (*n* = 3/8; Fig. 3 c and Video 4).

During the course of our work, studies from the Sanson lab reported hyperactivation of Myosin II via knockdown of the Myosin II phosphatase *flapwing* (*flw*; Urbano et al., 2018). They used NSlmb::vhhGFP4 (deGradFP [Caussinus et al., 2012]) to achieve efficient protein knockdown of Flw::GFP. However, the expression of NSlmb::vhhGFP4 with the enGal4 driver in *flw*::

GFP embryos (this work) produced adult flies, indicating complete dorsal closure (Table 1) and implying less potent Myosin II activation than achieved with the use of N-Rok::vhhGFP4$^{ZH-86Fb}$.

Overall, N-Rok::vhhGFP4$^{ZH-86Fb}$ turned out to be the most efficient specific myosin II activator among the tools we tested and the assays we used, together with ctMLCK (Kim et al., 2002), which, at least in some embryos, resulted in similarly strong dorsal closure phenotype.

## Evaluation of the general toxicity of N-Rok::vhhGFP4$^{ZH-86Fb}$ in *Drosophila*

Biochemical and cell culture studies reported several potential substrates for Rok/ROCK (reviewed in Amano et al. [2010]). We presumed that binder-dependent interaction of N-Rok::vhhGFP4$^{ZH-86Fb}$ with Sqh::GFP constitutes a more specific tool for directed myosin II targeting than the previously available, constitutively active effectors (Rok-CAT and Rok-cat; Winter et al., 2001; Verdier et al., 2006). However, the specificity of N-Rok::vhhGFP4$^{ZH-86Fb}$ toward a GFP fusion target is unlikely to be absolute, in particular, due to the high expression when using the Gal4 system, and it is important to discriminate between myosin II-dependent effects and the effects of other possible targets on the resulting phenotype.

To evaluate the general toxicity of N-Rok::vhhGFP4$^{ZH-86Fb}$ in *Drosophila* and compare it to the toxicities of Rok-CAT and Rok-cat, we used different Gal4 drivers and expressed them in flies that neither carried a *Sqh::GFP* nor any other GFP fusion target transgene. Doing so, we analyzed the viability and, for viable conditions, the adult fly phenotypes generated by the expression of either N-Rok::vhhGFP4$^{ZH-86Fb}$, N-RokDead::vhhGFP4$^{ZH-86Fb}$, Rok-CAT, or Rok-cat in different tissues and with different levels (Table S1).

Overall, the phenotypes increased in severity according to the potency of the Gal4 drivers used. Importantly, *N-Rok::vhhGFP4$^{ZH-86Fb}$* turned out to induce severe phenotypes in combination with all Gal4 drivers tested (Table S1). These phenotypes were similar to the phenotypes produced by *Rok-catT2A* in combination with the same Gal4 drivers (Verdier et al., 2006 and Table S1). As shown above (Fig. 3 a), embryos expressing N-Rok::vhhGFP4$^{ZH-86Fb}$ in the *en* stripes without a GFP-target closed mostly normally (*n* = 11/14) and did not show increased p-Sqh levels (Fig. 2 d); nevertheless, they were late embryonic/larval lethal. Of note, N-RokDead::vhhGFP4$^{ZH-86Fb}$ expressed without a GFP target did not cause any detrimental phenotype (Table S1), suggesting that the GFP-independent effect of *N-Rok::vhhGFP4$^{ZH-86Fb}$* was due to its active kinase domain.

**Table 1.  Comparison of the direct myosin II effectors available in *Drosophila***

| Effector | Genotype | Dorsal closure phenotype | Viability |
|---|---|---|---|
| Rok-cat[T2A] | sqh_Sqh::GFP, enGal4/+ ; Rok-cat[T2A]/+ | Aberrant | Larval lethal |
| Rok-CAT[3.1] | sqh_Sqh::GFP, enGal4/+ ; Rok-CAT[3.1]/+ | Wild type | Viable |
| N-Rok::vhhGFP4[ZH-86Fb] | sqh_Sqh::GFP, enGal4/+ ; N-Rok::vhhGFP4[ZH-86Fb]/+ | Aberrant/open | Embryonic lethal |
| N-Rok-HA[ZH-86Fb] | sqh_Sqh::GFP, enGal4/+ ; N-Rok-HA[ZH-86Fb]/+ | Wild type | Reduced viability |
| N-Rok::vhhGFP4[Vi] | sqh_Sqh::GFP, enGal4/+ ; N-Rok::vhhGFP4[Vi]/+ | Aberrant | Embryonic lethal |
| Sqh[E20E21] | enGal4/Sqh[E20E21] | Wild type | Viable |
| ctMLCK | sqh_Sqh::GFP, enGal4/+ ; ctMLCK/+ | Aberrant or normal but slower | Larval lethal |
| NSlmb::vhhGFP4 | flw::GFP, enGal4/+ ; NSlmb::vhhGFP4/+ | Wild type | Viable |

Dorsal closure in embryos and adult flies viability were used as a readout of myosin II activation in the *en* expression domain. Both the effectors and the full genotypes of the tested embryos are indicated to highlight all the components that the methods require. For every condition, more than 20 embryos were assessed. See text for details.

We noticed that *Rok-CAT3.1* induced milder phenotypes than *Rok-catT2A*, despite little differences between their sequences (Winter et al. 2001; Verdier et al., 2006). This was also true with regard to dorsal closure, which was mostly aberrant for Rok-catT2A and normal for Rok-CAT3.1 (described above, and Fig. 3 c). Hence, we assumed that the differences in general toxicity between the constitutively active N-Rok::vhhGFP4[ZH-86Fb], *Rok-catT2A,* and *Rok-CAT3.1* are most likely due to the different expression levels from the different genomic insertion sites.

**Generation of a less toxic N-Rok::vhhGFP4[Vi] genomic insertion**
To validate whether the observed differences in the overall toxicity of the tested genomic insertions are due to divergent expression levels of constitutively active Rok, we tried to generate insertions of *N-Rok::vhhGFP4* that would display as little toxicity as possible when combined with Gal4 drivers in the absence of a GFP-fusion target. We generated 25 independent random genomic insertions of *N-Rok::vhhGFP4* by P-element mediated transformation (Spradling and Rubin. 1982) and screened for an insertion that would be viable and fertile in combination with the *enGal4* driver.

Only one insertion (called *N-Rok::vhhGFP4[Vi]*, for "viable") among the 25 lines we generated was viable and fertile in combination with *enGal4*. Since the expression of *enGal4* is turned on in the posterior compartment of each segment of the epidermal structures, including the wings, we analyzed the wings of *N-Rok::vhhGFP4[Vi]* flies. We solely noticed a wing phenotype consisting of partially missing crossvein and a slight but significant reduction of the posterior compartment. Interestingly, the partially missing crossvein phenotype is reminiscent of the phenotype due to the ectopic activation of Rho1 during wing development, which is in line with N-Rok::vhhGFP4[Vi] mildly activating myosin II (Matsuda et al. 2013; Verdier et al., 2006). *N-Rok::vhhGFP4[Vi]* did not show any phenotype when combined with *elavGal4, 69BGal4,* or *daGal4* driver; in contrast, it turned out to be lethal in combination with *tubGal4*, which has a strong ubiquitous expression pattern (Table S1). Taken together, these results suggest that the *N-Rok::vhhGFP4[Vi]* line allows achieving expression levels of N-Rok::vhhGFP4 low enough to

prevent apparent off-target effects in *Drosophila* with most Gal4 drivers.

To test whether *N-Rok::vhhGFP4[Vi]* can be used to express enough N-Rok::vhhGFP4 to act as a useful source of an engineered kinase, we again used the *enGal4* driver and dorsal closure as an assay system to assess the effects on myosin II contractility. Similar to *N-Rok::vhhGFP4[ZH-86Fb]*, expression of N-Rok::vhhGFP4[Vi] in *sqh_Sqh::GFP* embryos resulted in the formation of large Sqh::GFP foci in every other segment of the embryonic epidermis that was most prominent around the dorsal hole (Fig. 4 a). As observed for *N-Rok::vhhGFP4[ZH-86Fb]* (Fig. 2 d), those large Sqh::GFP foci overlapped with p-Sqh clusters (n = 14/15, Fig. 4 a). In the same paradigm as for *N-Rok::vhhGFP4[ZH-86Fb]*, we performed live imaging in *Sqh::GFP* embryos expressing N-Rok::vhhGFP4[Vi] with *enGal4*. This condition led to abnormal dorsal closure (n = 10) and was lethal, similar to *N-Rok::vhhGFP4[ZH-86Fb]* (Fig. 4 b and Video 5).

We conclude from these results that *N-Rok::vhhGFP4[Vi]* allows expressing synthetic Rok kinase at a concentration both high enough to phosphorylate a GFP-fusion version of Sqh and low enough to prevent deleterious effects in flies deprived of a GFP target.

**Generation of a destabilized GFP-binder kinase**
During the course of our studies, the group of Cepko reported the isolation and characterization of a destabilized GFP-binding nanobody (dGBP1; Tang et al., 2016). dGBP1 fusion proteins are only stable in the presence of a GFP (or its close derivates), both in vitro (in cultured cells) and in vivo; in the absence of a GFP-fusion protein, dGBP1 is destabilized and the entire dGBP1–fusion protein is degraded by the ubiquitin–proteasome system, with no signs of aggregation in cells (Tang et al., 2016; Fig. 4 c).

To further minimize the potential off-target activity of our synthetic N-Rok kinase in the absence of a GFP-target, we generated flies carrying *N-Rok::dGBP1* (Fig. 1 e) in the same intergenic region on the third chromosome (ZH-86Fb) that was used to generate *N-Rok::vhhGFP4[ZH-86Fb]* flies. When we expressed N-Rok::dGBP1[ZH-86Fb] within the *en* domain in *Sqh::GFP*

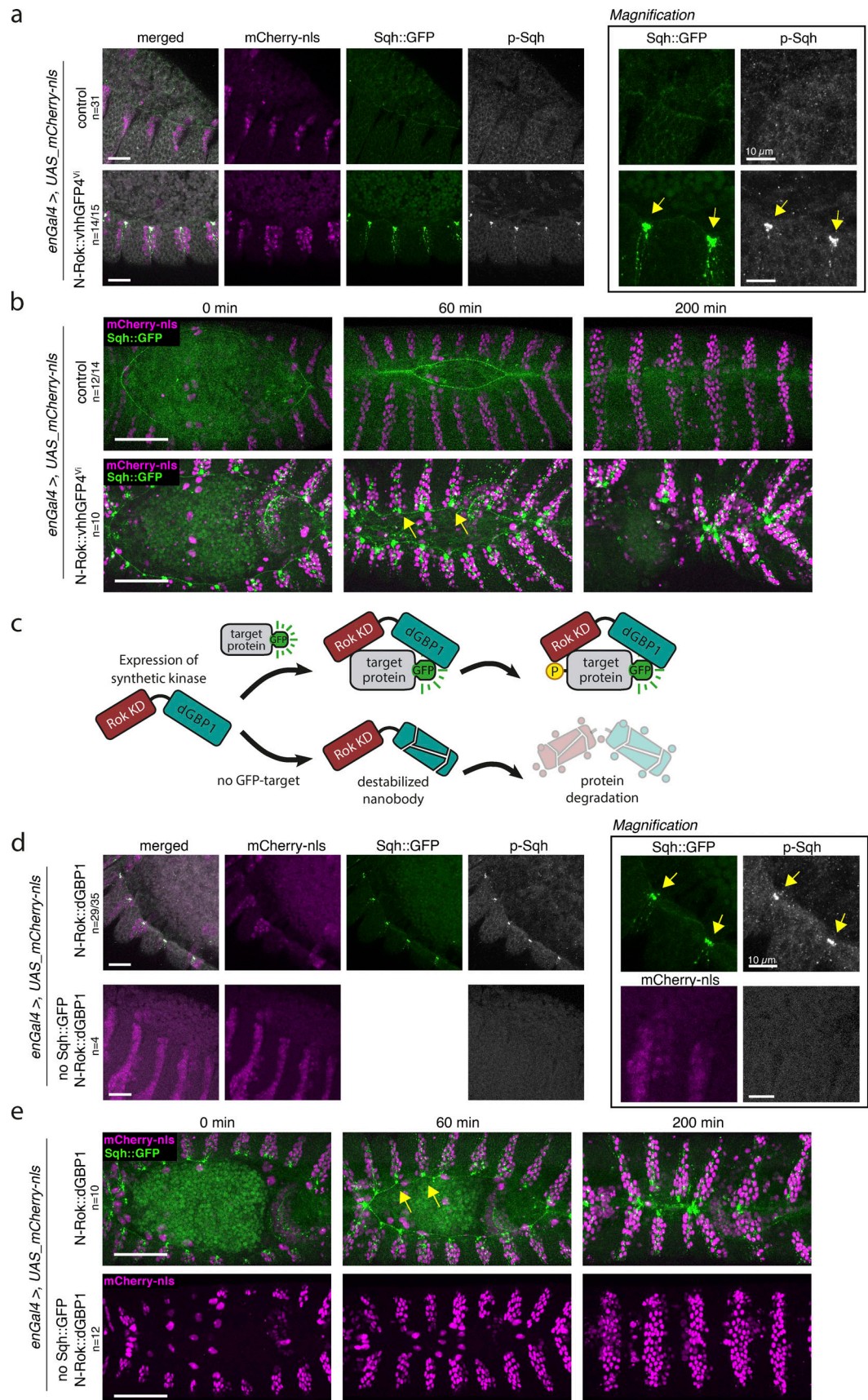

Figure 4. **N-Rok::vhhGFP4$^{Vi}$ and N-Rok::dGBP1 are optimized variants of synthetic N-Rok. (a)** Panels show lateral views of fixed *sqh_Sqh::GFP* embryos at stage 13–14 (dorsal closure) expressing N-Rok::vhhGFP4$^{Vi}$ in the *engrailed* domain (visualized by co-expression of mCherry-nls). Embryos were stained with

anti-phospho-Sqh/MRLC antibody. The panel on the right shows the magnification of the respective Sqh::GFP and p-Sqh images for each of the presented embryos. White arrows point to the actomyosin cable around the dorsal hole, yellow arrows point to Sqh::GFP and p-Sqh foci. **(b)** Dorsal closure was used to compare the functionality of N-Rok::vhhGFP4$^{Vi}$ to N-Rok::vhhGFP4$^{ZH-86Fb}$ (Fig. 3 a). Panels show stills from live-imaging with dorsal views of the developing *sqh_Sqh::GFP* embryos at stages 13/14–16 (dorsal closure) expressing N-Rok::vhhGFP4$^{Vi}$ in the *engrailed* domain (visualized by co-expression of mCherry-nls). Please note that both in a and b the "control" images are the same as on Figs. 2 d and 3 a, respectively, to have a single reference image for all N-Rok variants used. **(c)** Schematic illustration of the synthetic kinase with destabilized GFP nanobody (dGBP1) work concept. In a similar way as in Fig. 1 a, a synthetic kinase uses GFP to bring a constitutively active kinase domain (Rok KD) in close proximity to a fluorescent fusion substrate protein (target). The persistence of the kinase domain around the fluorescent fusion protein achieves efficient phosphorylation (P) of the target. In the absence of the GFP-target, dGBP1 nanobody is destabilized, and the whole nanobody-kinase fusion protein is targeted for degradation. **(d)** Panels show lateral views of stage 13–14 (dorsal closure) fixed *sqh_Sqh::GFP* or control *yw* embryos expressing N-Rok::dGBP1$^{ZH-86Fb}$ in the *engrailed* domain (visualized by co-expression of mCherry-nls). Embryos were stained with an anti-phospho-Sqh/MRLC antibody. The panel on the right shows the magnification of the respective Sqh::GFP and p-Sqh images. Yellow arrows point to Sqh::GFP and p-Sqh foci. **(e)** Dorsal closure was used to compare the functionality of N-Rok::dGBP1$^{ZH-86Fb}$ to N-Rok::vhhGFP4$^{ZH-86Fb}$ (Fig. 3 a) and N-Rok::vhhGFP4$^{Vi}$ (b). Panels show stills from live-imaging with dorsal views of the developing *sqh_Sqh::GFP* or control yw embryos expressing N-Rok::dGBP1 in the *engrailed* domain (visualized by co-expression of mCherry-nls). For every embryo genotype, the number of analyzed embryos is indicated (*n*). For N-Rok::dGBP1$^{ZH-86Fb}$ the *n* number is given as a proportion of embryos in which clear p-Sqh clumps corresponding to Sqh::GFP foci were visible to the total number of embryos included in the analysis. Scale bar, 50 μm in b and e; 20 μm in a and d; in the zoomed panel, 10 μm.

embryos, we observed spots of high p-Sqh levels overlapping with large Sqh::GFP foci in every other segment (Fig. 4 d). Despite large Sqh::GFP clusters, p-Sqh foci were less prominent than those observed for N-Rok::vhhGFP4$^{Vi}$ (Fig. 4 a) or N-Rok::vhhGFP4$^{ZH-86Fb}$ (Fig. 2 d); however, they were still clearly distinguishable in most embryos (*n* = 29/35). In contrast, the embryos expressing N-Rok::dGBP1$^{ZH-86Fb}$ alone displayed a dim, uniform p-Sqh signal (*n* = 4, Fig. 4 d).

Along the same lines as for N-Rok::vhhGFP4$^{Vi}$ (Fig. 4 b) and N-Rok::vhhGFP4$^{ZH-86Fb}$ (Fig. 3 a), we performed live imaging of dorsal closure in *sqh_Sqh::GFP* embryos carrying *N-Rok::dGBP1$^{ZH-86Fb}$* expressed in the *en* domain (*n* = 10). As expected, we observed the formation of ectopic cable structures that disrupted the uniform actomyosin cable string (Fig. 4 e and Video 5), leading to the abnormal closure and embryonic/early larvae lethality. Most importantly, embryos expressing only N-Rok::dGBP1$^{ZH-86Fb}$ in the *en* domain without the GFP target exhibited normal closure and did not show any obvious developmental defects and proceeded to adulthood (*n* = 12, Fig. 4 e and Video 5).

Our data show that N-Rok::dGBP1$^{ZH-86Fb}$ phosphorylates Sqh::GFP and efficiently modulates mechanical properties of cells through myosin II, similar to N-Rok::vhhGFP4$^{Vi}$ and N-Rok::vhhGFP4$^{ZH-86Fb}$. More importantly, expression of N-Rok::dGBP1$^{ZH-86Fb}$ did not show any obvious deleterious effects in flies lacking a GFP target (Fig. 4 e). We presume that this is due to the degradation of the N-Rok::dGBP1 fusion protein in the absence of a dGBP1 cognate antigen, thereby restricting the kinase activity to the time and place where a GFP target is present.

### Nanobody-based engineered kinases are modular tools

Nanobody-based engineered kinases would display an extra degree of modularity if one could target the kinase domain to other fluorescent proteins than GFP and its close derivatives. To that aim, we first generated two novel genomic insertions in ZH-86Fb: *N-Rok::2m22$^{ZH-86Fb}$* and *N-RokDead::2m22$^{ZH-86Fb}$*. *N-Rok::2m22$^{ZH-86Fb}$* and *N-RokDead::2m22$^{ZH-86Fb}$* are similar to *N-Rok::vhhGFP4$^{ZH-86Fb}$* and *N-RokDead::vhhGFP4$^{ZH-86Fb}$*, respectively, but a DARPin specifically recognizing mCherry (Brauchle et al., 2014) replaces the nanobody binding to GFP (Fig. 1 f).

Similar to our assays for N-Rok::vhhGFP4$^{ZH-86Fb}$ in *Sqh::GFP* embryos, we tested whether N-Rok::2m22$^{ZH-86Fb}$ can specifically modulate myosin II activity through the phosphorylation of Sqh::mCherry. As expected, the expression of N-Rok::2m22$^{ZH-86Fb}$ in *Sqh::mCherry* embryos with the enGal4 driver led to the formation of large Sqh::mCherry foci in every other segment of the embryonic epidermis, with a prominent p-Sqh signal corresponding to the foci (*n* = 8, Fig. S3). In contrast, *Sqh::GFP* embryos did not show any clusters in the *en* domain upon N-Rok::2m22$^{ZH-86Fb}$ expression and exhibited diffuse and wild-type-like p-Sqh staining (*n* = 8, Fig. S3).

We next performed live imaging experiments in *Sqh::mCherry* embryos. Again, we used enGal4 to drive the expression of N-Rok::2m22$^{ZH-86Fb}$ and analyzed dorsal closure. In contrast to the control *Sqh::mCherry* embryos, which closed normally (*n* = 8, Fig. S2 and Video 3), the expression of N-Rok::2m22$^{ZH-86Fb}$ in the presence of Sqh::mCherry led to abnormal closure (*n* = 4) as well as to the formation of cable structures under tension (Fig. S2 and Video 3) and was larval lethal. Importantly, in Sqh::mCherry embryos expressing *N-RokDead::vhhGFP4$^{ZH-86Fb}$* dorsal closure proceeded normally (*n* = 5, Fig. S2 and Video 3).

We conclude from these results that N-Rok::2m22$^{ZH-86Fb}$ can efficiently and specifically phosphorylate Sqh::mCherry and modulate myosin II activity in a way similar to N-Rok::vhhGFP4 in *Sqh::GFP* embryos.

### Evaluation of the specificity of engineered Rok kinase

We have shown that synthetic Rok kinase activates myosin II through the phosphorylation of Sqh::GFP, thereby influencing tissue mechanics in vivo. We next sought to test whether N-Rok::vhhGFP4 can phosphorylate and/or activate other GFP-fusion proteins that are not a direct target of Rok in cells.

We first used a *Tkv::YFP* line and expressed N-Rok::vhhGFP4$^{Vi}$ in the *en* compartment of the embryos. Thickveins (Tkv) is a receptor of the BMP pathway (Nellen et al., 1994). Upon ligand binding, Tkv is activated by the phosphorylation of specific cytoplasmic S/T residues by its co-receptor Punt, eventually resulting in Mad phosphorylation (Greaves et al., 1999; Kretzschmar et al., 1997; Macías-Silva et al., 1996; Raftery and Sutherland 1999; Souchelnytskyi et al., 1997). We, therefore, reasoned that Tkv could potentially be a substrate for our engineered N-Rok::vhhGFP4 (S/T kinase) and used p-Mad activation levels as a readout. Expression of N-Rok::vhhGFP4$^{Vi}$ in *Tkv::*

*YFP* embryos had no effect either on pMad staining levels (Fig. S4 a) or on dorsal closure. Moreover, by using another Tkv line, we verified that the expression of Tkv::GFP and N-Rok::vhhGFP4 in the *engrailed* domain did not influence p-Sqh levels (*n* = 5; Fig. S4 b).

Rok/ROCK has been shown to interact with the cadherin complex through direct binding to p120 catenin and to influence the stability of adherens junctions (Smith et al., 2012). E-Cadherin (DE-Cadherin; DE-Cad in *Drosophila*) is phosphorylated on a few serine residues, and this phosphorylation is required for binding to β-Catenin and the stability of this complex (Huber and Weis. 2001; Lickert et al., 2000). We, therefore, decided to use *DE-Cad::GFP* (Huang et al., 2009) embryos to ask if our synthetic Rok kinase could act on DE-Cad::GFP and cause any obvious developmental defects. We used the enGal4 driver to express N-Rok::vhhGFP4[Vi] and focused on dorsal closure and viability. We did not observe any detrimental effect (*n* = 5/6 embryos) in the presence of the activated kinase, and this condition was viable (*n* = 5/6 embryos; Fig. S4 c and Video 6), similar to *yw* control flies crossed with *DE-Cad::GFP* (8/8 embryos). Of note, we observed an enhanced GFP signal in the *en* domain, suggesting N-Rok::vhhGFP4[Vi] binding to GFP (see Fig. 2 d and Video 6 and Harmansa et al., 2017; Kirchhofer et al., 2010).

We assumed that since Rok kinase normally exerts its function in the vicinity of cell membranes (Truebestein et al., 2015), expression of N-Rok::vhhGFP4[Vi] in the presence of GFP-tagged DE-Cad may increase the chance of unspecific phosphorylation of the untagged Sqh protein present in the cell cortex. The increased GFP signal in the *en* domain observed in the above experiment implies that N-Rok::vhhGFP4[Vi] binds DE-Cad::GFP, suggesting it is enriched in the close proximity of the cell membrane. We, therefore, sought to examine if endogenous Sqh becomes a target when the engineered Rok kinase is brought in close proximity through neighboring GFP–binder interactions. We examined the levels of p-Sqh in *DE-Cad::GFP* embryos expressing N-Rok::vhhGFP4[Vi] with the enGal4 driver. We did not observe a p-Sqh signal increase in this setup (in contrast to p-Sqh foci observed for N-Rok::vhhGFP4[Vi] in the presence of Sqh::GFP; *n* = 6, Fig. S4 d). These results suggest that GFP-fusion kinases have to be brought into direct contact with their target proteins to affect their phosphorylation; simply being enriched in the same cellular region does not appear to be necessarily sufficient to direct their activity efficiently toward a given target protein.

### The tracheal system of *Drosophila* embryo develops abnormally upon excessive myosin II phosphorylation by engineered Rok kinase

We next used the engineered Rok kinase to study the effect of actomyosin activation on cell migration and cell rearrangements during branch formation in the tracheal system, the fly's respiratory organ. In the *Drosophila* embryo, the tracheal system develops from 10 bilaterally symmetrical clusters of ectodermal cells. Cells in each cluster invaginate and migrate to form one tracheal segment that further connects with neighboring segments to form the stereotypical network of tubes. Remarkably,

cells do not divide during this process, but arrange into a complex branched structure via cell migration, cell shape changes, and cell–cell neighbor exchange. During the formation of the dorsal branch (DB), individual DBs initially elongate as a cluster of cells, led by migrating distal-most cells at the front (tip cells), which induces tension on stalk cells trailing in the back, ultimately resulting in stalk cell intercalation (SCI) and elongation of stalk cells (Ribeiro et al., 2004; Caussinus et al., 2008). SCI occurs in all primary branches except for the ones that will eventually form the dorsal trunk, the largest tube in the tracheal system (Affolter and Caussinus. 2008; Ribeiro et al., 2004; Ochoa-Espinosa and Affolter. 2012). In sharp contrast to the actomyosin-driven junctional rearrangements observed in many intercalation events in *Drosophila* and vertebrate morphogenesis (Martin et al., 2009; Rauzi et al., 2008; Rauzi et al. 2010; Lecuit and Lenne 2007; Bertet et al. 2004; Fernandez-Gonzalez et al., 2009; Lye et al., 2015; Blankenship et al., 2006; Monier et al., 2010; Skoglund et al., 2008), tracheal cell intercalation occurs passively and does not require actomyosin contractility (Ochoa-Espinosa et al., 2017). DB cell intercalation occurs in the absence of myosin II function, and it is possible that stalk cells need to downregulate actomyosin activity to give in to the tension created by the migration of the tip cells (Ochoa-Espinosa et al., 2017).

To test whether the activation of myosin II in tracheal cells would lead to a failure in the release of tension and consequently to the inhibition of proper trachea development, we expressed the different variants of synthetic N-Rok using the pan-tracheal *breathless-Gal4 (btlGal4)* driver in the *sqh_Sqh::GFP* background. We first tested whether engineered kinases phosphorylate Sqh::GFP specifically in the developing trachea. Expression of all three active N-Rok variants (N-Rok::vhhGFP4[ZH-86Fb], N-Rok::vhhGFP4[Vi], and N-Rok::dGBP1[ZH-86Fb]) resulted in the formation of large Sqh::GFP foci in tracheal cells that represented highly phosphorylated Sqh, as confirmed by anti-p-Sqh antibody staining (*n* = 7/8, *n* = 7, *n* = 4/6, respectively, Fig. 5 a). In most cases, we observed clustering of groups of tracheal cells and a discontinued tracheal network, as shown for N-Rok::vhhGFP4[ZH-86Fb] and N-Rok::dGBP1[ZH-86Fb], which was visualized by mCherry-nls marking tracheal cell nuclei (yellow arrows in Fig. 5 a). We presume that these clusters arise from the accumulation of tracheal cells around the Sqh::GFP foci as a result of increased tension, "holding" the clusters of cells together and preventing them from proper migration and intercalation. In the most severe cases, almost no cells escaped from the "rosettes" around the Sqh::GFP foci, and the whole cluster of cells migrated together, without showing any signs of branch formation (see N-Rok::vhhGFP4[Vi] in Fig. 5 a). We noted a variety of phenotypes ranging from severe (several cells escaping from the rosettes and forming branches) to very severe (cells forming compacted rosettes around the Sqh::GFP clusters). Nevertheless, in all cases, we observed embryonic/early larval lethality. The images in Fig. 5 a were chosen to represent the strongest observed phenotype captured at a given stage (14–16) and in an adequate positioning of the embryo in the specimen. In contrast, embryos expressing inactive N-RokDead::vhhGFP4[ZH-86Fb] variant did not show clusters of Sqh::GFP nor p-Sqh (*n* = 6); tracheal cells

Figure 5. **Tracheal system of *Drosophila* embryo develops abnormally due to the excessive myosin II phosphorylation by engineered N-Rok kinase.** **(a)** All panels show lateral views of fixed *sqh_Sqh::GFP* embryos at stages 14–16 expressing the indicated variant of synthetic Rok kinase in the tracheal system

(*btl* expression domain visualized by co-expression of mCherry-nls). Embryos were stained with an anti-phospho-Sqh/MRLC antibody. For every expressed synthetic kinase, the number of embryos analyzed is indicated (*n*). For N-Rok::dGBP1, the *n* number is given as a proportion of embryos in which clear p-Sqh clumps corresponding to Sqh::GFP foci were visible, to the total number of embryos included in the analysis. Scale bar, 20 µm. **(b)** Aberrant tracheal development in *sqh_Sqh::GFP* embryos expressing the indicated variant of synthetic Rok kinase in the tracheal system. All panels show stills from live imaging with lateral views of *sqh_Sqh::GFP* embryos at stages 11/12, 14, and 17, corresponding to the stages depicted on the schematic drawing above. The panel on the right shows magnification of the indicated region from the last stage. Yellow arrows point to Sqh::GFP foci visible already at stage 11/12; asterisks indicate clusters of tracheal cells around the Sqh::GFP foci. Scale bars, 50 µm; in the zoomed panel, 20 µm.

migrated out and formed stereotypic branches, similar to what is seen in wild-type embryos (Fig. 5 a), and this condition was viable.

To better understand the timing and cell behaviors in the context of constitutive activation of myosin II by N-Rok in tracheal cells, we next performed live imaging in *Sqh::GFP* embryos expressing engineered Rok variants in the *btl* expression domain. Importantly, for all active N-Rok variants, we noticed Sqh::GFP clusters at the invaginating tracheal placodes already around stage 11 (yellow arrows in Fig. 5 b and Video 7). Sqh::GFP clusters were most prominent for N-Rok::vhhGFP4[ZH-86Fb] and for N-Rok::vhhGFP4[Vi] (*n* = 9 for both), and somewhat smaller for N-Rok::dGBP1[ZH-86Fb] (*n* = 16). As a result, tracheal cells were unable to migrate away from the placode, but rather formed rosettes around the Sqh::GFP foci, leading to a dramatically disorganized and fragmented tracheal architecture (Fig. 5 b, asterisks and Video 7). Again, as described above for the fixed samples, we noticed a range of phenotypes that differed in how many cells escaped the Sqh::GFP clusters and migrated out forming aberrant, yet distinct dorsal and ventral branches, or whether entire rosettes migrated together. In contrast, most *Sqh::GFP* embryos expressing the inactive kinase N-RokDead::vhhGFP4[ZH-86Fb], as well as simultaneously imaged control embryos, developed a morphologically normal tracheal network (*n* = 8/9 and *n* = 7/9, respectively, Fig. 5 b and Video 7), and this condition was viable.

We conclude from these experiments that tracheal-specific phosphorylation of Sqh::GFP by synthetic N-Rok variants strongly perturbs cell migration from early stages of trachea development onward, resulting in dramatic phenotypes, disrupted primary branch formation, and aberrant trachea development leading to embryonic/early larvae lethality. These results suggest that finely tuned levels of active myosin II are crucial for proper trachea morphogenesis and support the utility of our tool for resolving biological questions requiring ectopic actomyosin activation.

### Effect of actomyosin activation on cell intercalation and dorsal branch elongation

Since the use of a pan-tracheal driver (btlGal4) to express active N-Rok kinase resulted in robust Sqh::GFP phosphorylation and cell clustering already at the early stages of tracheal development (see above), this scenario prevented us from studying cell intercalation and adherens junction (AJs) remodeling in developing branches. To take a closer look at these processes in the context of excessive actomyosin activity, we switched to a knirps (kniGal4) driver line, which activates expression in dorsal and ventral branches, but not in the dorsal trunk (Jaspers et al., 2014).

At the onset of branch elongation (stage 13/14), stalk cells in dorsal branches are in a side-by-side arrangement; as the branch

elongation progresses and cells rearrange, intercellular AJs of adjacent cells are progressively converted into autocellular AJs as the stalk cells reach around the lumen and seal the lumen tube with these newly formed AJs (Fig. 6 a; Ribeiro et al., 2004; Caussinus et al., 2008). By stage 16, when SCI is completed, and one of the tip cells meets with the contralateral branch (fusion cell) and the other one becomes a terminal cell with long cellular extensions, stalk cells are in an end-to-end arrangement and show a characteristic pattern of lines and small rings of AJs, corresponding to auto- and intercellular AJs, respectively (Steneberg et al., 1999).

To test the hypothesis that actomyosin activity might need to be low in stalk cells so as not to produce forces counteracting tip cell migration and DB cell intercalation, we used the engineered Rok fused with dGBP1 to interrogate the role of actomyosin during branch elongation and the stereotypic junction formation in developing dorsal branches.

We first examined the influence of actomyosin activation in dorsal branches on filopodia formation and cell migration in time-lapse imaging using a LifeAct-mRuby reporter. Expression of the N-Rok::dGBP1[ZH-86Fb] in the *kni* expression domain in *Sqh::GFP* embryos resulted in the formation of Sqh::GFP clusters from stage 13/14 onward in the areas around cell junctions in the stalk cells and in between the two tip cells (Fig. 6 b). Such Sqh::GFP foci were never observed in the control embryos in the absence of N-Rok::dGBP1[ZH-86Fb] expression. Similar to the control embryos, Sqh::GFP embryos expressing N-Rok::dGBP1[ZH-86Fb] in the *kni* domain formed filopodia and showed high filopodia activity in the tip cells. In both classes of embryos, in most branches, stalks cells trailed after the filopodia-rich tip cells, leading to the elongation of the dorsal branches ($n_{branches}$ = 102/104 for dGBP1, from $n_{embryos}$ = 35 and $n_{branches}$ = 46/46 for control, from $n_{embryos}$ = 15, on average 3 central branches per embryo were analyzed; Fig. 6 b and Video 8). Therefore, the expression of N-Rok::dGBP1[ZH-86Fb] with *kniGal4* driver did neither preclude filopodia formation nor cell migration of tip cells.

However, many branches expressing N-Rok::dGBP1[ZH-86Fb] displayed several defects, most commonly: (1) some tip cells showed aberrant behavior in keeping persistent contact with adjacent branches ($n_{branches}$ = 16/104, arrow, Fig. 6 b and Video 8); (2) in some cases, tip cells detached from stalk cells ($n_{branches}$ = 10/104; arrowheads, Fig. 6 b and Video 8); and (3) in a few cases, after initial branch elongation, we observed clustering of tip and stalk cells accompanied by branch shortening ($n_{branches}$ = 5/104). As a result, some branches did not establish the proper 1:1 contact with the opposite partner branch from the other side of the embryo. Instead, DBs connected only with a neighbor branch or clustered with the adjacent branch and contacted together the contralateral partner branch, or in the case of branch shortening, no contact

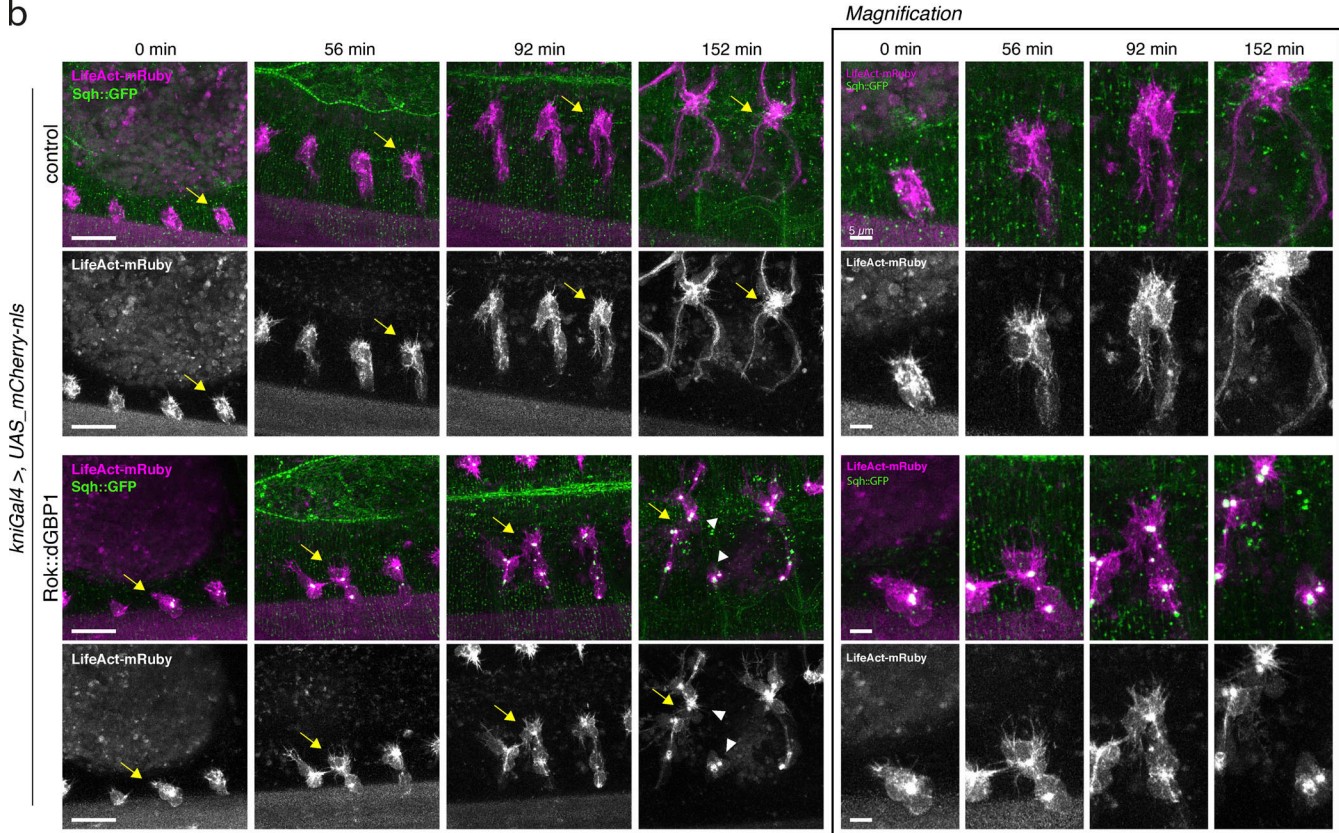

Figure 6. **Effect of constitutive activation of actomyosin with N-Rok::dGBP1 in dorsal branches on filopodia formation and cell migration.** **(a)** Schematic illustration of dorsal branch formation, showing cell junction rearrangements during cell intercalation. Cell intercalation is brought about by the migration of tip cells towards the fibroblast growth factor (FGF) source. **(b)** All panels show stills from live imaging with dorsolateral views of dorsal branches of *sqh_Sqh::GFP kniGal4* control embryos and *sqh_Sqh::GFP kniGal4 Rok::dGBP1^{ZH-86Fb}* embryos; LifeAct-mRuby (magenta) was used to visualize pools of F-actin. Arrows indicate the same dorsal branch for all time points; arrowheads point to the tip cell detached from the stalk cells. Scale bars, 20 or 5 µm on the magnification panel.

with the contralateral partner was established. In the control embryos, most of the branches analyzed migrated as expected ($n_{branches}$ = 44/46) and, after reaching the midline, contacted their contra-lateral partners in a 1:1 ratio (Fig. 6 b, last timepoint and Video 8).

We next performed anti-DE-Cad staining to investigate cell intercalation and autocellular junction formation in stage 16 embryos. Before intercalation, dorsal branches expressing engineered Rok kinase displayed prominent Sqh::GFP foci in the cluster of cells giving rise to the emerging branch (Fig. 7 a, stage 14, arrowheads). As the branches elongated, the foci moved together with the intercalating stalk cells, eventually persisting predominantly at the sites of the intercellular junctions (Fig. 7 a, stage 15–16, arrowheads). At stage 16, *Sqh::GFP* embryos expressing engineered N-Rok kinase showed a characteristic pattern of AJs similar to the control embryos, indicating correct cell intercalation despite the presence of ectopically activated actomyosin foci ($n_{branches}$ = 23/26 from $n_{embryos}$ = 13 for N-Rok::

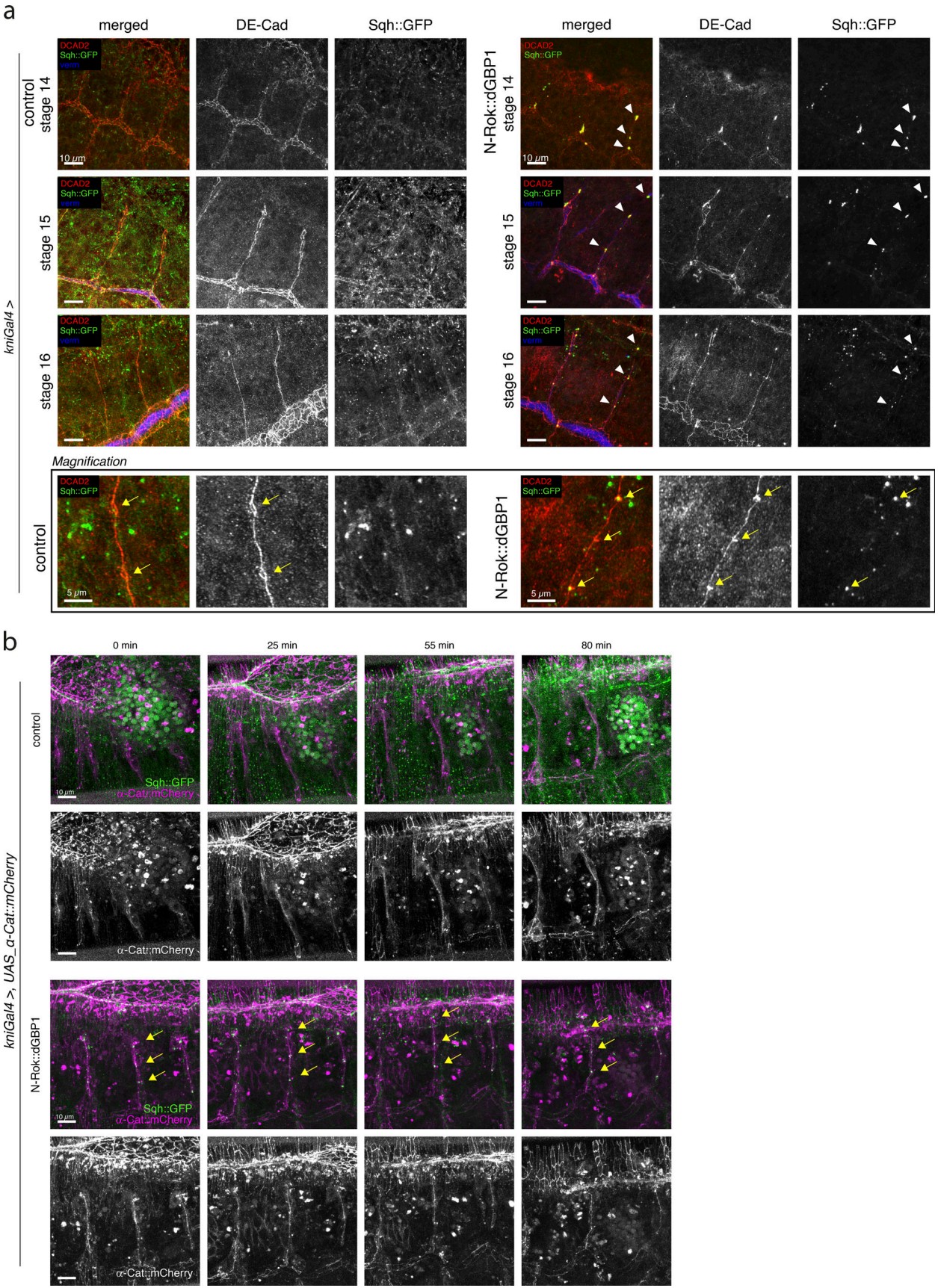

Figure 7. **Effect of constitutive activation of actomyosin with N-Rok::dGBP1 in the *kni* domain on cell intercalation. (a)** Confocal projections showing lateral views of fixed *sqh_Sqh::GFP kniGal4* control and *sqh_Sqh::GFP kniGal4 Rok::dGBP1* embryos stained for DE-Cad (red) and vermiform (blue). Note the

presence of ectopic Sqh::GFP foci in the embryos expressing the engineered kinase (arrowheads). The bottom panel shows the magnification of a single branch from the respective image of stage 16 embryos to show the characteristic pattern of lines and small rings of AJs, corresponding to auto- and intercellular AJs, respectively. Yellow arrows point to the rings corresponding to intercellular AJs. **(b)** Stills from time-lapse movies showing projections of dorsal branches of *sqh_Sqh::GFP kniGal4* control embryos and *sqh_Sqh::GFP kniGal4 Rok::dGBP1* embryos; α-Cat::mCherry (magenta) marks cell junctions. In the course of the experiments with UAS α-Cat::mCherry stock, we noticed background expression of α-Cat::mCherry in the epidermis, regardless of the driver used, and even in the absence of a driver. However, we verified a robust signal increase with the use of *enGal4* driver. Therefore, the observed background of α-Cat::mCherry expression, although limiting a clearer view of the imaged embryos (especially at the beginning of branch emerging), did not prevent the use of this line for studying intercalation in dorsal branches via live imaging. Scale bars, 10 or 5 μm on the magnification panel.

---

dGBP1$^{ZH-86Fb}$ and $n_{branches}$ = 14/14 from $n_{embryos}$ = 6 for the control; on average two branches per embryo were analyzed, Fig. 7 a, stage 16, and the magnification panel). Co-staining of the embryos with Vermiform, a marker for the tracheal lumen, revealed that in the majority of the imaged embryos, the lumen was formed both in the dorsal trunk (where no synthetic kinase was expressed), as well as in the kinase-expressing branches.

To follow cell intercalation in the context of excessive acto-myosin activation, we performed live-imaging in *Sqh::GFP* embryos expressing fluorescent α-Catenin (α-Cat::mCherry) to visualize cell junctions (Video 9). We used again the *kni* driver to express N-Rok::dGBP1$^{ZH-86Fb}$. In line with the live-imaging experiments using LifeAct-mRuby (see above), we observed Sqh::GFP clusters from stage 13/14 onward upon expression of engineered kinase N-Rok::dGBP1$^{ZH-86Fb}$ in the *kni* domain. These clusters moved along with the cells in elongating branches, finally accumulating mostly at intercellular AJs, including prominent foci at the tip of the branch (in between the two tip cells; Fig. 7 b, arrows and Video 10). In most cases, however, we observed adequate AJs rearrangements, reflecting two migrating tip cells and stalk cell intercalation and elongation ($n_{branches\ intercalated}$ = 35/36 from $n_{embryos}$ = 14 for N-Rok::dGBP1$^{ZH-86Fb}$ and $n_{branches}$ = 32/32 from $n_{embryos}$ = 12 for the control; on average 2–3 branches per embryo were analyzed, Fig. 7 b and Video 9 and Video 10).

We, therefore, conclude that excessive phosphorylation of Sqh::GFP by synthetic N-Rok::dGBP1$^{ZH-86Fb}$ specifically in the *kni* domain during dorsal branch formation did not preclude filopodia activity in tip cells, allowing for cell migration, branch elongation, and stalk cells intercalation in most branches, despite causing aberrant behavior/contacts establishment between some of the tip cells. In most cases, autocellular junctions were properly formed, although in some cases the regular spatial arrangement of stalk cells was somewhat disrupted or—in most severe cases—parts of the cells detached from the rest of the branch resulting in an aberrant branch network.

### Design of engineered Src kinase for targeted protein phosphorylation

To test the modularity of our synthetic kinase approach, we next generated synthetic Src kinases. Src family members are non-receptor tyrosine (Y) kinases expressed in a wide variety of cells and have been shown to play crucial roles in several cell biological processes, such as cell division, cell migration, cell-shape changes, cell–substratum, and cell–cell interactions (reviewed in Parsons and Parsons [2004]; Playford and Schaller [2004]). The function of *Src42A*, partially redundant with *Src68B*, is required in many processes during *Drosophila* development (Shindo et al., 2008; Takahashi et al., 2005; Tateno et al. 2000). Src42A is

activated by phosphorylation on Y400 and, more importantly, inhibited by phosphorylation on Y511 (Piwnica-Worms et al., 1987; Pedraza et al., 2004; Cartwright et al., 1987). Src42A has an N-terminal myristoylation site, two central SH3 and SH2 domains important for substrate recruitment, and a C-terminal kinase domain (Boggon and Eck. 2004). Src is normally present in cells in its inactive form, in which Y511 is phosphorylated and the protein folded in such a way that the SH2 domain binds to phosphorylated Y511 and inhibits Src42 activity. Upon dephosphorylation at Y511, the "lock" is released, resulting in the unfolding of the molecule and allowing for the intermolecular auto-phosphorylation of Y400, which keeps the kinase in the active state (Boggon and Eck 2004). Cellular Src acts as a crucial co-transducer of transmembrane signals from various growth factor receptors, allowing for rapid response to external cues (Okada 2012). Src phosphorylates multiple substrates, including proteins associated with the cytoskeleton and adherens junctions (FAK, cortactin, paxillin), as well as Crk-associated substrate (Cas), Ras, and the Jun amino-terminal kinase (JNK; Reynolds et al., 2014; Goldberg et al., 2003; Bunda et al., 2014; Zhang et al., 2004; Tateno et al. 2000). In *Drosophila*, the activation of the sole JNK homolog Basket (Bsk) downstream of Src42A is required for dorsal closure of the embryonic epidermis (Tateno et al. 2000). The activation of JNK/Bsk results in the phosphorylation of c-Jun and many other transcription factors that drive the cellular response.

To generate engineered Src versions, we cloned a fusion construct of the N-terminal dGBP1 with the C-terminal part of *Drosophila* Src42A spanning the ATP-binding site, the catalytic kinase domain, and the C-terminal regulatory tyrosine residue (Fig. 8, a and b). Engineered Src is therefore devoid of the two SH domains important for physiological substrate recognition (Waksman et al., 1992; Mayer 2001; Boggon and Eck. 2004). Similar to N-Rok::dGBP1, the engineered Src with dGBP1 should be only stable in the presence of GFP and degraded in the absence of a GFP fusion protein (Fig. 8 a; Tang et al., 2016). A constitutively active, full-length Src42A carrying a single amino acid substitution of the crucial regulatory tyrosine with phenylalanine (Y511F) was previously generated and validated (Tateno et al. 2000). We, therefore, introduced Y511F in all constructs, creating a non-phosphorylatable site to obtain a truncated, constitutively active kinase (SrcCA). In dGBP1-HA::Src Y400E and dGBP1-HA::Src Y400D, a second single amino acid substitution corresponding to Y400 in wild-type Src42A was introduced to attenuate potentially overactive SrcCA (Fig. 8 c). dGBP1-HA::SrcDead contains a single amino acid substitution corresponding to K276M in Src42A, which abolishes catalytic activity (Fig. 8 c; Pedraza et al., 2004).

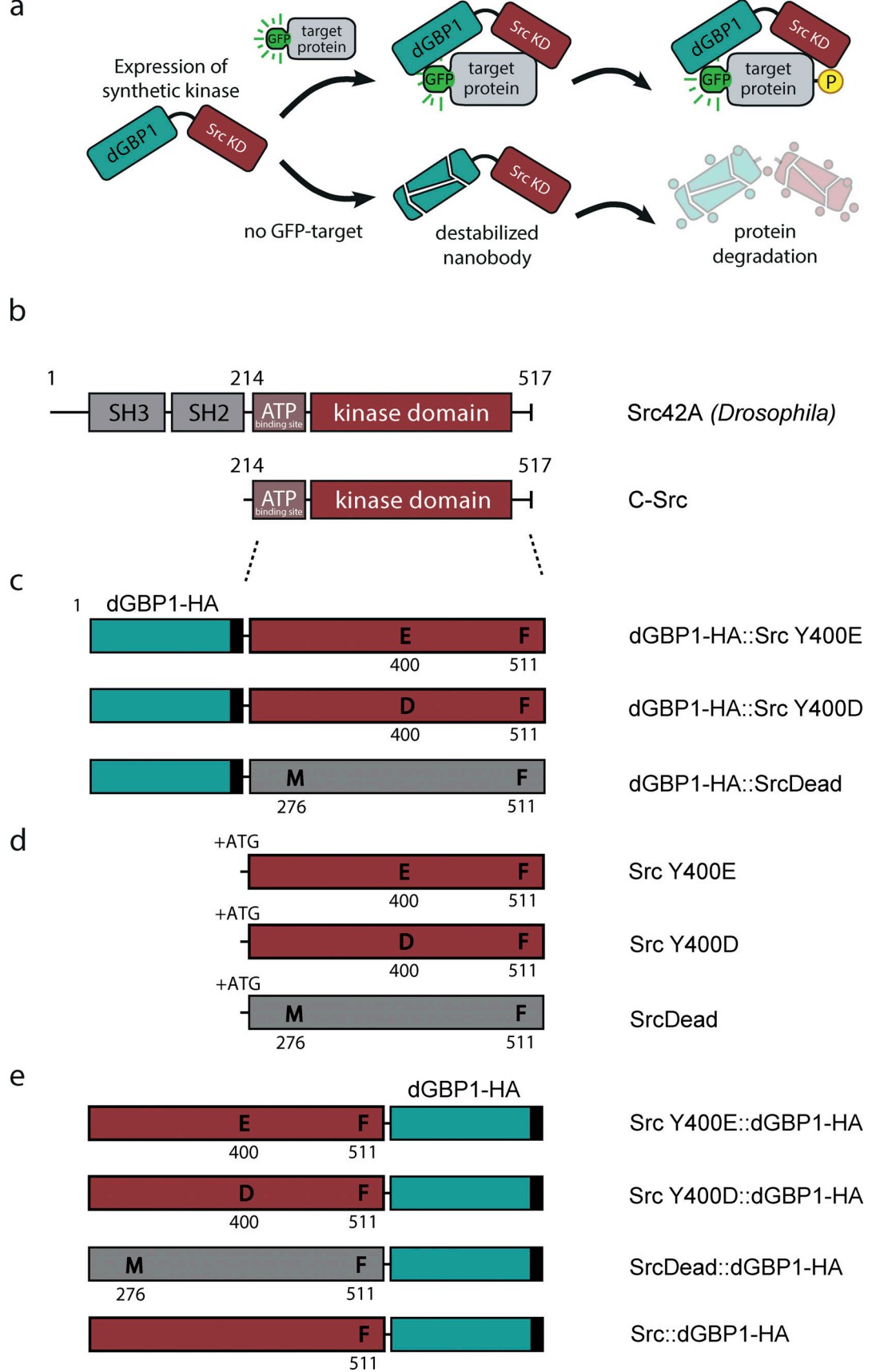

Figure 8. **Schematic illustration of engineered synthetic Src kinases. (a)** Schematic illustration of the synthetic Src kinase work concept. In a similar way as in Fig. 1 a and Fig. 4 c, synthetic kinase uses GFP to bring a constitutively active kinase domain (Src KD) in close proximity to a fluorescent fusion substrate

protein (target). The persistence of the kinase domain around the fluorescent fusion protein allows for efficient phosphorylation (P) of the target. In the absence of the GFP-target, dGBP1 nanobody is destabilized, and the whole nanobody-kinase fusion protein is targeted for degradation. **(b)** The structure of *Drosophila* Src42A protein. The C-terminal part of *Drosophila* Src42A (amino acid 214–517) spanning the ATP-binding site, Src catalytic domain, and the crucial regulatory tyrosine residue Y511 was used for synthetic kinase constructs shown in (c–e). **(c)** Linear representation of dGBP1-HA::Src Y400E, dGBP1-HA::Src Y400D and dGBP1-HA::SrcDead in which N-terminal dGBP1 with HA tag (black squares) was fused with C-terminal part of Src shown in b. **(d)** Linear representation of Src Y400E, Src Y400D, and SrcDead for which the exact same part of Src42A was used as for the constructs presented in c. These constructs lack the dGBP1 binder, and an in-frame START codon (ATG) was added N-terminally. **(e)** Linear representation of Src Y400E::dGBP1-HA, Src Y400D::dGBP1-HA, SrcDead::dGBP1-HA and Src::dGBP1-HA in which the order is changed: the dGBP-1-HA binder was fused C-terminally to the Src part and an in-frame START codon (ATG) was added N-terminally. In Src::dGBP-1-HA the regulatory tyrosine Y400 is left unmutated. SrcDead constructs contain a K276M single amino acid substitution (M) that abolishes catalytic activity. Numbers refer to amino acid positions from N-terminus (N) to C-terminus (C).

To investigate whether any observable effect is indeed dependent on the binding of the attached nanobody to a GFP-fusion target, we also generated a series of kinase constructs encoding SrcCA variants without a fused binder, namely Src Y400E, Src Y400D, and SrcDead (Fig. 8 d). To further interrogate any potential effect of the site of the placement of the protein binder, we cloned the same Src variants with a C-terminally added dGBP1, creating Src Y400E::dGBP1-HA, Src Y400D::dGBP1-HA, and SrcDead::dGBP1-HA (Fig. 8 e). For this set of Src constructs, we also cloned Src carrying only the Y511F mutation, without further manipulation of the auto-regulatory site Y400, and called this construct Src::dGBP1-HA (Fig. 8 e).

**Engineered Src efficiently phosphorylates Bsk::GFP in vivo**

Identical to the engineered Rok kinases, all Src constructs were introduced via the phiC31 integrase system (Bischof et al., 2007) in the third chromosome. To assess the function of the synthetic Src kinase in vivo, we used the sole JNK homolog Basket (Bsk) fused to GFP as a target. We chose Bsk/JNK as a substrate as it has been shown that Src42A regulates Bsk during dorsal closure (Tateno et al., 2000), and a specific antibody raised against mammalian phosphorylated JNK exists and has been shown to work in flies (Spindler et al., 2012). No endogenously tagged Bsk fly line is currently available, and we, therefore, used a *UAS_bsk::GFP* line and assessed the levels of phosphorylated Bsk upon the expression of the different Src variants with an anti-pJNK antibody, as well as the general phospho-tyrosine levels (pY).

We first assessed whether the overexpression of dGBP1-HA::Src Y400E and dGBP1-HA::Src Y400D with *enGal4* in the absence of a GFP-fusion target triggered any non-specific phosphorylation in the posterior compartment of each segment of the embryonic epidermis. We did not observe an increase in the phosphorylation levels for either of the active Src variants, suggesting that Src constructs are indeed rather inert in the absence of a GFP-target ($n = 13$ and $n = 16$ for dGBP1-HA::Src Y400E and dGBP1-HA::Src Y400D, respectively; Fig. 9 a). As expected, when the inactive variant of Src was used (dGBP1-HA::SrcDead), we also did not see any effect on pJNK nor pY ($n = 10$; Fig. 9 a).

We next used *enGal4* with *UAS_bsk::GFP* to drive the expression of the GFP-target and the different Src constructs. We first assessed pJNK and pY levels in embryos expressing only *UAS_bsk::GFP* and noticed no increase in the phosphorylation signal in the posterior compartment ($n = 12$; Fig. 9 b). In contrast, embryos expressing engineered dGBP1-HA::Src Y400E and dGBP1-HA::Src Y400D robustly phosphorylated GFP-tagged Bsk,

as indicated by increased pJNK and pY levels ($n = 10$ for each; Fig. 9 b). Such effects were not observed when the inactive variant of Src was used (dGBP1-HA::SrcDead, $n = 14$), and the staining pattern looked similar to the one seen in the control embryos expressing only Bsk::GFP (Fig 9 b) or synthetic Src variants in the absence of GFP target (Fig. 9 a).

To further confirm that the increase in the phosphorylation of Bsk::GFP was indeed dependent on the binding of the kinase to the fluorescent fusion protein, we analyzed pJNK and pY levels in embryos expressing Src variants without the fused nanobody, namely SrcDead, Src Y400E, and Src Y400D ($n = 15$, $n = 13$, $n = 10$, respectively; Fig. S5 a). We observed no increase in the phosphorylation signal in the posterior compartment for any of the Src constructs, confirming that direct interaction between the protein binder and the GFP target was indeed essential for eliciting efficient phosphorylation of the tagged substrate.

Lastly, we expressed the reversely organized Src variants (Src Y400E::dGBP1-HA, Src Y400D::dGBP1-HA, SrcDead::dGBP1-HA, and Src::dGBP1-HA) and Bsk::GFP in a similar set of experiments. As expected, the expression of SrcDead::dGBP1-HA did not lead to pJNK nor pY signal increase ($n = 10$; Fig. S5 b). In embryos expressing Src Y400E::dGBP1-HA or Src Y400D::dGBP1-HA, we detected a moderate increase in pY signal in the posterior compartment and a slight increase in pJNK signal ($n = 5$ and $n = 7$, respectively; Fig. S5 b), visibly lower than for the dGBP1-HA::Src Y400E and dGBP1-HA::Src Y400D variants (Fig. 9 b). Interestingly, the expression of Src::dGBP1-HA with the regulatory tyrosine Y400 unmutated resulted in very robust phosphorylation of Bsk::GFP ($n = 9$; Fig. S5 b).

Overall, our results show that engineered Src is a functional enzyme and its activity can be attenuated via Y400 amino acid substitution. Together with synthetic Rok, we provide solid evidence that engineered kinases can be used as a general tool to specifically phosphorylate a tagged substrate in a chosen tissue in vivo.

## Discussion

The study of post-translational modifications is essential to understand how proteins regulate complex cell behaviors. While much is known from in vitro studies, dissecting the functional role of specific protein modifications requires their manipulation in vivo. In this study, we generated and characterized a series of genetically encoded synthetic kinases using protein binders. We provide evidence that such constructs can be used to directly phosphorylate specific target proteins in vivo.

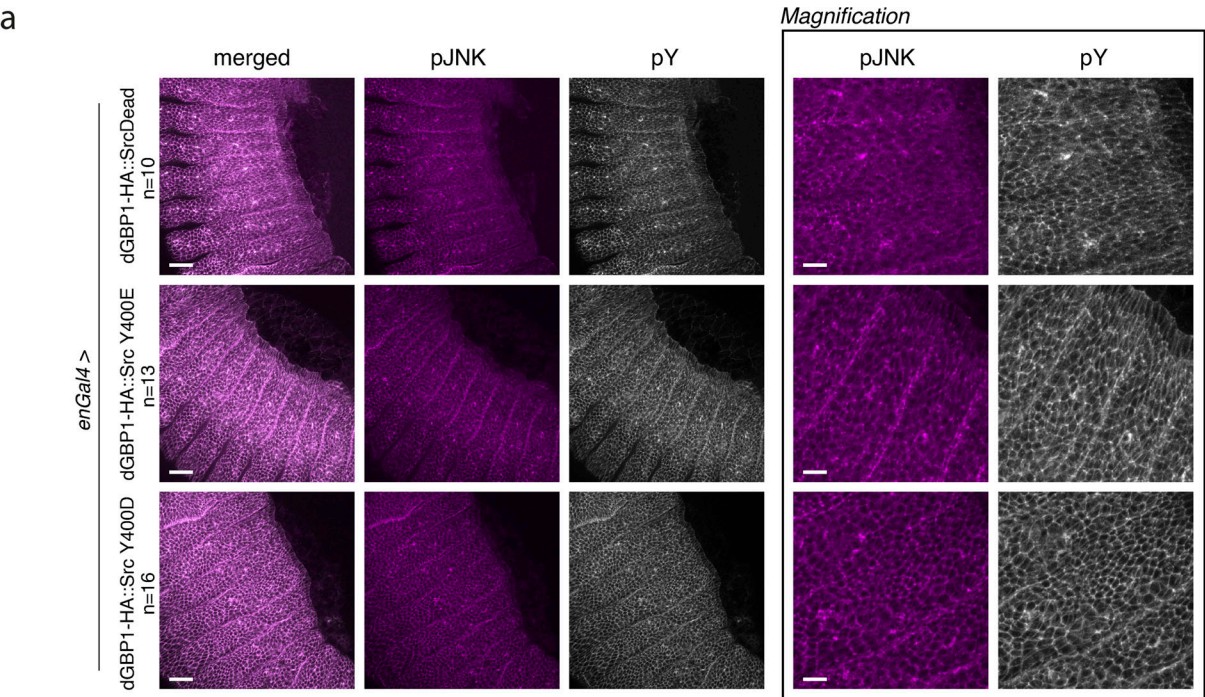

Figure 9. **Engineered dGBP1-HA::Src is a functional enzyme and efficiently phosphorylates bsk::GFP in vivo. (a)** Panels show lateral views of fixed *Drosophila* embryos at stages 13–14 (dorsal closure) expressing synthetic Src kinase variants in the *engrailed* domain in the absence of GFP-fusion substrate. Embryos were stained with anti-phospho-JNK antibody that recognizes Bsk, the *Drosophila* JNK ortholog, and an anti-phosphotyrosine antibody (pY), that detects tyrosine phosphorylated proteins in all species. **(b)** Panels show lateral views of fixed *Drosophila* embryos at stages 13–14 (dorsal closure) expressing

bsk::GFP and synthetic Src kinase variants in the *engrailed* domain. Embryos were stained as in panel a. The right panel in both a and b shows the magnification of the respective pJNK and pY images for each of the embryo genotypes shown on the left. Arrows point to the areas of increased pJNK and pY signal in the *engrailed* domain where the active kinase is expressed. For every expressed synthetic kinase, the number of considered embryos is indicated (*n*). Scale bar, 25 µm; in the zoomed panel, 10 µm.

We show that protein binders recognizing fluorescent proteins can be used to direct the activity of otherwise inert catalytic domains of Rok. Tissue-specific expression of these engineered Rok kinases results in efficient phosphorylation of the fluorescently tagged Sqh/MRLC, which results in actomyosin contractility and causes aberrant cell behavior. To demonstrate the utility of the method, we have studied these engineered Rok kinases in two morphogenetic processes in the developing fly embryo, in dorsal closure, and in tracheal development. Given the modularity of our approach, we assume that this tool can be adapted to other kinases and targets in various eukaryotic genetic systems. To support this proposal, we have generated a series of synthetic Src kinases and validated their ability to specifically phosphorylate a tagged protein substrate. In spite of its great potential, the extension of this method to other substrate/kinase interactions will require careful validation of each novel synthetic kinase generated.

The use of protein binder-based approaches to direct a given enzymatic activity to a chosen target opens new avenues for unravelling the importance of given protein–protein interactions. Directed protein manipulation in specific cell types has multiple applications, from developmental studies to designing synthetic biology systems and eventually to therapeutic uses. A similar nanobody-based paradigm has been recently applied to induce glycosylation and deglycosylation of desired target proteins in cells (Ramirez et al., 2019; Ge et al., 2021). Furthermore, in the elegant work by Karginov et al. (2014b), a rapamycin-regulated engineered Src kinase was used in cultured cells to dissect the effect of Src-dependent phosphorylation of its two substrates, FAK and p130Cas, on cell morphology and filopodia formation.

The effect of activation of myosin II-dependent contractility has been investigated in various biological systems to study the shaping of cells and tissues during development, to validate mechanisms regulating myosin II function, and to study tumor migration and invasiveness. To achieve this, several tools that allow for myosin II activation have been developed (Winter et al., 2001; Verdier et al., 2006; Kim et al., 2002; Urbano et al., 2018). Many of the recent methods that allow acute, inducible activation of myosin II rely on optogenetic or optochemical tools. Despite permitting precise spatiotemporal control, most of these tools act upstream of Rok (i.e., light-controlled activation of small GTPase RhoA or the optochemical control of $Ca^{2+}$ levels), exhibiting pleiotropic effects on cell contractility (Valon et al., 2017; Wagner and Glotzer. 2016; Oakes et al., 2017; Izquierdo et al., 2018; Kong et al., 2019), and lacking the specificity we present here. Also, we show that the destabilized variant of the engineered Rok (N-Rok::dGBP1) allows for minimal interference in in vivo studies when the target fusion protein is not present in the cell. Using destabilized nanobodies as fusion partners to reduce non-specific activities is a major step closer to using even more specific tools to manipulate protein function.

**Effect on tracheal cell intercalation**

Past work from our lab has shown that tracheal cell intercalation relies on the mechanical pulling of the stalk cells by leading tip cells and does not require actomyosin contractility (Ochoa-Espinosa et al., 2017). During the extensive junctional rearrangements, cells establish autocellular junctions, which are cell self-contacts. Surprisingly, previous studies have shown that autocellular contacts are normally eliminated in cultured cells through cell-self fusion, which requires actomyosin contractility (Sumida and Yamada. 2013). The exclusion of E-cadherin from contact sites has been proposed to be involved in eliminating self-contacts in different cultured cells (Sumida and Yamada. 2013; Sumida and Yamada. 2015). We, therefore, hypothesized that the persistence of cell self-contacts in tracheal branches might result from insufficient levels of active actomyosin, thereby preserving the autocellular junctions.

Interestingly, herein we showed that ectopic activation of actomyosin via engineered Rok kinase did not result in a general failure of stalk cell intercalation nor did it interfere with the formation of autocellular junctions. Nevertheless, we observed a gradation in the observed phenotypes ranging from normal branch elongation and cell intercalation (despite evident ectopic points of myosin activation at cell-cell contacts) to more severe phenotypes displaying aberrant contacts between the tip cells. We even observed the detachment of some of the migrating cells from the rest of the branch resulting in an aberrant tracheal network. Strikingly, expression of the synthetic Rok kinase only resulted in the accumulation of Sqh foci along the intercellular junctions and not along the autocellular junctions. It thus remains possible that autocellular junctions are maintained by preventing the accumulation of actomyosin, particularly at the latter, and this exclusion cannot be overcome with the synthetic kinase used here. The importance of precisely dissecting the role of actin and myosin II in junctional rearrangements and autocellular junction formation is emphasized by the recent findings that autocellular AJs are also present in the retinal and adult mouse brain microvasculature (Kotini et al., 2019).

Of note, a recent study from the Llimargas lab has shown that moderate levels of constitutively active Diaphanous (Dia), a protein that plays important roles in the nucleation and elongation of F-actin filaments, disturb cell elongation but not cell intercalation, while high levels of constitutively activated Dia impair both cell elongation and intercalation, resulting in altered overall tracheal morphology (Casani et al., 2020). Their data and ours (this manuscript) support the notion that cortical tension gradients resulting from local modulation of different pools of actomyosin lead to various impacts on cell fitness/deformation. Further work is needed to fully understand the effect of

modulating different pools of actin on cell intercalation and elongation in tracheal branches.

### Engineered Src

It was unclear from previous studies whether the isolated kinase domain of c-Src could be used as a constitutively active form and how exactly the Y400 mutation influences Src activity. Our results show different efficacies of various engineered Src variants and may help in future attempts to create phosphomimetic tyrosine variants that often employ substitution of tyrosine with glutamate (E) or aspartate (D). In our context, it seems that maintaining the Y400 site in an unmutated state and relying on the kinase auto-activation ability results in much stronger kinase activity than if Y400 is substituted with E or D (compare Src::dGBP1-HA with the regulatory tyrosine Y400 unmutated to Src Y400E::dGBP1-HA or Src Y400D::dGBP1-HA [Fig. S5 b]).

During the course of our study, we also generated Src variants with an anti-GFP DARPin 3G86.32 (3G86.32::Src Y400E, 3G86.32::Src Y400D, and 3G86.32::SrcDead; see Brauchle et al., 2014). We observed very strong pJNK and general phosphotyrosine signal enhancement upon expression of these two active Src variants fused to anti-GFP DARPin in the presence of Bsk::GFP (data not shown). However, similar increased levels of pJNK and pY were also observed in the absence of the GFP-fusion protein, presumably due to the stable nature of DARPin binders, resulting in the persistent activity of engineered Src in the cells and leading to non-specific robust phosphorylation of untagged endogenous proteins.

In sharp contrast, active Src variants fused to the destabilized nanobody effectively phosphorylated Bsk::GFP, while being innocuous in the absence of the GFP target. Of note, and as seen also for N-Rok kinase, the substitution of the anti-GFP nanobody (vhhGFP4) or the anti-GFP DARPin (3G86.32) with the destabilized variant greatly improved the tool in being similarly efficient but non-toxic. Together, this underlines the importance of appropriate binder choice for reducing the off-target effect and justifies the use of N-Rok::dGBP1 variant for the biological assays we performed in the fly trachea (see Table S2 for the summary of binders used in this study).

In summary, we observed a gradient of potency of various engineered Src variants, which was dependent on several features: (1) the presence or absence of Y400 substitution (Y400 unmutated vs. Y400E vs. Y400D); (2) the type of the binder used (3G86.32 DARPin vs. dGBP1), and (3) the position of the nanobody (N-terminal vs. C-terminal). We propose that, together with the choice of the linker between the two engineered kinase parts, the features listed above can be used as a general guide for the generation and optimization of other synthetic kinases based on our system or other synthetic tools employing protein binders.

### Future perspectives

Further possible avenues for expanding the use of engineered kinases and other enzymes include phosphorylation of target proteins in specific subcellular compartments, for example, only in the apical domain of a cell. This can be achieved by adding another module to our tool, in this case containing discrete localization motifs. During the course of this study, we have made use of an N-Rok synthetic kinase fused with an apical localization domain (ALD::N-Rok::vhhGFP4-HAZH-86Fb) to restrict the activity of N-Rok to the apical cortex in interphasic *Drosophila* neuroblasts (Roubinet et al., 2017). An important future improvement of protein binder-based synthetic biology would be to make target binding acutely inducible, either by light or by chemicals (Natsume et al., 2016; Banaszynski et al., 2006; Chung et al., 2015; Bonger et al., 2011; Niopek et al., 2016; Buckley et al., 2016; Rost et al. 2017; Tischer and Weiner 2014). Combining the target-specific action presented in this manuscript with acute control over the activity of the fusion protein would allow an unprecedented level of spatiotemporal control of enzymatic activity and specificity. Due to the large collection of available *Drosophila* strains with in-frame GFP or YFP insertions, our engineered Rok and Src kinases (or, in a broader context, any other chosen kinase-protein binder against GFP/YFP) can now be easily tested on other GFP-fusion targets. Another intriguing avenue would be to define whether such engineered kinases can be reprogrammed to phosphorylate targets that are not their natural substrates. This would provide means for rewiring the existing signaling pathways in a cell or the entire organism. Pioneering work on re-engineered PAK4 kinase (Miller et al., 2019), DYRK1A kinase (Lubner et al., 2016), and Pim1 kinase (Chen et al., 2017) provided proof-of-principle for reprogramming kinases to recognize novel substrates, in which the recognition site and motif are well known.

In summary, we present the generation and detailed characterization of a promising tool to control kinase activity and show a clear relationship between a kinase-selected target pair and cellular behavior in morphogenetic processes in vivo.

## Materials and methods

### Generation of engineered kinase constructs

(1) Engineered Rok: To generate pUASTattB_N-Rok::vhhGFP4 and pUAST_N-Rok::vhhGFP4, the N-terminal kinase region of Drosophila Rok (N-Rok) was fused to vhhGFP4 and inserted into respectively pUASTattB vector (Bischof et al., 2007) or pUAST with EcoRI and XbaI. In pUASTattB_N-Rok::vhhGFP4-HA, a C-terminal Human influenza hemagglutinin tag (HA) was added. In N-RokDead variants, a K116G single amino acid substitution (K->G) that abolishes catalytic activity (Winter et al., 2001) was introduced by site-directed mutagenesis. In pUASTattB_N-Rok::dGBP1, vhhGFP4 was replaced by a destabilized GFP-binding nanobody (dGBP1; Tang et al., 2016). To generate pUASTattB_N-Rok::2m22 and pUASTattB_N-RokDead::2m22, the N-Rok variants were fused to a DARPin recognizing mCherry (2m22; Brauchle et al., 2014).

(2) Engineered Src: To generate pUASTattB_dGBP1::C-Src variants (dGBP1::Src Y400E, dGBP1::Src Y400D, and dGBP1::SrcDead), we fused dGBP1 (Tang et al., 2016) containing the C-terminal part of *Drosophila Src42A* spanning the ATP-binding site, the catalytic kinase domain, and the C-terminal regulatory tyrosine residue (amino acid 214–517) with mutated Y511F and inserted it into pUASTattB vector. A C-terminal Human influenza hemagglutinin tag (HA) was added to dGBP1 (dGBP1-HA).

dGBP1-HA was amplified by PCR and inserted with EcoRI/XhoI and C-Src was inserted with XhoI/XbaI. In dGBP1-HA::Src Y400E and dGBP1-HA::Src Y400D, a second single amino acid substitution corresponding to Y400 in wild-type Src42A was introduced to attenuate potentially overactive SrcCA. In dGBP1-HA::SrcDead, a single amino acid substitution corresponding to K276M in Src42A was added, which abolishes catalytic activity (Pedraza et al., 2004). To generate Src constructs without a binder, the same C-terminal part of *Drosophila Src42A* was amplified by PCR and inserted with XhoI/XbaI. In these constructs, an in-frame START codon (ATG) was added N-terminally. To generate pUASTattB_C-Src variants with a C-terminally added dGBP1 (Src Y400E::dGBP1-HA, Src Y400D::dGBP1-HA, SrcDead::dGBP1-HA, and Src::dGBP-HA), the N-Rok domain of the pUASTattB_N-Rok::dGBP1-HA was removed with EcoR1/Eag1 and replaced with the different PCR amplified Src domains (amino acid 214–517). Src::dGBP-HA carries only the Y511F mutation without further manipulation of the auto-regulatory site Y400.

The resulting constructs were all verified by sequencing. Oligos used are available on demand.

See Data S1 for sequences.

### Immunofluorescence staining of S2 cells

Stable *Drosophila* S2 cell line expressing Sqh::GFP (S2 Sqh::GFP; Rogers et al., 2004) was grown in Schneider's *Drosophila* medium (Invitrogen) supplemented with 10% heat-inactivated fetal bovine serum (FBS) and penicillin–streptomycin (at a final concentration of 50 units penicillin G and 50 µg streptomycin per ml of medium) at 25°C. Cells were transfected with the indicated plasmids using FuGENE HD (Roche) according to the manufacturer's instructions. Cells were fixed with 4% PFA for 30 min at RT 40 h after transfection and processed for immunofluorescence using standard procedures. Briefly, blocking and permeabilization wereperformed in TBS (20 mM Tris-HCl, pH 7.5, 154 mM NaCl, 2 mM EGTA, 2 mM MgCl2) containing 2% BSA and 0.02% saponin for 1.5 h. Staining was done using the same solution. Antibodies used were mouse anti-HA (12CA5, 1:200; Roche), rabbit anti-Sqh2P (1:200; Thermo Fisher Scientific), rat anti-Sqh2P (1:200; Zhang and Ward. 2011), and rabbit anti-Sqh1P (1:50; Zhang and Ward. 2011). Cells were resuspended in Vectashield mounting medium containing DAPI, transferred to microscope slides and covered with a 22-mm² coverslip. Statistical significance was determined with Student's *t*-test using GraphPad Prism software. The Prism convention is used: n.s., $P > 0.05$; *, $P \leq 0.05$; **, $P \leq 0.01$; ***, $P \leq 0.001$; and ****, $P \leq 0.0001$.

### Experimental model

*Drosophila melanogaster* was cultured using standard techniques at 22–25°C on a standard corn meal *Drosophila* medium supplemented with yeast. For virgin collection, fly vials were often kept at 18°C for the period of collection; both male and female animals were used.

### Generation of transgenic fly lines

To achieve the same level of expression of the different engineered kinase constructs with the same Gal4 driver, we used the phiC31 integrase (Bischof et al., 2007) system and a unique site of integration (ZH-86Fb; Bischof et al., 2007) on the third chromosome to generate transgenic lines carrying *N-Rok::vhhGFP4*[ZH-86Fb], *N-RokDead::vhhGFP4*[ZH-86Fb], *N-Rok::vhhGFP4-HA*[ZH-86Fb], *N-RokDead::vhhGFP4-HA*[ZH-86Fb], *N-Rok-HA*[ZH-86Fb], *N-RokDead-HA*[ZH-86Fb], *N-Rok::2m22*[ZH-86Fb], *N-RokDead::2m22*[ZH-86Fb], *N-Rok::dGBP1*[ZH-86Fb], *dGBP1-HA::Src Y400E*, *dGBP1-HA::Src Y400D*, *dGBP1-HA::SrcDead*, *Src Y400E*, *Src Y400D*, *SrcDead*, *Src Y400E::dGBP1-HA*, *Src Y400D::dGBP1-HA*, *SrcDead ::dGBP1-HA, and Src::dGBP1-HA*. The *N-Rok::vhhGFP4*[Vi] fly line was generated by random genomic insertion of N-Rok::vhhGFP4 by P element-mediated transformation (Spradling and Rubin 1982).

### *Drosophila* stocks and genetics

All fly strains used in this study are listed in Table S3. Details on the exact crossing schemes are available upon request.

### Antibody staining of *Drosophila* embryos

Embryos were collected on grape juice agar plates supplemented with yeast paste after overnight egg laying at 25°C and processed for immunofluorescence using standard procedures. Briefly, embryos were collected on a nitex basket, dechorionated in 50% bleach (sodium hypochlorite, stock solution 13% w/v technical grade; AppliChem GmbH), washed thoroughly with dH2O, and fixed in 50:50 heptane: 4% paraformaldehyde solution for ~20 min with vigorous shaking. Embryos were devitellinized by replacement of the fix phase with the same volume of MeOH and vigorous handshaking for 1 min. After washing four times with MeOH, the embryos were either stored at –20°C to accumulate enough material or directly subjected to staining. After brief rehydration of fixed embryos in 1:1 MeOH:1xPBS solution, embryos were rinsed with 1xPBS, washed 4 × 15 min in PBST (PBS + 0.3% Triton X-100), blocked in PBTN (PBST + 2% normal goat serum) for 1 h, and incubated overnight with primary antibodies in PBTN at 4°C on a rotator. Embryos were rinsed twice and then washed 4 × 20 min with PBST and incubated with secondary antibodies in PBTN for 2–3 h, washed as before, and rinsed 2× with PBS. Embryos were mounted on a microscope slide in Vectashield mounting medium (H-1000; Vector Laboratories) and covered with a 22-mm² coverslip.

The monophosphorylation on serine 21 in embryos was detected with a guinea pig anti-Sqh1P antibody serum in 50% glycerol (gift from R. Ward; Zhang and Ward. 2011) at 1:400 concentration or a rabbit anti-phospho-Myosin Light Chain 2 (Ser19; #3671; Cell Signaling Technology) at 1:50 concentration. Other antibodies used are as follows: rat anti-DCAD2 at 1:25 (DSHB), rabbit anti-vermiform 1:300 (Luschnig et al., 2006), rabbit anti-phospho-SAPK/JNK (Thr183/Tyr185) mAb (81E11; #4668; Cell Signaling Technology) 1:300, mouse anti-phosphotyrosine, and clone 4G10 (05–321; Sigma-Aldrich) 1:400. Secondary antibodies used were Alexa Fluor 488/Fluor 568/Fluor 647 coupled (1:500; Thermo Fisher Scientific).

### Embryo mounting for live time-lapse imaging

Embryos were dechorionated in 30–50% bleach (see above) and extensively rinsed in water. Embryos at desired stages were selected manually on grape juice agar plate under a dissection

scope and mounted either with heptane-glue method (I) or on glass-bottom dish (II).

(I) Briefly, selected embryos were manually aligned on agar plates and attached to heptane-glue coated coverslips by gently pressing the trail of glue against the embryos. A piece of permeable bio-foil (bioFOLIE 25, In Vitro System and Services; GmbH) was stretched over custom-made metal slides with a hole, and embryos were mounted in a drop of halocarbon oil 27 (Sigma-Aldrich) between the foil and the coverslip (see Caussinus et al. [2013] for details of the procedure).

(II) Embryos were transferred with cleaned forceps to a glass-bottom dish (MatTek, 35 mm dish, co. 1.5 coverslip, uncoated, P35G-1.5-10-C). After adding 1xPBS, embryos were gently rolled to the desired positions using a cut gel loading pipette tip (flexible, to minimize embryo disruption). Properly dechorionated embryos adhere to the glass-bottom; the part to be imaged was facing down for the inverted microscope.

### Confocal and time-lapse imaging

Images of fixed embryos were acquired with 1,024 × 1,024 frame size with a 40×/1.25 NA oil objective (HCX PL APO) on a Leica TCS SP5 II inverted scanning confocal microscope with or with a 40× silicon oil objective on Point Scanning Confocal Zeiss LSM880. A series of z-stacks was acquired for each embryo at 0.5–1 μm step size, 1–2.5 zoom using a 488, 561, and 633 nm laser line.

Time-lapse sequences of dorsal closure or whole trachea development were imaged under either (a) Point Scanning Confocal Zeiss LSM880 AiryScan inverted microscope (Confocal mode) with PLAN APO 40×/1.2NA objective (LD LCI PLAN APO, Imm Corr DIC M27, water immersion) with 0.9 zoom using Zen Black software at 25°C or (b) Leica TCS SP5 II inverted scanning confocal microscope with 40×/1.25NA oil objective (HCX PL APO; embryos mounted in oil with heptane glue method) or 40×/1.10NA water objective (HCX PLAN APO CS; glass-bottom dish mounted embryos) using LAS AF software at room temperature. A series of z-stacks was acquired for each embryo at 0.5–1 μm steps using a 488- and 561-nm laser line. Imaging was carried out at 3–5 min intervals (movies corresponding to Fig. 3, b and c; and Figs. 6 and 7) or 10-min intervals (movies corresponding to Fig. 3 a; Fig. 4, b and e; and Fig. 5 b).

For dorsal branch imaging, a series of z-stacks were acquired for each embryo at 0.25 μm steps at 3–5 min intervals using Point Scanning Confocal Zeiss LSM880 AiryScan inverted microscope with AiryScan detector (FAST Optimized mode) with the voxel size: 0.0993 × 0.0993 × 0.2455 micron^3. Raw images were AiryScan processed with Zen Black software using the "auto" settings to obtain the best reconstruction possible.

### Confocal image processing

Z-stack maximum projections were assembled in Fiji (ImageJ; Schindelin et al., 2012) or OMERO (Allan et al., 2012). If needed, the signal was enhanced by multiplication, i.e., until the Sqh::GFP signal clearly marked the lateral epidermis boundaries. For movies, x–y drift in time was corrected using the StackReg plugin with "rigid body" transformation (Thévenaz et al. 1998). Timepoint zero was defined as the beginning of germ-band retraction (trachea imaging) or right before the onset of zippering (70–90 min after germband retraction completion) at stage 13/14 for dorsal closure. Figures were prepared using OMERO Figure (Allan et al., 2012) and Adobe Illustrator.

### Quantification and exclusion criteria

The number of analyzed embryos of each genotype is shown on the respective figures and indicated in the corresponding text. The following cases were excluded from analysis and were not counted: embryos that were rotated at the time zero or rotated during imaging to an extent that the part of interest was no longer visible/was out of focus and embryos with clear signs of mechanical disruption (pinching of embryo membrane while mounting). For fixed samples, only embryos at the stage range indicated below the figure were analyzed. Embryos with clear mechanical disruption signs (independent of the phenotype) or embryos with aberrantly low control mCherry-nls signal were excluded. The data shown in Fig. 2 b was collected from at least three independent experiments. For the analysis, "n" refers to the number of cells analyzed and is mentioned in the figure legend. Statistical significance was determined with the Mann–Whitney test using GraphPad Prism software and the Prism convention is used on the graph. The data distribution was tested for normality using both D'Agostino-Pearson and Anderson-Darling tests, which revealed non-normal distributions.

### Online supplementary material

Table S1 supports Fig. 3 and shows the viability and/or adult fly phenotypes induced by the expression of different synthetic kinase constructs in the absence of the GFP-target. Table S2 shows the summary of protein binders used in this study. Table S3 contains detailed information on the reagents, genetically modified organisms, and cell lines used. Data S1 is a multi-FASTA file corresponding to all the inserts used in this work. Figs. S1 and S3 support Fig. 2 d; Fig. S2 supports Fig. 3 and shows further evaluation of various engineered Rok variants. Fig. S4 supports Figs. 2 d and 4 and shows the evaluation of the specificity of engineered Rok kinase. Fig. S5 supports Fig. 9 and shows further evaluation of synthetic Src kinase variants. Video 1 supports Fig. 3 a; Video 2 and Video 3 - Fig. S2; Video 4 - Fig. 3, b and c; Video 5 - Fig. 4, b and e; Video 6 - Fig. S4 c. Videos 1, 2, 3, 4, 5, and 6 show dorsal closure in embryos expressing variants of engineered Rok. Video 7 supports Fig. 5 b and shows tracheal development in embryos expressing variants of engineered Rok. Video 8 supports Fig. 6 b and shows the influence of actomyosin activation on dorsal branch elongation. Videos 9 and 10 support Fig. 7 b and show the influence of actomyosin activation on cell intercalation in dorsal branches.

## Data availability

The data supporting the findings of this study are available from the corresponding author on request.

## Acknowledgments

We thank the Bloomington Stock Center, Giorgos Pyrowolakis (Centre for Integrative Biological Signalling Studies, University

of Freiburg, Freiburg, Germany), Stefan Luschnig (University of Münster, Münster, Germany), and Robert Ward (Case Western Reserve University, Cleveland, OH) for providing fly stocks and antibodies; Bénédicte Sanson (University of Cambridge, Cambridge, UK) for the access to unpublished results; the Biozentrum Imaging Core Facility for maintenance of microscopes and support.

The work in the laboratory of M. Affolter was supported by grants from the Swiss National Science Foundation (310030_192659/1) and by funds from the Kantons Basel-Stadt and Basel-Land. The work in the laboratory of C. Cabernard was supported by grants from the Swiss National Science Foundation (PP00P3_159318) and the National Institutes of Health (1R01GM126029).

The authors declare no competing financial interests.

Author contributions: Conceptualization: K. Lepeta, E. Caussinus, O. Kanca, D. Bieli, M. Affolter, C. Cabernard; Investigation: K. Lepeta, C. Roubinet, M. Bauer, M.A. Vigano, G. Aguilar, A. Ochoa-Espinosa; Writing–Review & Editing: K. Lepeta, C. Roubinet, M. Affolter; Funding acquisition: M. Affolter, C. Cabernard.

Submitted: 1 July 2021

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

# Supplemental material

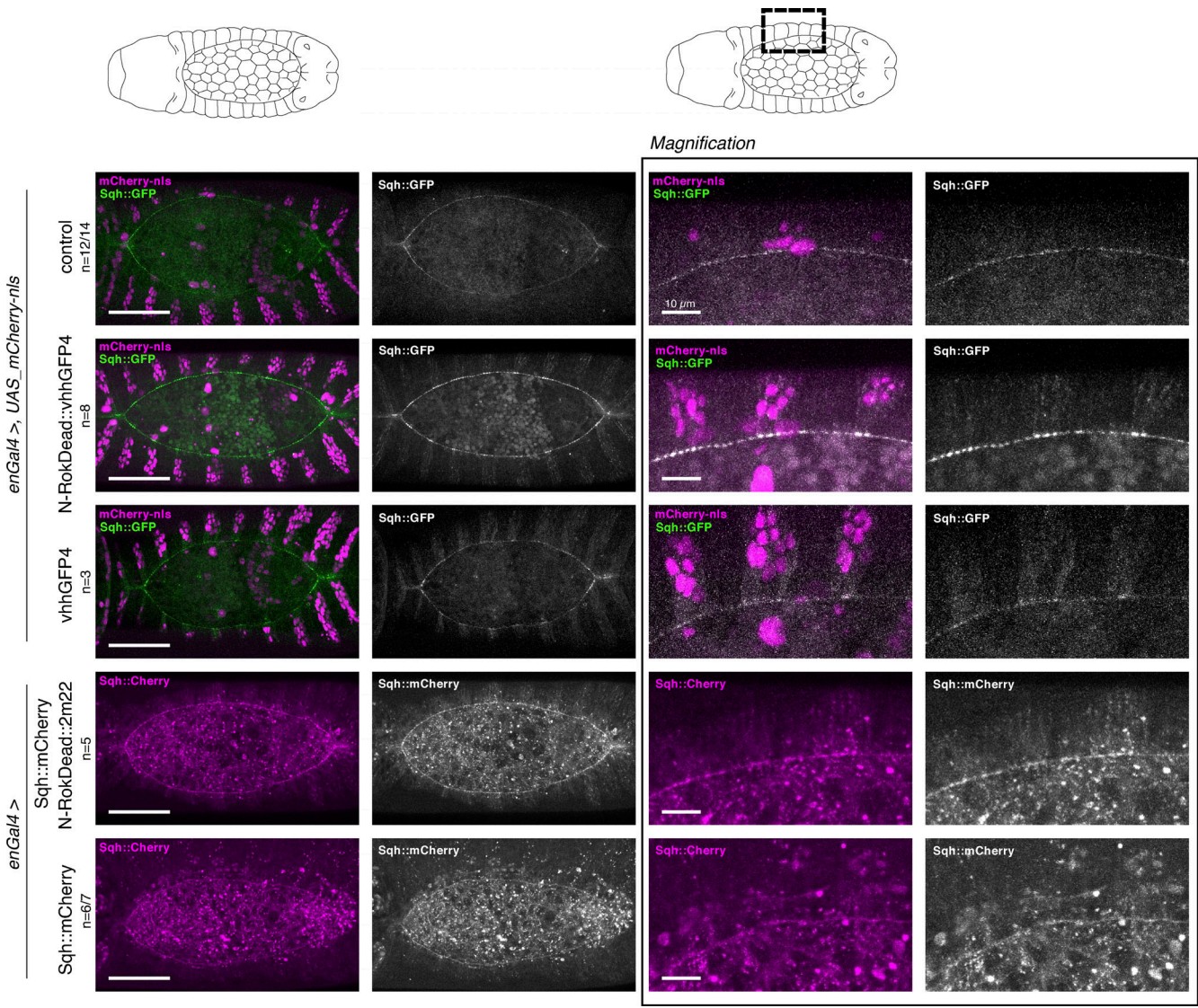

Figure S1.   **Binding of vhhGFP4 to GFP results in an increase of the fluorescence signal.** All panels show stills from live-imaging with dorsal views of either *sqh_Sqh::GFP* or *sqh_Sqh::mCherry* embryos at stage 13/14 (dorsal closure) expressing either N-RokDead variants or vhhGFP4 nanobody alone in the *en* domain (visualized by co-expression of mCherry-nls). Note the enhancement of fluorescent signal due to the binding of vhhGFP4 or 2m22 to GFP and to mCherry, respectively. The right panel shows magnification of the respective images for each of the embryo genotypes shown on the left. Please note that the "control" image is the same as on Figs. 3 a and 4 b to have a single reference image for all N-Rok variants used. For every expressed synthetic kinase, the number of considered embryos is indicated (*n*). Scale bars, 50 µm; in the zoomed panel, 10 µm.

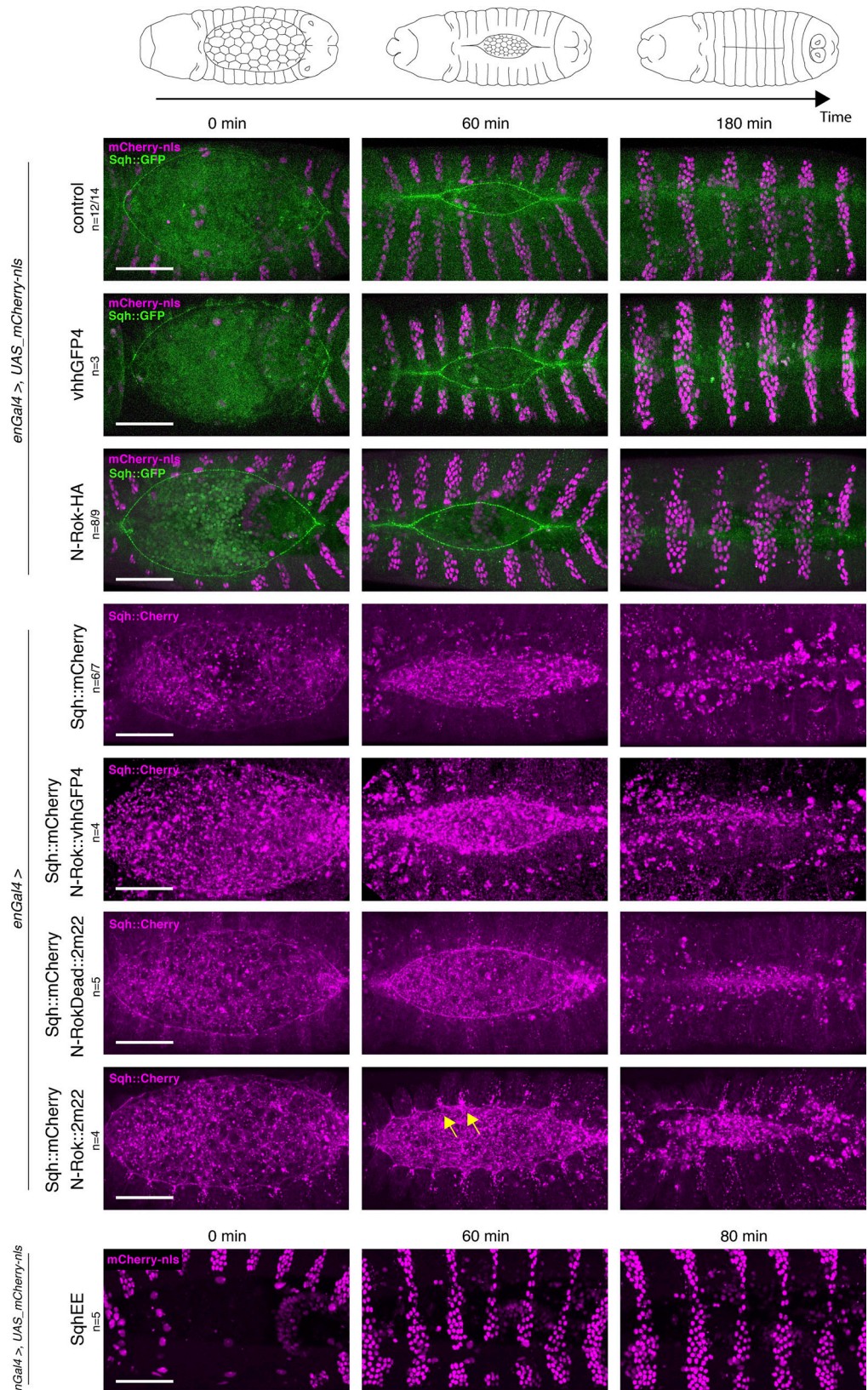

**Figure S2.** **All panels show stills from live-imaging with dorsal views of the developing *sqh_Sqh::GFP* or *sqh_Sqh::mCherry* embryos at stages 13/14–16 (dorsal closure) expressing variants of the synthetic kinase or the vhhGFP4 binder alone in the *en* domain (visualized by co-expression of mCherry-nls).** Bottom panel shows dorsal views of embryo expressing SqhE20E21 under the control of *enGal4* driver. Yellow arrows point to the Sqh::mCherry foci and actomyosin cable invaginations in N-Rok::2m22 panel. Please note that the "control" image is the same as on Figs. 3 a, 4 b, and Fig. S1, to have a single reference image for all N-Rok variants used. For every genotype, the number of considered embryos is indicated (*n*). Scale bars, 50 μm.

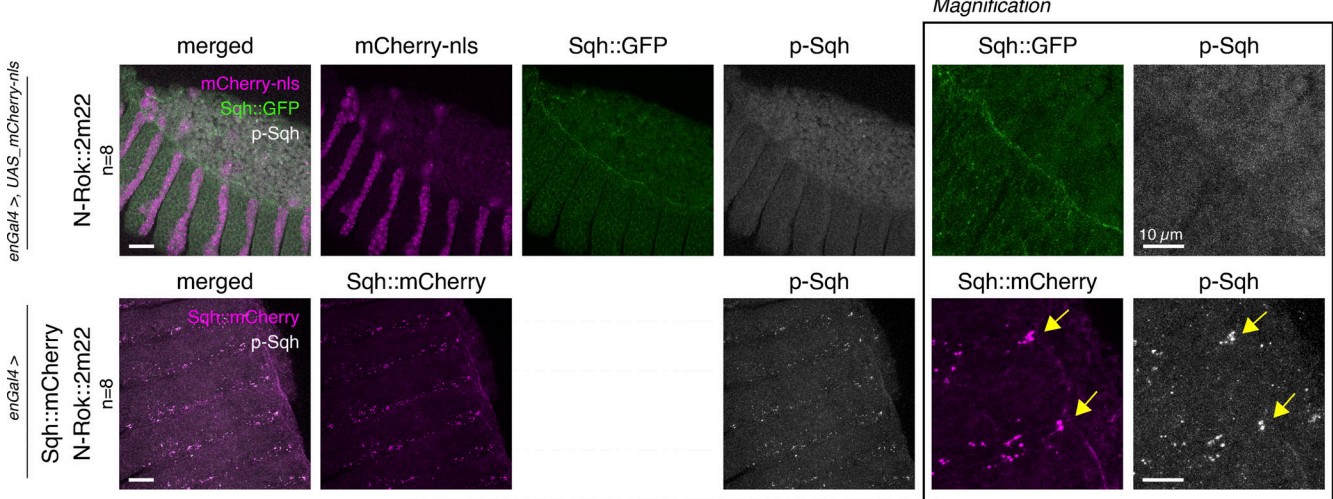

Figure S3.    **N-Rok::2m22 is a functional enzyme and efficiently phosphorylates Sqh::mCherry in vivo in a tissue-specific manner.** Panels show lateral views of fixed *sqh_Sqh::GFP* (top panel) or *sqh_Sqh::mCherry* (bottom panel) embryos at stage 13–14 (dorsal closure) expressing N-Rok::2m22 in the *en* domain (visualized by co-expression of mCherry-nls in the top panel). Embryos were stained with anti-phospho-Sqh/MRLC antibody. The panel on the right shows magnification of the respective Sqh and p-Sqh images for the presented embryos. Yellow arrows point to Sqh::mCherry and p-Sqh foci. The number of embryos analyzed is indicated (*n*). Scale bars, 20 µm; in the zoomed panel, 10 µm.

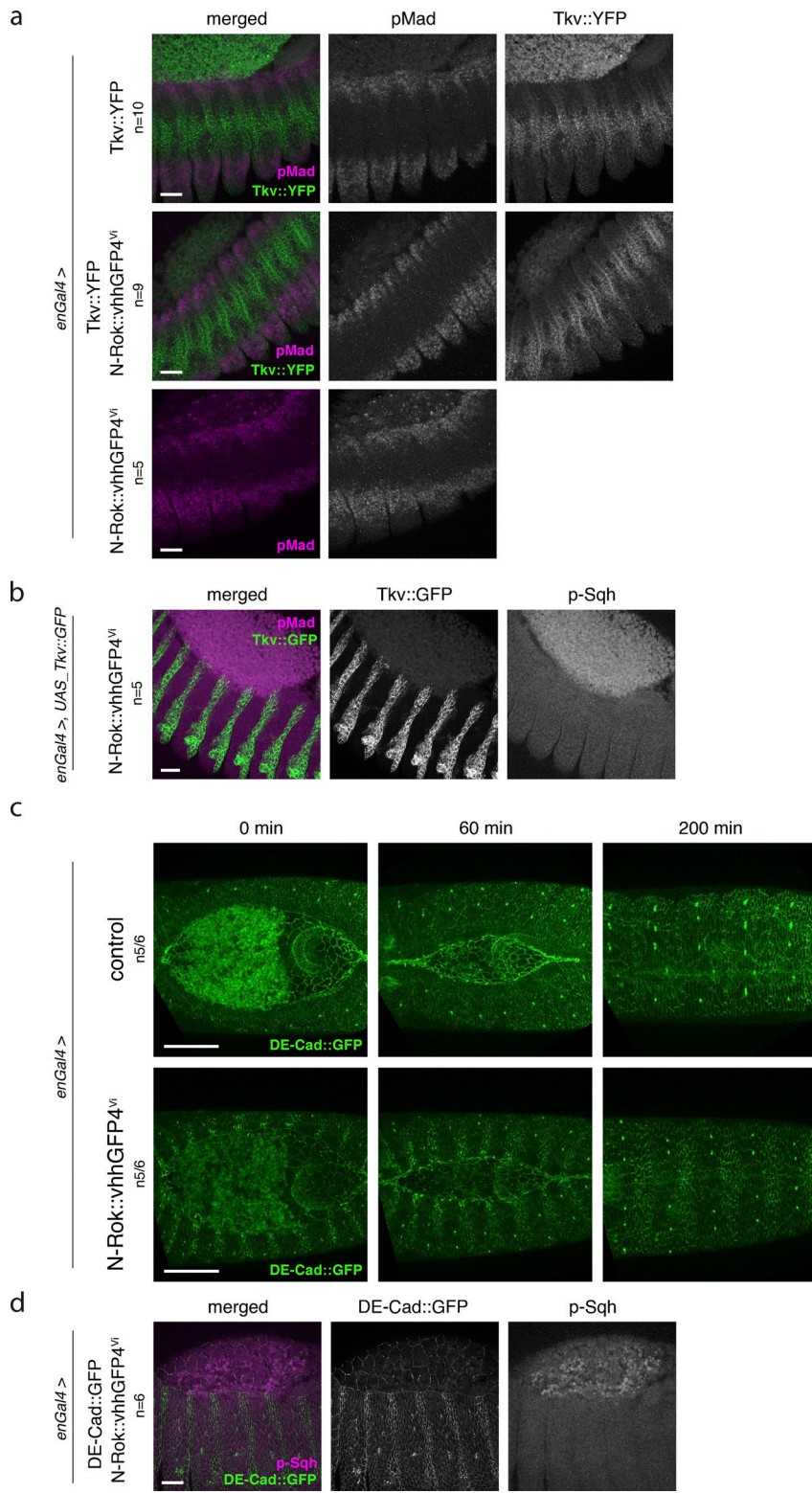

**Figure S4. Expression of N-Rok::vhhGFP4$^{Vi}$ in *Tkv::YFP* embryos has no effect on pMad levels. (a)** Panels show lateral views of fixed *Tkv::YFP* embryos at stage 13/14 (dorsal closure) expressing N-Rok::vhhGFP4$^{Vi}$ in the *en* domain (visualized by co-expression of mCherry-nls). Embryos were stained with anti-phospho-Mad antibody. **(b)** Panels show lateral views of fixed embryos at stage 13/14 (dorsal closure) expressing Tkv::GFP and N-Rok::vhhGFP4$^{Vi}$ in the *en* domain. Embryos were stained with anti-phospho-Sqh/MRLC antibody. **(c)** Panels show stills from live-imaging with dorsal views of developing *DE-Cad::GFP* embryos at stages 13/14–16 (dorsal closure) expressing N-Rok::vhhGFP4$^{Vi}$ in the *en* domain (visualized by co-expression of mCherry-nls). **(d)** Panels show lateral views of fixed *DE-Cad::GFP* embryo at stage 13/14. Embryos were stained with anti-phospho-Sqh/MRLC antibody. Note the enhancement of fluorescent signal due to the binding of vhhGFP4 to *DE-Cad::GFP* in the *en* domain. The number of considered embryos is indicated (*n*). Scale bars, 20 μm in a, b, and d; 50 μm in c.

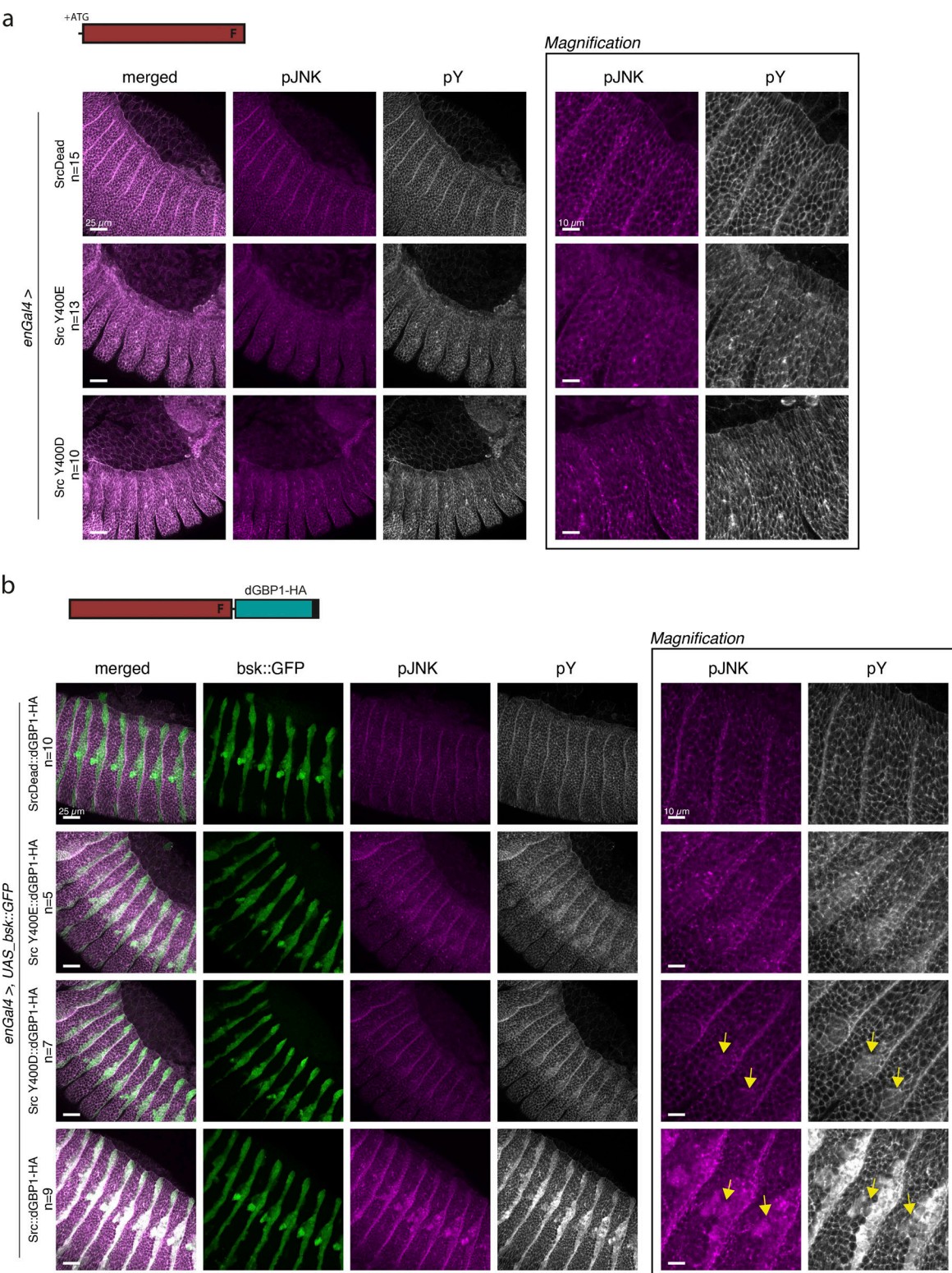

**Figure S5. Engineered Src::dGBP1-HA is a functional enzyme and efficiently phosphorylates Bsk::GFP in vivo while being innocuous in the absence of the GFP target. (a)** Panels show lateral views of fixed *Drosophila* embryos at stage 13–14 (dorsal closure) expressing synthetic Src kinase variants without the nanobody in the *engrailed* domain. Embryos were stained with anti-phospho-JNK antibody that recognizes Bsk, the *Drosophila* JNK ortholog and an anti-phosphotyrosine antibody (pY), that detects tyrosine phosphorylated proteins in all species. **(b)** Panels show lateral views of fixed *Drosophila* embryos at stage 13–14 (dorsal closure) expressing Bsk::GFP and synthetic Src kinase variants with C-terminal dGBP1 expressed in the *engrailed* domain. Embryos were stained as in panel a. The right panel in both a and b shows magnification of the respective pJNK and pY images for each of the embryo genotypes shown on the left. Arrows point to the areas of increased pJNK and pY signal in the *engrailed* domain where the active kinase is expressed. For every expressed synthetic kinase, the number of considered embryos is indicated (*n*). Scale bar, 25 µm; in the zoomed panel, 10 µm.

Video 1.   **Expression of synthetic Rok in *Sqh::GFP* embryos leads to abnormal dorsal closure.** Dorsal closure was used as a model to assess myosin II activation by means of Sqh::GFP phosphorylation with synthetic Rok. In *sqh_Sqh::GFP* embryos, expression of N-Rok::vhhGFP4 leads to abnormal dorsal closure due to formation of Sqh::GFP foci and additional cable structures. All panels show live-imaging of dorsal views of the developing *sqh_Sqh::GFP* embryos from early dorsal closure stage until late stages of embryogenesis; embryos express variants of the synthetic kinase in the *engrailed* domain (visualized by co-expression of mCherry-nls). Control (top left): *sqh_Sqh::GFP enGal4 UAS_mCherry-nls/+.* Synthetic Rok (top right): *sqh_Sqh::GFP enGal4 UAS_mCherry-nls/+; UAS_N-Rok::vhhGFP4$^{ZH-86Fb}$/+.* Inactive synthetic Rok (bottom left): *sqh_Sqh::GFP enGal4 UAS_mCherry-nls/+; UAS_N-RokDead::vhhGFP4$^{ZH-86Fb}$/+.* Synthetic Rok without a GFP-fusion target (bottom right): *enGal4 UAS_mCherry-nls/+; UAS_N-Rok::vhhGFP4$^{ZH-86Fb}$/+.* Imaging at 10 min intervals, 40× objective.

Video 2.   **Further evaluation of engineered Rok kinase.** All panels show live-imaging of dorsal views of the developing *sqh_Sqh::GFP* embryos from early dorsal closure stage until late stages of embryogenesis. Embryos express variants of the synthetic kinase in the *engrailed* domain (visualized by co-expression of mCherry-nls). Control (top left): *sqh_Sqh::GFP enGal4 UAS_mCherry-nls/.* Rok without the fused nanobody (top right): *sqh_Sqh::GFP enGal4 UAS_mCherry-nls/+; UAS_N-Rok-HA$^{ZH-86Fb}$/+.* vhhGFP4 nanobody control (bottom left): *sqh_Sqh::GFP enGal4 UAS_mCherry-nls/+; UAS_vhhGFP4/+.* SqhE20E21 (bottom right): *enGal4 UAS_mCherry-nls/+; UAS_ SqhE20E21.* Please note that the "control" video is the same as in Videos 1 and 5, to have a single reference movie for all N-Rok variants used. Imaging at 10 min intervals, 40× objective.

Video 3.   **Nanobody-based engineered kinases are modular tools.** Dorsal closure was used as a model to assess myosin II activation by means of Sqh::mCherry phosphorylation with synthetic Rok. In *sqh_Sqh::mCherry* embryos, expression of N-Rok::2m22 leads to abnormal dorsal closure due to formation of Sqh::mCherry foci and additional cable structures. All panels show live-imaging of dorsal views of the developing *Drosophila* embryos from early dorsal closure stage until late stages of embryogenesis. Embryos express Sqh::mCherry and variants of the synthetic kinase in the *engrailed* domain. Control (top left): *enGal4 sqh_Sqh::mCherry/+.* N-Rok::2m22 (top right): *enGal4 sqh_Sqh::mCherry/+; UAS_N-Rok::2m22$^{ZH-86Fb}$/+.* Inactive N-Rok::2m22 (bottom left): *enGal4 sqh_Sqh::mCherry/+; UAS_N-RokDead::2m22$^{ZH-86Fb}$/+.* N-Rok::vhhGFP4 control (bottom right): *enGal4 sqh_Sqh::mCherry/+; UAS_N-Rok::vhhGFP4$^{ZH-86Fb}$/+.* Imaging at 10 min intervals, 40× objective.

Video 4.   **Comparison of engineered Rok with previously available tools for direct activation of myosin II.** All panels show live-imaging of dorsal views of the developing *sqh_Sqh::GFP* embryos during dorsal closure stage, embryos express variants of the synthetic kinase in the *engrailed* domain (visualized by co-expression of mCherry-nls). For this set of experiments, embryos were mounted via gluing to the coverslip and the *sqh_Sqh::GFP* embryos expressing *N-Rok::vhhGFP4$^{ZH-86Fb}$* show more severe dorsal open phenotypes than embryos imaged on a glass-bottom dish (as shown in Videos 1, 2, and 3). Synthetic Rok (top left): *sqh_Sqh::GFP enGal4 UAS_mCherry-nls/+; UAS_N-Rok::vhhGFP4$^{ZH-86Fb}$/+.* Rok-cat (top right): *sqh_Sqh::GFP enGal4 UAS_mCherry-nls/+; UAS_Rok-catT2A/+.* Rok-CAT (bottom left): *sqh_Sqh::GFP enGal4 UAS_mCherry-nls/+; UAS_Rok-cat3.1/+.* ctMLCK (bottom right): *sqh_Sqh::GFP enGal4 UAS_mCherry-nls/+; UAS_ctMLCK/+.* Imaging at 5 min intervals, 40× objective.

Video 5.   **N-Rok::vhhGFP4$^{Vi}$ and N-Rok::dGBP1 are optimized variants of synthetic N-Rok.** In *sqh_Sqh::GFP* embryos, expression of N-Rok::vhhGFP4$^{Vi}$ or N-Rok::dGBP1 leads to abnormal dorsal closure due to formation of Sqh::GFP foci and additional actin cable structures. *N-Rok::vhhGFP4$^{Vi}$* allows to express a synthetic kinase whose concentration is both high enough to phosphorylate *Sqh::GFP*, and low enough to prevent deleterious effects in flies deprived of a GFP target. The substitution of the anti-GFP nanobody (vhhGFP4) with the destabilized variant (dGBP1) greatly improved the tool in being similarly efficient but non-toxic. All panels show live-imaging of dorsal views of the developing *Drosophila* embryos from early dorsal closure stage until late stages of embryogenesis, embryos express Sqh::GFP and variants of the synthetic kinase in the *engrailed* domain (visualized by co-expression of mCherry-nls). Control (top left): *sqh_Sqh::GFP enGal4 UAS_mCherry-nls/+.* Rok with the destabilized nanobody (dGBP1) (top right): *sqh_Sqh::GFP enGal4 UAS_mCherry-nls/+; UAS_N-Rok::dGBP1$^{ZH-86Fb}$/+.* N-Rok::vhhGFP4$^{Vi}$ (bottom left): *sqh_Sqh::GFP enGal4 UAS_mCherry-nls/UAS_N-Rok::vhhGFP4$^{Vi}$.* Rok with the destabilized nanobody without a GFP-fusion target (bottom right): *enGal4 UAS_mCherry-nls/+; UAS_N-Rok::dGBP1$^{ZH-86Fb}$/+.* Please note, that the "control" video is the same as for Videos 1 and 2, to have a single reference movie for all N-Rok variants used. Imaging at 10 min intervals, 40× objective.

Video 6.   **Evaluation of the specificity of engineered Rok kinase. Dorsal closure was used as a model to test whether engineered Rok can activate other GFP-fusion proteins, which are not a direct target of Rok, and cause any obvious developmental defects.** Please note an enhanced GFP signal in the *engrailed* domain, suggesting N-Rok::vhhGFP4$^{Vi}$ binding to GFP. Control (left): *cadherin (shg)::GFP/+.* Synthetic Rok (right): *cadherin::GFP/UAS_N-Rok::vhhGFP4$^{Vi}$.* Imaging at 10 min intervals, 40× objective.

Video 7.   **The tracheal system of *Drosophila* embryo develops abnormally upon excessive myosin II phosphorylation by engineered Rok kinase.** Time-lapse imaging of the development of the tracheal system with dorso-lateral views of *sqh_Sqh::GFP* embryos from germband retraction stage until late stages of embryogenesis. Expression of all three active N-Rok variants (N-Rok::vhhGFP4$^{ZH-86Fb}$, N-Rok::vhhGFP4$^{Vi}$ and N-Rok::dGBP1$^{ZH-86Fb}$) resulted in the formation of large Sqh::GFP foci and clustering of groups of tracheal cells, leading to a dramatically disorganized tracheal architecture. Variants of the synthetic kinase were expressed in the *breathless* domain (visualized by co-expression of mCherry-nls). Top panel: control (left): *sqh_Sqh::GFP btlGal4 UAS_mCherry-nls/+.* Synthetic Rok (middle): *sqh_Sqh::GFP btlGal4 UAS_mCherry-nls/+; UAS_N-Rok::vhhGFP4$^{ZH-86Fb}$/+.* Synthetic Rok with the destabilized nanobody (right): *sqh_Sqh::GFP btlGal4 UAS_mCherry-nls/+; UAS_N-Rok::dGBP1$^{ZH-86Fb}$/+.* Bottom panel: inactive synthetic Rok (left): *sqh_Sqh::GFP btlGal4 UAS_mCherry-nls/+; UAS_N-RokDead::vhhGFP4$^{ZH-86Fb}$/+.* Synthetic N-Rok::vhhGFP4$^{Vi}$ (left): *sqh_Sqh::GFP btlGal4 UAS_mCherry-nls/UAS_*N-Rok::vhhGFP4$^{Vi}$. Imaging at 10 min intervals, 40× objective, Leica SP5.

Video 8.   **Effect of constitutive activation of actomyosin with N-Rok::dGBP1 in dorsal branches on filopodia formation and cell migration.** Live-imaging with dorsolateral views of dorsal branches of *sqh_Sqh::GFP kniGal4* control embryos and *sqh_Sqh::GFP kniGal4 Rok::dGBP1$^{ZH-86Fb}$* embryos; LifeAct-mRuby (magenta) was used to visualize pools of F-actin. In the control conditions, tip cells of dorsal branches meet with the contralateral partner branches to form a stereotypic and interconnected branched network. Upon synthetic Rok expression, aberrant contact of two neighboring branches and the detachment of the tip cell from the stalk cells can be observed. Control (left): *sqh_Sqh::GFP UAS_ LifeAct-mRuby/kniGal4.* Synthetic Rok (right): *sqh_Sqh::GFP UAS_LifeAct-mRuby/kniGal4; UAS_Rok::dGBP1$^{ZH-86Fb}$/+.* Top panel shows merged colors, bottom panel shows single channel with LifeAct-mRuby. Imaging at 4 min intervals, 40× objective.

Video 9.   **Cell intercalation in dorsal branches.** Live-imaging with dorsolateral views of dorsal branches of *sqh_Sqh::GFP kniGal4* embryos; α-Catenin (alpha-cat::mCherry) was used to visualize cell junctions. Top panel shows merged colors, bottom panel shows single channel with alpha-cat::mCherry. Imaging at 5 min intervals, 40× objective.

Video 10.   **Effect of constitutive activation of actomyosin with N-Rok::dGBP1 in the *kni* domain on cell intercalation.** Live-imaging with dorsolateral views of dorsal branches of *sqh_Sqh::GFP kniGal4 Rok::dGBP1$^{ZH-86Fb}$* embryos; α-Catenin (alpha-cat::mCherry) was used to visualize cell junctions. Sqh::GFP clusters are visible in elongating branches, accumulating mostly at intercellular AJs. Top panel shows merged colors, bottom panel shows single channel with alpha-cat::mCherry. Imaging at 5 min intervals, 40× objective.

**Provided online are Table S1, Table S2, Table S3, and Data S1. Table S1 supports Fig. 3 and shows the viability and/or adult fly phenotypes induced by the expression of different synthetic kinase constructs in the absence of the GFP-target. Table S2 shows the summary of protein binders used in this study. Table S3 contains detailed information on the reagents, genetically modified organisms, and cell lines used. Data S1 is a multi-FASTA file corresponding to all the inserts used in this work.**

