## [Peer Review File · The Journal of Cell Biology]

Engineered kinases as a tool for phosphorylation of selected targets in vivo

Katarzyna Lepeta, Chantal Roubinet, Milena Bauer, Maria Alessandra Vigano, Gustavo Aguilar, Oguz Kanca, Amanda Ochoa-Espinosa, Dimitri Bieli, Clemens Cabernard, Emmanuel Caussin, and Markus Affolter

Corresponding Author(s): Katarzyna Lepeta, Biozentrum, University of Basel

Review Timeline:

Submission Date:	2021-07-01
Editorial Decision:	2021-08-12
Revision Received:	2022-05-19
Editorial Decision:	2022-06-15
Revision Received:	2022-07-26

Monitoring Editor: Kenneth Yamada

Scientific Editor: Andrea Marat

Transaction Report:

DOI: <https://doi.org/10.1083/jcb.202106179>

August 12, 2021

Re: JCB manuscript #202106179

Dr. Markus Affolter
Biozentrum
Klingelbergstrasse 50/70
Basel

Dear Drs. Affolter and Caussin,

Thank you for submitting your manuscript entitled "In vivo regulation of fluorescent fusion proteins by engineered kinases" to the Journal of Cell Biology at a Tools manuscript. The manuscript has now been assessed by three expert reviewers, whose reports are appended below. As you can see from the reviews provided by three leaders in various overlapping research areas spanning this paper, there was potential interest in the conclusions, but there was also one major reservation about the general applicability of this approach that would need to be resolved. After an assessment of the reviewer feedback, which varied in their level of enthusiasm for this study, our editorial decision is to invite you to submit a revision if you can address the reviewers' key concerns, as outlined here.

Two of the expert reviewers raised the important question of whether this approach would be of general applicability - which is a key requirement for a JCB Tools manuscript. As one reviewer notes, even though it need not be in detail, there would need to be direct evidence for practical applicability to a second kinase besides only the Rok kinase. We strongly support this concern raised by two of the reviewers. In addition, these expert reviewers have each listed a variety of additional points that appear to us to be important to address while avoiding unnecessarily expanding the scope of this study (except for the needed demonstration of the generality of the approach). For example, mass spectrometry would not need to be added. Since generating a second application of this potentially exciting approach might require considerable amounts of time beyond the JCB resubmission deadline, we will be happy to consider a considerably later resubmission if the methodology in this excellent manuscript remains sufficiently novel in the eyes of the original reviewers

GENERAL GUIDELINES:

Text limits: Character count for an Tools is < 40,000, not including spaces. Count includes title page, abstract, introduction, results, discussion, acknowledgments, and figure legends. Count does not include materials and methods, references, tables, or supplemental legends.

Figures: Tools may have up to 10 main text figures. Figures must be prepared according to the policies outlined in our Instructions to Authors, under Data Presentation, <https://jcb.rupress.org/site/misc/ifora.xhtml>. All figures in accepted manuscripts will be screened prior to publication.

*****IMPORTANT:** It is JCB policy that if requested, original data images must be made available. Failure to provide original images upon request will result in unavoidable delays in publication. Please ensure that you have access to all original microscopy and blot data images before submitting your revision. ***

Supplemental information: There are strict limits on the allowable amount of supplemental data. Tools may have up to 5 supplemental figures. Up to 10 supplemental videos or flash animations are allowed. A summary of all supplemental material should appear at the end of the Materials and methods section.

As you may know, the typical timeframe for revisions is three to four months. However, we at JCB realize that the implementation of social distancing measures that limit spread of COVID-19 also pose challenges to scientific researchers. Therefore, JCB has waived the revision time limit. We recommend that you reach out to the editors to decide on an appropriate time frame for resubmission. Please note that papers are generally considered through only one revision cycle, so any revised manuscript will likely be either accepted or rejected.

Thank you for this interesting contribution to Journal of Cell Biology. You can contact us at the journal office with any questions, cellbio@rockefeller.edu or call (212) 327-8588.

Sincerely,

Kenneth Yamada, MD, PhD
Editor

Andrea L. Marat, PhD
Senior Scientific Editor

Journal of Cell Biology

Reviewer #1 (Comments to the Authors (Required)):

Lepeta and colleagues developed a new method, using nanobody based protein binding, to specifically phosphorylate a GFP fusion protein by one defined kinase in vivo in a tissue specific manner. This method is interesting and significant as specific manipulation of protein kinase - substrate interactions in vivo remains challenging.

The authors apply this method to Rho kinase (Rok), a very prominently regulated kinase that controls nonmuscle myosin activity by phosphorylating its regulatory light chain Sqh in various cell types at multiple steps during embryonic development.

Specifically, the authors use an active N-terminal fragment of Rok (N-Rok) C-terminally fused to a GFP-nanobody or Cherry-DARPin that bind to Sqh-GFP or Sqh-Cherry, respectively, when expressed in the same cell.

The authors test their method by recruiting phosphorylated Sqh-GFP to the cortex of S2 cells and by creating large ectopic foci of phosphorylated Sqh-GFP in 'engrailed' stripes of epithelial cells in fly embryos. The formation of these excessive p-Sqh signal depends on both Sqh-GFP and N-Rok nanobody fusion.

As expected, these ectopic pSqh accumulations have important consequences. The supracellular actin cable is severely disrupted and dorsal closure is strongly defective, which is beautifully documented by 2 colour live imaging. This phenotype is stronger than phenotypes caused by various other previously existing tools to create hyperactivated Rok that were also tested by the authors.

A not unexpected limitation is that N-Rok-nanobody is not only specific to activate sqh-GFP but also causes lethality when expressed highly in wild type embryos or larvae with various GAL4 drivers. To overcome this limitation, the authors generated a low expressing UAS-N-Rok-nanobody fly line as well as we as a fusion with a destabilised nanobody version resulting in lower expression levels of the N-Rok-nanobody and hence lower off target effects. Importantly, both of these lines show similar ectopic pSqh production and dorsal closure phenotypes when co-expressed with Sqh-GFP in engrailed stripes, but their ectopic effects are severely reduced making them very specific tools.

A second tissue investigated by the authors was the tracheal system. The different N-Rok nanobody versions can all generate the ectopic P-Sqh clusters in the tracheal cells and entirely block tracheal migration if expressed early with btl-GAL4, showing that regulated myoII activity is essential for early tracheal cell placode invagination and migration.

To specifically study the effects in tracheal cell intercalation, the authors expressed the destabilised N-Rok nanobody version with kni-GAL4 specifically in the tracheal branches.

This led to ectopic pSqh-GFP and to some detachment of tip from stalk cells, however surprisingly most but not all cell-cell intercalations and autocellular junctions formed normally.

I find the idea of generating a tool that can ectopically generate hyper-phosphorylated and thus active myoII in vivo a very cool idea that will likely have many other applications. This is certainly worth publishing, as all the data presented are of high quantity and all necessary controls are included.

However, before advertising it as a general tool to generate hyper-phosphorylated substrates a second example might be necessary or text should be toned down as sqh-GFP might have been the lucky shot.

I am also not entirely convinced why the absence of an ectopic effect in tracheal cell in intercalation is stressed to strongly in text and abstract. As the authors appreciate, there is an effect, even if weak. This one might be stronger if the stronger N-Rok-nanobody fusion would have been used.

1. Without reading the abstract, the manuscript title is hard to understand. Maybe replace 'regulation' by 'phosphorylation'?

2. The abstract sounds like the Rho kinase-nanobody fusion is not very effective in hyperactivating Sqh-GFP, as no gain of function phenotypes are mentioned despite plenty are shown in the paper and only the surprising negative results on tracheal intercalation are highlighted, which alone do not mean much.

3. Can phosphorylation levels of Sqh be quantified better with and without kinase using western blots from S2 cell extracts, similar to PMID 20920606? This would allow optimising the design of the fusion protein. The authors report death of many cells preventing western blotting, however, why waiting for 40hrs before doing the analysis? This might not be needed for the

biological conclusions of the paper, but might be insightful for effective future designs of kinase substrate pairs.

4. As the authors suggest this method could be of general use for the application of other kinases/substrates. However, I wonder how easy the 2 fusion proteins can interact in order to phosphorylate the key amino acid(s), as the GFP-nanobody binding will be rigid and largely irreversible. Have the authors tested differently long flexible linkers between the compact nanobody and the kinase to vary the potential accessibility of the phosphorylation site on the GFP-target? A larger linker may strongly increase the activity on the substrate. This could be tested at least in S2 cells or in vivo (see point 3).

5. Can a similar method be used to block phosphorylation of a target, e.g. by recruiting a phosphatase fused to the GFP nanobody? This could even be more powerful than creating ectopic phosphorylation as it may create loss of function alleles. As this is beyond the current scope of the paper it could be added to the discussion.

Minor.

1. Should the scheme in Figure 1A not better show a kinase fused at its C-term to the GFP nanobody as the case for N-Rok in Figure 1D? Currently, I find the scheme a bit misleading as it does not show the design principle that was used. This is also related to potential sterical hindrance to reach the target amino acid, see my point 4.

2. Formally the authors do not directly show that 'N-Rok::vhhGFP4ZH-86Fb is able to phosphorylate Sqh::GFP on Ser21' as stated on page 10, only that it does phosphorylate Sqh::GFP. The site was not tested, neither by a mutation nor by mass spec.

3. First para of destabilized GFP binder: 'with no signs of aggregation in cells33 (Fig. 4a).' should read Figure 4d?

4. Why are the mCherry-nls levels so variable in the different embryos shown in Figure 5b? These differences are not obvious in movie 7.

5. FBrf0212807 (Hatan 2011) should be included as normal citation in the resource table as the other references instead of being listed as Flybase reference.

6. Last results page 'revealed that in the vast majority of the imaged embryos (add)' add numbers?? Why are the Vermiform data not shown?

'Supplementary Movie 9_Intercalation_control and Supplementary Movie 10_Intercalation_Rok_dGBP1)', fix formatting.

7. Why are unpublished data discussed in the discussion?

Reviewer #2 (Comments to the Authors (Required)):

In this paper, Lepeta et al show that a kinase (ROK) can be specifically directed to its substrate (Sqh) and phosphorylate it in vivo. The work is performed in *Drosophila* but there is no reason why it would not be applicable to other species. They show that an overexpressed ROK-Nanobody fusion (N-Rok::vhhGFP4) activates Myosin2 in the presence of a GFP-Sqh fusion protein (Sqh::GFP). Using several controls, they show that the enzyme is primarily targeted to its substrate via the Nb-GFP interaction. They then reduce the expression level of the Rok-Nb fusion protein to lessen off target effects, thus making it a more physiological reagent. The method is also shown to work with another pair of binders (Sqh::mCherry). The authors then demonstrate the effectiveness of their approach in the context of dorsal closure and use it to investigate the role of myosin phosphorylation in cell intercalation during tracheal development.

The demonstration that enzyme substrate interactions can be reconstituted in vivo is an important advance, as it paves the way toward engineering signalling networks in vivo. For this reason, this paper is a good contender for publication in JCB. Nevertheless, there are shortcomings and limitations, which must be addressed or recognised before publication.

As the authors acknowledge, there are already several approaches available to activate ROK in vivo (including optogenetics). The authors argue that their approach is more effective and has milder side effects. Both statements need to be qualified/better justified. This, in my view, is the most important improvement that the manuscript needs.

The comparison with other means of regulating ROK depends on the level expression of the reagent used. While the authors show that their approach is more effective with a given gal4 driver, their evidence depends on the choice of UAS-transgene. For example, the effectiveness of NSimb::vhhGFP4 could be improved by boosting expression, which is possible by changing the 5' or 3' UTR of the UAS vector (PMID: 22493255).

The authors go through great effort to limit the off target activity of N-Rok::vhhGFP4 by reducing expression and also by using a clever trick based on destabilised nanobody that is stabilised in the presence of its target. This is to be commended. However, the off-target assay is rather crude, as it relies on lethality. A better assay would be to measure global phosphorylation patterns by mass spec.

In my view, the best way to assess the specificity and effectiveness of an engineered kinase would be to compare phosphorylation of the target relative to that of non-specific proteins. This could be achieved by measuring the extent of p-Sqh relative to the increase in other phosphorylation events in the rest of the proteome. This could be achieved by mass spec and the result would greatly improve the quality of the manuscript, especially if the same was done with an existing means of activating phosphorylation of Sqh. I understand however that access to a suitable mass spec facility would be required for such an experiment.

Minor points

It is important to point out that this remains a GOF approach. It will be interesting to know if a reproducible response will be induced by expressing the engineered kinase at physiological level, in the absence of the endogenous gene. Could the authors comment?

Although the approach described has universal potential, the tools described are specific to ROK and the effort devoted to reducing off target effects are only relevant to this protein. This must be recognised.

I am somewhat surprised by the observation that N-Rok::HAZH-86Fb without the fused nanobody (n=6) had no effect on the p-Sqh pattern. This protein remains dominant active, which phosphorylate proteins albeit non-specifically. I would expect phosphorylation of its cognate target to be increased (although to a much lesser extent than with the nanobody target system).

Reviewer #3 (Comments to the Authors (Required)):

Review of "In vivo regulation of fluorescent fusion proteins by engineered kinases" by Lepeta et al.

Regulation of cellular processes by phosphorylation/dephosphorylation reactions is extraordinarily common and extraordinarily important. While an enormous variety of tools are currently available for manipulation of such reactions, a large hole remains: regulation of phosphorylation of specific kinase substrates. At present, the go-to approach is to replace the relevant residues with either nonphosphorylatable amino acids (typically A) or phosphomimetic amino acids (D or E). The problems with this approach are that 1) it is not time resolved; 2) there is no guarantee that the mutations don't result in some other deficit in protein function and 3) for phosphomimetics, there is no guarantee that they actually mimic phosphorylation.

In this study, the authors describe an approach for ectopic upregulation of phosphorylation of specific proteins by targeting a constitutively active kinase (in this case, Rok) to a fluorescent protein-fused version of its substrate (in this case the regulatory light chain of myosin-2, aka RLC) via a nanobody. The authors detail a long list of experiments in which they vet this approach in *Drosophila* embryos and provide strong in vivo evidence that they can indeed specifically upregulate the phosphorylation of RLC with the most specific results being obtained via the use of a nanobody that is unstable in the absence of its fluorescent protein target. They show that ectopic RLC phosphorylation disrupts dorsal closure, as expected based on previous studies and further show that ectopic RLC phosphorylation perturbs tracheal development in a manner that depended on the timing and location of induction.

The results presented in this study are promising in that the authors appear to have developed an approach that overcomes the second and third problems alluded to above (they can truly increase phosphorylation of RLC without concerns about changing protein structure except insofar as the addition of the fluorescent protein to the substrate is required). It might be argued that without better time resolution and reversibility this approach will be of limited utility but it seems clear that this is an important first step. Similarly, it might be argued that it would have been more useful to develop this approach for targeted dephosphorylation, but phosphatases are more complex than kinases, typically comprising three subunits for full function, so that too seems like too much to ask.

There are, however, other points that should be addressed. Perhaps most importantly, it should be determined whether this approach will be generally useful with other substrates and kinases. A single additional example would suffice, and it would not require the detailed phenotypic work up used for RLC, but as it stands, it is impossible to assess how broadly applicable this approach will be.

In addition, to both bolster the authors interpretation of their results and to support the idea that this approach could be used to counteract loss-of-function conditions, it would be helpful to show that ectopic RLC phosphorylation can alleviate Flw overexpression.

Minor comments:

The text of the results section is frequently digressive. The authors should consider trimming it down to better emphasize the important points.

The authors may want to check out Elife 2016 Dec 30:5:e20828 which clearly shows that phosphomimetic mutations of RLC do not faithfully mimic true phosphorylation events when it comes to myosin-2 motor activity. This finding supports the potential utility of the approach employed here, particularly for RLC.

Based on the reviewer's comments and after discussions with the Editors, we have made the following changes to the manuscript:

- 1) We have made an extensive new series of synthetic kinase constructs based on Src kinase and have tested their activity *in vivo*. We found that such constructs can indeed phosphorylate a selected target protein if the kinase carries a nanobody binder recognizing the targets; phosphorylation does not occur in the absence of the nanobody. We therefore shown now data for a second kinase (Figure 8, Figure 9, Supplementary Figure 5), and this was a major point raised by the reviewers. It was a large effort to generate and analyze these constructs, and we added three additional authors to the list since they contributed to this work.
- 2) We have changed the title of the manuscript.
- 3) We have made a number of smaller changes based on the reviewer's comments (outlined in detail below).
- 4) Supplementary Figure 3 (images of fly wing phenotype) was removed to accommodate new Src data (Supplementary Figure 5)
- 5) Colors in the Figure 7b were changed – magenta was used instead of red
- 6) For all figures showing embryo images/stills from videos the driver used and the genotype were added on the left side of the figure panels

We have uploaded a clean copy of our revised manuscript that does not show changes made as our "Manuscript" file. In addition, we have included a marked-up copy of our manuscript showing the changes we have made since the original submission.

Reviewer #1 (Comments to the Authors (Required)):

Lepeta and colleagues developed a new method, using nanobody based protein binding, to specifically phosphorylate a GFP fusion protein by one defined kinase *in vivo* in a tissue specific manner. This method is interesting and significant as specific manipulation of protein kinase - substrate interactions *in vivo* remains challenging.

The authors apply this method to Rho kinase (Rok), a very prominently regulated kinase that controls nonmuscle myosin activity by phosphorylating its regulatory light chain Sqh in various cell types at multiple steps during embryonic development. Specifically, the authors use an active N-terminal fragment of Rok (N-Rok) C-terminally fused to a GFP-nanobody or Cherry-DARPin that bind to Sqh-GFP or Sqh-Cherry, respectively, when expressed in the same cell. The authors test their method by recruiting phosphorylated Sqh-GFP to the cortex of S2 cells and by creating large ectopic foci of phosphorylated Sqh-GFP in 'engrailed' stripes of epithelial cells in fly embryos. The formation of these excessive p-Sqh signal depends on both Sqh-GFP and N-Rok nanobody fusion.

As expected, these ectopic pSqh accumulations have important consequences. The supracellular actin cable is severely disrupted and dorsal closure is strongly defective, which

is beautifully documented by 2 colour live imaging. This phenotype is stronger than phenotypes caused by various other previously existing tools to create hyperactivated Rok that were also tested by the authors.

A not unexpected limitation is that N-Rok-nanobody is not only specific to activate sqh-GFP but also causes lethality when expressed highly in wild type embryos or larvae with various GAL4 drivers. To overcome this limitation, the authors generated a low expressing UAS-N-Rok-nanobody fly line as well as we as a fusion with a destabilised nanobody version resulting in lower expression levels of the N-Rok-nanobody and hence lower off target effects. Importantly, both of these lines show similar ectopic pSqh production and dorsal closure phenotypes when co-expressed with Sqh-GFP in engrailed stripes, but their ectopic effects are severely reduced making them very specific tools.

A second tissue investigated by the authors was the tracheal system. The different N-Rok nanobody versions can all generate the ectopic P-Sqh clusters in the tracheal cells and entirely block tracheal migration if expressed early with btl-GAL4, showing that regulated myoII activity is essential for early tracheal cell placode invagination and migration. To specifically study the effects in tracheal cell intercalation, the authors expressed the destabilised N-Rok nanobody version with kni-GAL4 specifically in the tracheal branches. This led to ectopic pSqh-GFP and to some detachment of tip from stalk cells, however surprisingly most but not all cell-cell intercalations and autocellular junctions formed normally.

I find the idea of generating a tool that can ectopically generate hyper-phosphorylated and thus active myoII in vivo a very cool idea that will likely have many other applications. This is certainly worth publishing, as all the data presented are of high quality and all necessary controls are included.

However, before advertising it as a general tool to generate hyper-phosphorylated substrates a second example might be necessary or text should be toned down as sqh-GFP might have been the lucky shot.

We would like to thank the reviewer for the positive comments on the data shown in the manuscript. Since the referee wanted to see a second example of such an engineered kinase, we undertook an extensive, novel series of experiments involving the Src kinase, and we now present this data in the revised manuscript. We sincerely hope that the reviewer is encouraged and satisfied with the new data and supports publication of the revised manuscript.

As requested, we have made a second engineered kinase based on the activated catalytic domain of Src42A combined with protein binders. We used this tool to activate the Drosophila ortholog of JNK kinase – Basket (Bsk) and show that engineered Src42A with dGBP1 efficiently phosphorylates Bsk-GFP and that this effect is dependent on the presence of the nanobody; Src without a binder did not elicit increase in pJNK nor pY levels as shown by immunostaining. We also confirmed that engineered Src hyper-phosphorylated only the GFP-fusion and not the endogenous, untagged JNK substrate.

I am also not entirely convinced why the absence of an ectopic effect in tracheal cell in intercalation is stressed to strongly in text and abstract. As the authors appreciate, there is an

effect, even if weak. This one might be stronger if the stronger N-Rok-nanobody fusion would have been used.

We thank the reviewer for this comment. We have now changed the abstract to better explain our results on intercalation in dorsal branches of tracheal cells.

As known through previous work from our and other labs, tracheal cell intercalation occurs passively and does not require actomyosin contractility, in sharp contrast to the actomyosin-driven junctional rearrangements observed in many intercalation events in drosophila morphogenesis. Using synthetic N-Rok::dGBP1, we show that excessive phosphorylation of Sqh::GFP specifically in the kni expression domain did not result in a failure of stalk cell intercalation, despite evident ectopic clusters of tension in the areas most likely corresponding to cell junctions; stalk cell intercalation and autocellular junction formation were seen in most branches upon actomyosin activation. These results are in line with the notion that tip-cell pulling force is a key player in proper formation of the dorsal branch and that this force is able to overcome some counteracting excessive points of tension. We have toned down and shortened this section in the manuscript.

1. Without reading the abstract, the manuscript title is hard to understand. Maybe replace 'regulation' by 'phosphorylation'?

*We thank the reviewer for this valuable point. We have changed the title to:
“Engineered kinases as a novel tool for phosphorylation of selected targets in vivo”*

2. The abstract sounds like the Rho kinase-nanobody fusion is not very effective in hyperactivating Sqh-GFP, as no gain of function phenotypes are mentioned despite plenty are shown in the paper and only the surprising negative results on tracheal intercalation are highlighted, which alone do not mean much.

We have changed the abstract accordingly and added the novel data on the second engineered kinase. Please see also answer above on the ectopic effect in tracheal cell in intercalation.

3. Can phosphorylation levels of Sqh be quantified better with and without kinase using western blots from S2 cell extracts, similar to PMID 20920606? This would allow optimising the design of the fusion protein. The authors report death of many cells preventing western blotting, however, why waiting for 40hrs before doing the analysis? This might not be needed for the biological conclusions of the paper, but might would be insightful for effective future designs of kinase substrate pairs.

We agree with the referee that more cell culture data using western blotting would have been an alternative approach to design and optimize novel engineered kinases. Since we were not very successful with this approach after several trials (low transfection efficiency, problems with cell survival upon transfection with the active N-Rok, the fact that Sqh is strongly phosphorylated in mitotic cells, which increases the level of P-Sqh in the cells that do not

express the nanobody, etc.), we used the drosophila system which allows for both the targeted expression of the kinases and the in vivo biological context.

We have tested different time points after transfection to do the analysis, and 40 hours seemed the best compromise. When the immunostaining was done after shorter time (i.e., 30 hours) we could not detect a decent number of cells with cortical Sqh-GFP during the interphase despite the expression of the nanobody constructs (as confirmed by anti-HA staining).

4. As the authors suggest this method could be of general use for the application of other kinases/substrates. However, I wonder how easy the 2 fusion proteins can interact in order to phosphorylate the key amino acid(s), as the GFP-nanobody binding will be rigid and largely irreversible. Have the authors tested differently long flexible linkers between the compact nanobody and the kinase to vary the potential accessibility of the phosphorylation site on the GFP-target? A larger linker may strongly increase the activity on the substrate. This could be tested at least in S2 cells or in vivo (see point 3).

We agree with the reviewer that testing of different linkers would be very interesting. We did not start such analyses since we saw strong activation using the relatively short linker in the initial N-Rok-nanobody fusion. However, we indeed find differences in activity in the various Src constructs when fusing the nanobody either N- or C-terminally. We also observed differences when using other protein binders, such as the rather stable DARPins. So, there are certainly many parameters that can still be optimized, but this would go beyond this study. Furthermore, each kinase might need to be optimized with regard to its constitutive activation, as shown by the results obtained with the Src kinase and presented in the revised manuscript.

5. Can a similar method be used to block phosphorylation of a target, e.g., by recruiting a phosphatase fused to the GFP nanobody? This could even be more powerful than creating ectopic phosphorylation as it may create loss of function alleles. As this is beyond the current scope of the paper it could be added to the discussion.

This is a very interesting idea. Indeed, we have also thought about generating “synthetic” phosphatases along the same way described here. However, since many of the phosphatases act as heterodimers, we have not started such experiments as of now. As the reviewer states, such constructs might represent even more powerful tools, and we are now thinking again about initiating such experiments and have started to collect resources to do so.

Minor.

1. Should the scheme in Figure 1A not better show a kinase fused at its C-term to the GFP nanobody as the case for N-Rok in Figure 1D? Currently, I find the scheme a bit misleading as it does not show the design principle that was used. This is also related to potential sterical hindrance to reach the target amino acid, see my point 4.

We thank the reviewer for making us aware of this mistake. We have now changed the scheme in Figure 1a and Figure 4c such as to represent the real organization of the functional domains

in the transgenes we generated. In addition, we used “kinase domain” instead of “kinase” in the schemes shown in Figure 1a, Figure 1d and Figure 4c.

2. Formally the authors do not directly show that 'N-Rok::vhhGFP4ZH-86Fb is able to phosphorylate Sqh::GFP on Ser21' as stated on page 10, only that it does phosphorylate Sqh::GFP. The site was not tested, neither by a mutation nor by mass spec.

We used two p-Sqh abs and both recognize Ser21 phosphorylation:

1) Commercial ab “Phospho-Myosin Light Chain 2 (Ser19) Antibody detects endogenous levels of myosin light chain 2 (smooth muscle) only when phosphorylated at serine 19 (=Ser21 in drosophila).

2) Sqh1P ab from Zhan and Ward paper that describes it like this:

“we generated two site-specific antibodies against the phosphorylated forms of Sqh: one directed against the monophosphorylated form (with phospho-Ser21; referred to hereafter as Sqh1P), and the other directed against the diphosphorylated form (with phospho-Thr20 and phospho-Ser21; referred to as Sqh2P) To generate phosphorylation site-specific antibodies, two peptides were commercially synthesized corresponding to regions of the Sqh protein with phosphorylated derivatives at Ser21 or Thr20/Ser21. The sequences of the peptides used were Sqh1P: A-Q-R-A-T-S-N-V-F-A-M-C and Sqh2P: A-Q-R-A-T*-S*-N-V-F-A-M-C (* indicates phosphorylated amino acid).”*

3. First para of destabilized GFP binder: 'with no signs of aggregation in cells33 (Fig. 4a)!' should read Figure 4d?

We corrected the reference to the appropriate figure depicting the scheme of dGBP1 action (Fig. 4c instead of Figure 4a).

4. Why are the mCherry-nls levels so variable in the different embryos shown in Figure 5b? These differences are not obvious in movie 7.

We adjusted the signal intensity of magenta channel on the Figure 5b for N-Rok::vhhGFP4[Vi] and N-Rok::dGBP1 for the time 180 min so that it matches better the intensity of mCherry for the other Rok variants shown on this figure. The differences in the intensity of mCherry signal come from the varying depth of tracheal cells depending on when exactly the cells clustered due to the kinase activity, as well as on the exact angle/position of the embryo which changes slightly due to morphogenetic movements during the live-imaging. In the whole video therefore such intensity variability may not be so noticeable, while in a particular still for a given timepoint it may be more obvious.

5. FBrf0212807 (Hatan 2011) should be included as normal citation in the resource table as the other references instead of being listed as Flybase reference.

We have added the missing reference.

6. Last results page 'revealed that in the vast majority of the imaged embryos (add)' add numbers?? Why are the Vermiform data not show?

'Supplementary Movie 9_Intercalation_control and Supplementary Movie 10_Intercalation_Rok_dGBP1)', fix formatting.

*We have toned down the text on proper lumen formation (page 17 of the revised manuscript):
“Co-staining of the embryos with Vermiform, a marker for the tracheal lumen, revealed that in the majority of the imaged embryos, the lumen was formed both in the dorsal trunk (where no synthetic kinase was expressed), as well as in the kinase-expressing branches (data not shown).*

Due to limited space on the Figure we decided not to include this particular piece of data, but we are willing to show the vermiform data upon request.

We have now fixed the formatting.

7. Why are unpublished data discussed in the discussion?

We have added the missing reference replacing this statement (with which we wanted to stress in which laboratory the data was generated) with the review from our lab in which the data is shown (Kotini et al., 2019).

We would like to thank the reviewer again for the constructive criticism and overall appreciation of our work.

Reviewer #2 (Comments to the Authors (Required)):

In this paper, Lepeta et al show that a kinase (ROK) can be specifically directed to its substrate (Sqh) and phosphorylate it in vivo. The work is performed in *Drosophila* but there is no reason why it would not be applicable to other species. They show that an overexpressed ROK-Nanobody fusion (N-Rok::vhhGFP4) activates Myosin2 in the presence of a GFP-Sqh fusion protein (Sqh::GFP). Using several controls, they show that the enzyme is primarily targeted to its substrate via the Nb-GFP interaction. They then reduce the expression level of the Rok-Nb fusion protein to lessen off target effects, thus making it a more physiological reagent. The method is also shown to work with another pair of binders (Sqh::mCherry). The authors then demonstrate the effectiveness of their approach in the context of dorsal closure and use it to investigate the role of myosin phosphorylation in cell intercalation during tracheal development.

The demonstration that enzyme substrate interactions can be reconstituted in vivo is an important advance, as it paves the way toward engineering signalling networks in vivo. For this reason, this paper is a good contender for publication in JCB. Nevertheless, there are shortcomings and limitations, which must be addressed or recognised before publication.

As the authors acknowledge, there are already several approaches available to activate ROK in vivo (including optogenetics). The author argue that their approach is more effective and has milder side effects. Both statements need to be qualified/better justified. This, in my view, is the most important improvement that the manuscript needs.

We would like to thank the reviewers for the valuable suggestions. Since the other two reviewers asked for the direct evidence for practical applicability of our approach to a second kinase and this request was strongly supported by the editors, we undertook an extensive, novel series of experiments involving the Src kinase, and we now present this data in the revised manuscript. We sincerely hope that the reviewer is encouraged and satisfied with the new data and supports publication of the revised manuscript.

We wanted to make the point that our method is equally or more effective than previously published tools, but also that our method is more directed since we activate a single target of ROK (Sqh), due to the Nb-GFP interaction.

*We looked at dorsal closure as a readout and assessed the severity of phenotype (compare Fig. 3b with Fig. 3c as well as the bottom panel in Supp. Fig 2 showing SqhEE). These data clearly support the superior effectiveness of the novel tool compared to the tools available in the *drosophila* system.*

The comparison with other means of regulating ROK depends on the level expression of the reagent used. While the author shows that their approach is more effective with a given gal4 driver, their evidence depends on the choice of UAS-transgene. For example, the effectiveness

of NSlmb::vhhGFP4 could be improved by boosting expression, which is possible by changing the 5' or 3' UTR of the UAS vector (PMID: 22493255).

We thank the reviewer for this suggestion. We have not tried to boost the effectiveness of our system by playing around with 5' or 3' UTR sequences, but this is certainly a direction to be followed up. Other labs have started to undertake such efforts, often with the aim to increase efficiency in germ cells or in order to target the RNA (and thus the translation of the protein) to distinct cell compartments or distinct stages of oogenesis and distinct locations in the early embryos.

The authors go through great effort to limit the off target activity of N-Rok::vhhGFP4 by reducing expression and also by using a clever trick based on destabilised nanobody that is stabilised in the presence of its target. This is to be commended. However, the off-target assay is rather crude, as it relies on lethality. A better assay would be to measure global phosphorylation patterns by mass spec.

In my view, the best way to assess the specificity and effectiveness of an engineered kinase would be to compare phosphorylation of the target relative to that of non-specific proteins. This could be achieved by measuring the extent of p-Sqh relative to the increase in other phosphorylation events in the rest of the proteome. This could be achieved by mass spec and the result would greatly improve the quality of the manuscript, especially if the same was done with an existing means of activating phosphorylation of Sqh. I understand however that access to a suitable mass spec facility would be required for such an experiment.

We do agree with the reviewer that a global phosphoprotein analysis would be very helpful and insightful. We have initiated such studies, but since we work with embryos and spatial expression patterns of our effector proteins, we ran into several problems. We are still working on the improvement of such analyses, using more globally expressed driver lines, but we are unable to include any results in this first publication on such synthetic kinases. Instead, we have generated a second synthetic kinase in order to find out whether our approach is a more general one, and indeed found that a synthetic Src kinase can also be specifically targeted to a substrate via the Nb-GFP interaction.

Minor points

It is important to point out that this remains a GOF approach. It will be interesting to know if a reproducible response will be induced by expressing the engineered kinase at physiological level, in the absence of the endogenous gene. Could the authors comment?

The reviewer raises a very interesting point here. Yes, what we present here are GOF approaches. However, these engineered kinases also loose the specificity for other targets, so at the same time our approach includes a LOF aspect. Engineering kinases at their endogenous loci is possible, but in an animal model system as we use it here, this is expected to lead to lethality, since the kinase also loses its specificity towards targets other than the GFP fused

once. We have not considered to undertake such an approach, but this is an interesting avenue to think about.

Although the approach described has universal potential, the tools described are specific to ROK and the effort devoted to reducing off target effects are only relevant to this protein. This must be recognised.

Since the other reviewers asked us to find out whether we can use our approach to generate a second synthetic kinase, we set up experiments to design, generate and test a second large series of synthetic kinases centered around Src. Indeed, we found that such synthetic, activated Src kinases also showed a specificity guided by Nb_GFP interaction. This large series of experiments has now been included in the manuscript. Still, we would not dare to propose that such an approach is generally applicable. As we stated in the original manuscript, we still think that “our system should be carefully tested for each kinase-substrate pair separately, and some fine-tuning (i.e., optimizing the linker length and amino-acid composition) might be needed for optimal tool performance.”

Based on the varying potency of activity of the new Src constructs and the previously presented Rok data, we can now comment on a few points that can be useful for optimizing engineered kinase, including the reduction of off-target effects:

- 1) the differences in expression levels due to different insertion sites in the genome*
- 2) the type of the binder used (i.e., vhhGFP4 vs dGBP1 or 3G86.32 DARPin vs dGBP1 tested for Src)*
- 3) the position of the nanobody (N-terminal vs C-terminal)*
- 4) the mutations introduced in the regulatory sites (for example Y400 substitution (Y400E vs Y400D)) or the lack of manipulation at this site*

We propose that, together with the choice of the linker between the two engineered kinase parts, the features listed above can be used as a general guide for the generation and optimisation of other synthetic kinases based on our system or even other synthetic tools employing protein binders.

I am somewhat surprised by the observation that N-Rok::HAZH-86Fb without the fused nanobody (n=6) had no effect on the p-Sqh pattern. This protein remains dominant active, which phosphorylate proteins albeit non-specifically. I would expect phosphorylation of its cognate target to be increased (although to a much lesser extent than with the nanobody target system).

The reviewer is rightly pointing out that one could have expected that N-Rok::HAZH-86Fb without the fused nanobody could or should increase phosphorylation of Sqh. However, we did not observe this using the p-Sqh antibody (see Figure 2). It is possible that this synthetic fusion protein phosphorylates some protein in a non-specific manner, leading to lethality. In any case, we have now resorted to using unstable nanobodies for our constructs, in order to further reduce the possible low level, non-specific phosphorylation.

Reviewer #3 (Comments to the Authors (Required)):

Review of "In vivo regulation of fluorescent fusion proteins by engineered kinases" by Lepeta et al.

Regulation of cellular processes by phosphorylation/dephosphorylation reactions is extraordinarily common and extraordinarily important. While an enormous variety of tools are currently available for manipulation of such reactions, a large hole remains: regulation of phosphorylation of specific kinase substrates. At present, the go-to approach is to replace the relevant residues with either nonphosphorylatable amino acids (typically A) or phosphomimetic amino acids (D or E). The problems with this approach are that 1) it is not time resolved; 2) there is no guarantee that the mutations don't result in some other deficit in protein function and 3) for phosphomimetics, there is no guarantee that they actually mimic phosphorylation.

In this study, the authors describe an approach for ectopic upregulation of phosphorylation of specific proteins by targeting a constitutively active kinase (in this case, Rok) to a fluorescent protein-fused version of its substrate (in this case the regulatory light chain of myosin-2, aka RLC) via a nanobody. The authors detail a long list of experiments in which they vet this approach in *Drosophila* embryos and provide strong *in vivo* evidence that they can indeed specifically upregulate the phosphorylation of RLC with the most specific results being obtained via the use of a nanobody that is unstable in the absence of its fluorescent protein target. They show that ectopic RLC phosphorylation disrupts dorsal closure, as expected based on previous studies and further show that ectopic RLC phosphorylation perturbs tracheal development in a manner that depended on the timing and location of induction.

The results presented in this study are promising in that the authors appear to have developed an approach that overcomes the second and third problems alluded to above (they can truly increase phosphorylation of RLC without concerns about changing protein structure except insofar as the addition of the fluorescent protein to the substrate is required). It might be argued that without better time resolution and reversibility this approach will be of limited utility but it seems clear that this is an important first step. Similarly, it might be argued that it would have been more useful to develop this approach for targeted dephosphorylation, but phosphatases are more complex than kinases, typically comprising three subunits for full function, so that too seems like too much to ask.

There are, however, other points that should be addressed. Perhaps most importantly, it should be determined whether this approach will be generally useful with other substrates and kinases. A single additional example would suffice, and it would not require the detailed phenotypic work up used for RLC, but as it stands, it is impossible to assess how broadly applicable this approach will be.

We would like to thank the referee for her/his comments of appreciation concerning the approach we have developed. Since the referee wanted to see a second example of such an engineered kinase, we undertook an extensive, novel series of experiments involving the Src kinase, and we now present this data in the revised manuscript.

We have generated a second engineered kinase based on Src42A catalytic domain and the destabilized nanobody against GFP (dGBP1). We used it to activate the Drosophila ortholog of JNK kinase – Basket (Bsk) and show that engineered Src42A with dGBP1 efficiently phosphorylates Bsk-GFP and that this effect is dependent on the presence of the nanobody; Src kinase domain without a binder does not elicit increase in pJNK nor pY immunostaining signal levels. We also confirm that engineered Src hyper-phosphorylates only the GFP-fusion and not the endogenous, untagged JNK substrate.

In addition, to both bolster the authors interpretation of their results and to support the idea that this approach could be used to counteract loss-of-function conditions, it would be helpful to show that ectopic RLC phosphorylation can alleviate Flw overexpression.

We thank the reviewer for this interesting point. We have not yet done this experiment since we concentrate on the second engineered kinase, but we do agree it would be an interesting follow-up experiment.

Minor comments:

The text of the results section is frequently digressive. The authors should consider trimming it down to better emphasize the important points.

We have tried to follow the advice of the reviewer and have shorted and clarified many aspects in the revised version. Since we included a large set of new experimental data, we have also eliminated a number of sentences/paragraphs and tried to be as clear as possible in all the statements we make.

The authors may want to check out Elife 2016 Dec 30:5:e20828 which clearly shows that phosphomimetic mutations of RLC do not faithfully mimic true phosphorylation events when it comes to myosin-2 motor activity. This finding supports the potential utility of the approach employed here, particularly for RLC.

Indeed, this data supports our approach. We thank the reviewer for making us aware of this study.

June 15, 2022

RE: JCB Manuscript #202106179R

Dr. Markus Affolter
Biozentrum
Klingelbergstrasse 50/70
Basel

Dear Dr. Affolter:

Thank you for submitting your revised manuscript entitled "Engineered kinases as a novel tool for phosphorylation of selected targets in vivo". We would be happy to publish your paper in JCB pending final revisions necessary to meet our formatting guidelines (see details below). Congratulations on this superb paper!

A. MANUSCRIPT ORGANIZATION AND FORMATTING:

1) Text limits: Character count for Tools is < 40,000, not including spaces. Count includes abstract, introduction, results, discussion, and acknowledgments. Count does not include title page, figure legends, materials and methods, references, tables, or supplemental legends.

2) Figures limits: Tools may have up to 10 main text figures.

3) Figure formatting: Scale bars must be present on all microscopy images, including inset magnifications. Molecular weight or nucleic acid size markers must be included on all gel electrophoresis.

4) Statistical analysis: Error bars on graphic representations of numerical data must be clearly described in the figure legend. The number of independent data points (n) represented in a graph must be indicated in the legend. Statistical methods should be explained in full in the materials and methods. For figures presenting pooled data the statistical measure should be defined in the figure legends. Please also be sure to indicate the statistical tests used in each of your experiments (either in the figure legend itself or in a separate methods section) as well as the parameters of the test (for example, if you ran a t-test, please indicate if it was one- or two-sided, etc.). Also, if you used parametric tests, please indicate if the data distribution was tested for normality (and if so, how). If not, you must state something to the effect that "Data distribution was assumed to be normal but this was not formally tested."

5) Abstract and title: The abstract should be no longer than 160 words and should communicate the significance of the paper for a general audience. The title should be less than 100 characters including spaces. Make the title concise but accessible to a general readership.

* The term "novel" cannot be used in your title.

6) Materials and methods: Should be comprehensive and not simply reference a previous publication for details on how an experiment was performed. Please provide full descriptions in the text for readers who may not have access to referenced manuscripts.

7) Please be sure to provide the sequences for all of your primers/oligos and RNAi constructs in the materials and methods. You must also indicate in the methods the source, species, and catalog numbers (where appropriate) for all of your antibodies. Please also indicate the acquisition and quantification methods for immunoblotting/western blots.

8) Microscope image acquisition: The following information must be provided about the acquisition and processing of images:

- a. Make and model of microscope
- b. Type, magnification, and numerical aperture of the objective lenses
- c. Temperature
- d. Imaging medium
- e. Fluorochromes
- f. Camera make and model

g. Acquisition software

h. Any software used for image processing subsequent to data acquisition. Please include details and types of operations involved (e.g., type of deconvolution, 3D reconstitutions, surface or volume rendering, gamma adjustments, etc.).

9) * References: There is no limit to the number of references cited in a manuscript. References should be cited parenthetically in the text by author and year of publication. Abbreviate the names of journals according to PubMed.*

10) Supplemental materials: There are strict limits on the allowable amount of supplemental data. Tools may have up to 5 supplemental figures. Please also note that tables, like figures, should be provided as individual, editable files. A summary of all supplemental material should appear at the end of the Materials and methods section.

13) ORCID IDs: ORCID IDs are unique identifiers allowing researchers to create a record of their various scholarly contributions in a single place. At resubmission of your final files, please consider providing an ORCID ID for as many contributing authors as possible.

Please note that JCB now requires authors to submit Source Data used to generate figures containing gels and Western blots with all revised manuscripts. This Source Data consists of fully uncropped and unprocessed images for each gel/blot displayed in the main and supplemental figures. Since your paper includes cropped gel and/or blot images, please be sure to provide one Source Data file for each figure that contains gels and/or blots along with your revised manuscript files. File names for Source Data figures should be alphanumeric without any spaces or special characters (i.e., SourceDataF#, where F# refers to the associated main figure number or SourceDataFS# for those associated with Supplementary figures). The lanes of the gels/blots should be labeled as they are in the associated figure, the place where cropping was applied should be marked (with a box), and molecular weight/size standards should be labeled wherever possible.

B. FINAL FILES:

Thank you for this interesting contribution, we look forward to publishing your paper in Journal of Cell Biology.

Sincerely,

Kenneth Yamada, MD, PhD
Editor

Andrea L. Marat, PhD
Senior Scientific Editor

Journal of Cell Biology

Reviewer #1 (Comments to the Authors (Required)):

I want to congratulate the authors for their fantastic efforts to further improve this exciting manuscript. In particular, the newly added data for a second kinase domain, from Src, which even belongs to a different kinase class, the tyrosine kinases, now generalizes the strategy introduced in this manuscript how to selectively phosphorylate only a single chosen target in a cell type of choice at a chosen time in a living organism.

These new data are key to now support the general conclusions of this manuscript: a new strategy of using nanobodies against GFP or RFP fused to engineered kinase domains expressed in vivo together with GFP or RFP-fused substrates that phosphorylate these substrates only. This is an appealing strategy to a large audience. I recommend publication.

Minor:

1. For my taste the 'preview' of the results in the last page of introduction is now a bit too long with already referring to figures. The sections for both kinases could be fused to reduce redundancy and only summarize the technical achievement and less of the complex biological insights. The latter come again in detail in results and discussion sections. However, this is a matter of taste.
2. Figure 4C and Figure 8A appear to show identical schemes. Why do we need both? I suggest to either add the specific kinases to each or delete the second scheme.
3. page 17 last para: the text refers to Figure 8c and a reference to Figure 8b is missing. Also, the text referring to Figures 8d and e seems incorrect.

Reviewer #3 (Comments to the Authors (Required)):

The authors have made the essential addition to the manuscript needed to address my concern.